# RADAR: Benchmarking Language Models on Imperfect Tabular Data

**Ken Gu**[1,3,†], **Zhihan Zhang**[1,3*], **Kate Lin**[1*], **Yuwei Zhang**[1*], **Akshay Paruchuri**[1*],
**Hong Yu**[1*], **Mehran Kazemi**[2], **Kumar Ayush**[1], **A. Ali Heydari**[1], **Maxwell A Xu**[1],
**Girish Narayanswamy**[1,3], **Yun Liu**[1], **Ming-Zher Poh**[1], **Yuzhe Yang**[1], **Mark Malhotra**[1],
**Shwetak Patel**[1,3], **Hamid Palangi**[1], **Xuhai Xu**[1], **Daniel McDuff**[1], **Tim Althoff**[1,3], **Xin Liu**[1]

[1]Google Research    [2]Google DeepMind    [3] University of Washington
[†]Work done during an internship at Google    [*]Equal contribution
kengu@cs.washington.edu, {dmcudff, althoff, xliucs}@google.com

## Abstract

Language models (LMs) are increasingly being deployed to perform autonomous data analyses. However, their *data awareness*—the ability to recognize, reason over, and appropriately handle data artifacts such as missing values, outliers, and logical inconsistencies—remains underexplored. These artifacts are especially common in real-world tabular data and, if mishandled, can significantly compromise the validity of analytical conclusions. To address this gap, we present RADAR, a benchmark for systematically evaluating data-aware reasoning on tabular data. We develop a framework to simulate data artifacts via programmatic perturbations to enable targeted evaluation of model behavior. RADAR comprises 2980 table query pairs, grounded in real-world data spanning 9 domains and 5 data artifact types. In addition to evaluating artifact handling, RADAR systematically varies table size to study how reasoning performance holds when increasing table size. Our evaluation reveals that, despite decent performance on tables without data artifacts, frontier models degrade significantly when data artifacts are introduced, exposing critical gaps in their capacity for robust, data-aware analysis. Designed to be flexible and extensible, RADAR supports diverse perturbation types and controllable table sizes, offering a valuable resource for advancing tabular reasoning.[1]

## 1 Introduction

Language models (LMs) are increasingly deployed as autonomous data science agents, capable of performing basic data analyses on tabular data, such as summarizing trends, identifying relationships, and manipulating data [42, 36, 62, 13]. But can we truly rely on LMs for real-world data analysis? It remains unclear whether they are merely repeating templated analyses or engaging in genuine **data-aware reasoning**—making decisions based on the actual state and structure of the dataset, much like an experienced data scientist would (Fig. 1). This distinction is critical in real-world settings, where data artifacts such as missing values, outliers, and

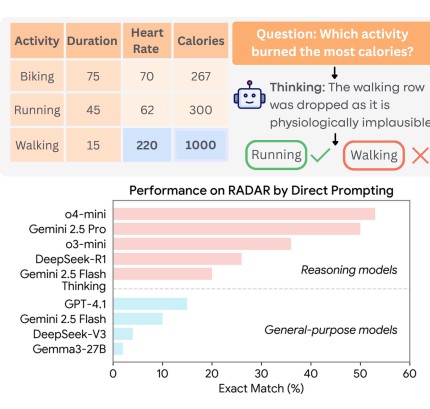

Figure 1: **Reasoning over tabular data containing data artifacts** and corresponding performance of language models.

---

[1] 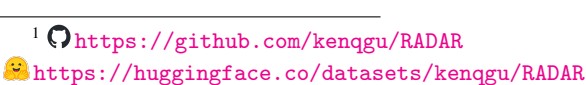 https://github.com/kenqgu/RADAR
🤗 https://huggingface.co/datasets/kenqgu/RADAR

39th Conference on Neural Information Processing Systems (NeurIPS 2025) Track on Datasets and Benchmarks.

logical inconsistencies are ubiquitous, non-trivial to detect, and can significantly affect analysis and interpretation [28, 27, 39, 23]. Understanding whether models can detect and appropriately respond to such artifacts—without explicit instruction and across diverse schema and domain contexts—is key to assessing their reliability.

In high-stakes settings, such errors can lead to harmful or misleading conclusions. For example, a patient's electronic health record might erroneously indicate a resting heart rate of 220 bpm (an implausible value for an adult) due to a data entry mistake or sensor malfunction. If left uncorrected, such an anomaly could trigger false automated alerts, lead to clinical misdiagnosis, prompt unnecessary or risky interventions, or distort downstream clinical research.

Evaluating **data awareness**—the ability to detect, navigate the implications of, and correct data artifacts—is both a practical necessity and a **challenging reasoning task for language models**. It requires more than surface-level understanding of the data types or table structure; models must apply nuanced reasoning over context (e.g., recognizing that 5 calories is implausible for a 60-minute run), units (e.g., distinguishing between kilometers and miles), statistical expectations (e.g., identifying a resting heart rate of 220 bpm as an outlier), and inter-column relationships (e.g., ensuring distance, duration, and speed are consistent). To succeed, models must learn the table schema and inter-column relationships, navigate long-context inputs in large tables, resist distractors, reason inductively from examples, identify subtle anomalies such as outliers or missingness, and execute appropriate data analysis and interpretation in the presence of data artifacts. Yet, despite its importance, data-aware reasoning—and how it scales with larger, more complex tables—remains underexplored. Larger tables require models to reason over longer contexts, filter irrelevant data, and track complex dependencies across columns and rows. These demands compound the challenge of robust tabular reasoning.

Prior work has primarily focused on tabular reasoning or analysis execution over "clean" data tables, where data awareness is neither required nor explicitly evaluated. While recent efforts have explored structural perturbations—such as shuffling rows or columns [2, 74, 55]—they do not require semantic understanding of the table (e.g., a New York City borough column is mismatched with the borough identifier column). Moreover, they often overlook key factors such as table size, leaving open questions about how tabular reasoning scales with increasing input complexity. In contrast, real-world analysis demands understanding the domain of the data to detect imperfections, adapting to noisy or inconsistent inputs, and adjusting both interpretation and analysis strategies accordingly.

In this work, we introduce the **R**obust **A**nd **D**ata **A**ware **R**easoning (RADAR) benchmark (Fig. 2), an evaluation framework that systematically assess the reasoning capabilities of models in the presence of challenging data artifacts that demand **semantic and schema-level understanding**. RADAR assesses the extent to which models can correctly recognize and handle specific types of data artifacts and analyzes how this capability varies across different models, dataset sizes, and artifact types. RADAR specifically addresses five types of data artifact common in real world data [43, 38, 16, 9, 25]: 1) **Missing Data**: Empty or null entries simulating incomplete information; 2) **Bad Values**: Clearly erroneous or placeholder entries (e.g., -1, 9999, TEST, #REF!); 3) **Outliers**: Implausible extreme values that distort analysis (e.g., 220 bpm resting heart rate); 4) **Inconsistent Formatting**: Variations in representing the same value (e.g., 22 lbs, 22 pounds, weight = 22); and 5) **Inconsistent Logic**: Cross-field contradictions violating common-sense logic (e.g., end time before start time).

To evaluate data-aware tabular reasoning in realistic settings, we constructed RADAR by crowdsourcing 53 diverse datasets and queries from domains like education, health, and business, focusing on large, complex tables (§3). We implemented 260 query-specific perturbation functions to inject realistic, context-sensitive artifacts (e.g., an implausibly high calories burned for a short walk), ensuring that naive computations on these modified tables lead to incorrect answers. This, in turn, supports large-scale generation of verifiable, high-quality task instances. We benchmark language models on RADAR under both direct prompting and code-agent settings (§4), revealing that while models like o4-mini achieve 100% on clean tables aided with code execution, performance drops by 59% on perturbed ones (§5), highlighting critical limitations in only evaluating on clean data. In summary, RADAR provides a precise, scalable, objective, and systematic approach to automatically assessing LM tabular reasoning. Our main contributions are:

1. **A scalable framework for data-aware QA generation.** We introduce a dataset-agnostic framework for automatically synthesizing thousands of high-quality, rigorously verified table QA pairs targeting artifact-sensitive reasoning. The framework is dataset-agnostic and extensible to new

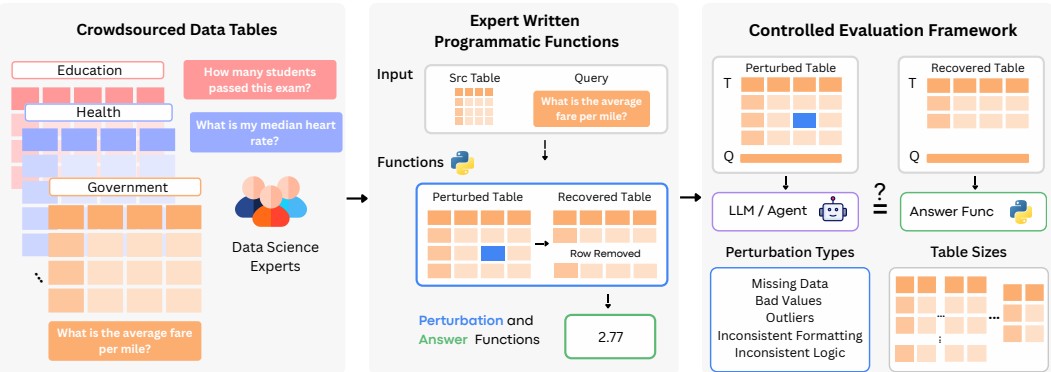

Figure 2: **Overview of RADAR**. Expert-written programmatic functions are used to: (1) **generate ground truth answers** (via answer functions invariant to table dimensions), and (2) **simulate data artifacts** by producing perturbed and recovered versions of the original table. We evaluate LMs on perturbed tables by computing the ground-truth answer over the corresponding recovered table, enabling a controlled and consistent evaluation across data artifact types and varying table sizes.

tables, larger data scales, and additional artifact types, enabling fine-grained and comprehensive evaluation of data awareness.

2. **A benchmark of expert-curated real-world data tables and expert-authored contextual perturbations.**
   We construct RADAR, a suite of 53 table QA tasks comprising 2980 instances across 9 application domains vetted by data science experts. Each task is paired with a library of hand-engineered perturbation functions that inject five classes of data artifacts designed to precisely evaluate models' data awareness.

3. **A large-scale analysis of LM data-aware reasoning uncovering implications for building real-world data science agents.** We perform a comprehensive study on how data artifacts and table size influence the reasoning accuracy of language models. Our results uncover novel and systematic failure modes, including pronounced brittleness to logically inconsistent entries. Consequently, we provide concrete guidelines for developing more reliable LMs.

## 2 Background and Related Work

**Tabular Reasoning in Language Models.** Language models show promise in answering natural language queries over structured data (i.e., table QA) [69, 61]. Existing benchmarks assess reasoning over tabular data [69, 44, 8, 55], but typically assume clean, artifact-free tables. As a result, they offer limited insight on how models handle imperfect real-world tables [19, 52, 43, 38]. Recent work studied the robustness of LMs to table perturbations [2, 74, 40, 56, 68]. In the context of table QA, these efforts largely involve structural modifications (e.g., row or column shuffling) or changes to cells containing the answer. These works test whether models rely on the actual table content, or lean on their parametric knowledge or position-based heuristics [75, 3, 34]. While valuable, these evaluations do not focus on whether models identify and reason about data quality issues.

RADAR advances this line of research by introducing perturbations aimed at evaluating a challenging and practically relevant notion of robustness: whether models can *detect* and appropriately *handle* semantically-grounded table-informed data artifacts during analytical reasoning. For example, a model should recognize that a taxi ride with a short duration but an extremely long distance is implausible—and take appropriate action such as excluding it from a fare average calculation.[2] See Appendix B.1 for additional task examples.

To support such evaluations, we crowdsource complex, domain-diverse tables and introduce schema-aware perturbations that reflect the scale, messiness, and irregularities typical of real-world data. Moreover, little is known about how model performance scales with table size, despite real-world data

---

[2]To support objective evaluation, we design tasks such that there is an objective correct action.

often spanning hundreds or thousands of rows and requiring long-context reasoning. Existing benchmarks, however, typically use small tables and do not support control over table sizes. RADAR fills this gap by enabling systematic variation in table size while holding task semantics and complexity constant, drawing inspiration from recent long-context reasoning benchmarks [54, 72, 33, 20]. This allows precise evaluation of how table size affects tabular reasoning performance.

**Data Analysis with Language Models.** Beyond table QA, LMs are increasingly being developed as general-purpose data science agents, capable of writing code, using tools, and executing end-to-end analyses [42, 36, 13, 62, 15], along with a wave of new benchmarks [70, 18, 21, 35, 73, 7]. However, existing benchmarks often assume clean datasets and none systematically assess whether models are resistant to data quality issues that precede any valid analysis, a foundational first step for reliable inference and reproducibility [17, 58]. RADAR complements these efforts by introducing schema-aware data imperfections—capturing real-world challenges such as missing data, outliers, formatting irregularities, and logical inconsistencies. This enables a targeted evaluation of whether models are truly *data-aware*: capable of detecting and responding to the kinds of imperfections that frequently compromise real-world data analyses.

# 3 The RADAR Benchmark

We design RADAR around three core goals: (1) **Enable objective evaluation**—ensuring that model performance is assessed reliably, transparently, and deterministically through unambiguous tasks and objective evaluation criteria. This contrasts with language model based evaluation, which can introduce stochasticity and undesirable biases [67, 30]; (2) **Support realistic and challenging tabular reasoning**—designing tasks that reflect real-world data imperfections and are sufficiently difficult to differentiate model capabilities and meaningfully measure progress; and (3) **Isolate the effects of table size and artifact type**—enabling fine-grained analysis of how specific data artifacts or table size impact reasoning, without introducing confounding factors. These goals are supported by our task generation framework (§3.1) and rigorous data collection procedure (§3.2).

## 3.1 Data Artifact Generation and Evaluation Framework

**Problem Definition.** We evaluate the robustness of language models when answering data analysis questions over perturbed tables containing realistic artifacts. To do so, we crowdsource high-quality, complex data tables and create programmatic functions to introduce data artifacts (Fig. 2).

Let $T$ denote a clean source table—unperturbed, logically consistent, and free from data artifacts. From $T$, we derive a perturbed version $T_p$ by introducing targeted artifacts, and a corresponding cleaned (recovered) version $T_r$ that reflects the intended correction of those artifacts. The set of cleaning operations is denoted by $\Delta T = T_p - T_r$ and may involve dropping rows (e.g., removing a row containing invalid values such as a negative fare) and/or overwriting cell values (e.g., recovering a missing BMI value from the corresponding height and weight columns). During task construction, we ensure that these are the only set of corrective actions. See Appendix B.2 for details.

Given a natural language query $Q$ (e.g., "*What is the average fare per mile?*") and a table $T$, let $f : (Q, T) \mapsto A$ denote a programmatic function that computes the correct answer $A$ by applying the logical operations implied by $Q$ over the contents of $T$. We assume that $Q$ is clear, unambiguous, and specifies an objective question whose answer can be deterministically computed from $T$.

The ground truth answer is defined as $A = f(Q, T_r)$. We assess robustness by checking whether the model's prediction satisfies $\text{LM}(Q, T_p) = A$. In other words, obtaining the correct answer on $T_p$ requires recognizing, reasoning over, and cleaning data artifacts (executing $\Delta T$) such that a model could compute a correct answer over the recovered table $T_r$.

**Programmatic Perturbations.** Programmatic perturbations introduce the *same* type of data artifacts across tables of varying sizes sharing the same schema. Combined with the answer function, this enables automatic generation of many high-quality tasks with verifiable ground-truth answers, thus allowing for controlled evaluation of how models' data awareness reasoning varies with table sizes.

We define a programmatic perturbation function as $g : (T, Q) \mapsto (T_p, T_r)$. Given a clean table $T$ and a query $Q$, the function $g$ generates a perturbed table $T_p$ by introducing artifacts targeted with respect to $Q$, along with $T_r$, the recovered table after handling the artifact. Each perturbation is

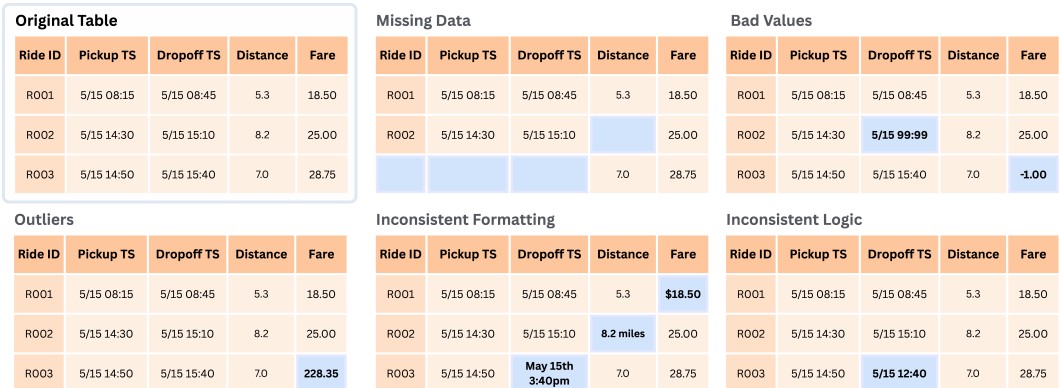

Figure 3: **Data Artifact Types**. Given a table $T$ without artifacts and a query $Q$ (e.g., *"What is the average fare per mile?"*), we perturb tables to simulate different data artifacts.

constructed so that directly applying $f(Q, T_p)$ yields an incorrect or undefined result (e.g., due to execution errors), thus requiring the artifact to be addressed for correct reasoning.[3] For example, if $Q$ requires filtering for rides longer than 6 miles, the perturbation will ensure it affects at least rows where the distance exceeds 6 miles.

**Data Artifact Types.** Using $g$, we programmatically introduce the following data artifact types:

1. **Missing Data**: Replacing valid cell entries with empty values (e.g., an empty string).
2. **Bad Values**: Injecting clearly erroneous or placeholder values that reflect data entry mistakes or system artifacts (e.g., -1, 9999, TEST, #REF!).
3. **Outliers**: Inserting extreme, contextually implausible values into numeric fields (e.g., 220 bpm for resting heart rate, or a taxi fare of 10,000 USD for a short ride).
4. **Inconsistent Cell Formatting**: Variations in units, formats, or styles (e.g., 22 lbs, 22 pounds, weight = 22) or inconsistent dates/casing that represent the same data.
5. **Inconsistent Logic**: Introducing contradictions across table cells that violate internal consistency (e.g., an end timestamp earlier than the start time, a mismatched BMI value given the height and weight, or a runner's rank that does not correspond with their finish time), requiring multi-column and/or multi-row reasoning to detect.

In this work, we study data artifact types independently (e.g., Fig. 3). However, the answer and perturbation functions ($f$ and $g$) are flexible and can easily extend to incorporate additional artifact types or combinations of artifact types within a single table.

**Scalability Across Table Sizes.** Task complexity may vary with table size as larger tables require operating over a larger context. To support different table sizes, we define the core schema $\mathcal{C}$ as the set of columns comprising (i) the fields required to answer the query $Q$, and (ii) any auxiliary fields used to introduce perturbations.

Both the perturbation function $g$ and the answer function $f$ are programmatically defined to operate over tables with varying numbers of rows and columns, as long as the required fields in $\mathcal{C}$ are present. Given a cleaned table $T^{(n,m)}$ with $n$ rows and $m$ columns (with $m \geq |\mathcal{C}|$), $g$ generates a perturbed version $T_p^{(n,m)}$ and a recovered counterpart $T_r^{(n_r \leq n, m)}$. The function $f$ computes the answer $A$ consistently across these variants, provided that $\mathcal{C}$ is preserved.

As a result, both the perturbations and corresponding ground truth function remain agnostic to table size by design, as long as tables of sufficient size can be collected or generated. This scalability enables data awareness evaluations to be systematically extended across diverse table sizes and expanded schemas. In this setup, the perturbation logic remains fixed, and the primary axis of variation is table size—whether in the number of rows or the number of auxiliary columns—allowing for consistent and controlled evaluation across different table configurations.

---

[3]While our framework supports multiple cleaned tables to reflect different valid ways of handling a perturbed table, we refer to a single cleaned table in our description for simplicity.

## 3.2 Constructing RADAR

**Crowdsourced Data Tables.** To build a challenging benchmark grounded in real-world data analysis scenarios, we recruited 12 data science experts, each holding graduate degrees in computer science, statistics, and related fields. Experts were encouraged to draw on their personal experiences to design examples of logical inconsistencies and analytical failures that frontier LMs struggle with.

Next, experts curated clean, publicly available source tables, each denoted as $T_s$, containing at least 500 rows and 20 columns, along with an associated natural language query $Q$. See Appendix A for data collection instructions and dataset details. For each selected table, experts ensured that the data was "clean"—free from pre-existing artifacts or inconsistencies. In many cases, this involved manually cleaning, wrangling, and normalizing data to ensure a blank canvas before introducing controlled perturbations. Each $(T_s, Q)$ pair defines a unique task in RADAR and includes annotations for the relevant core schema columns $\mathcal{C}$.

**Expert Written Programmatic Functions.** For each collected task, a dedicated team of experts authored the corresponding answer functions ($f$) and perturbation functions ($g$), with one $g$ defined for each data artifact type (e.g., missing data, outliers etc.), where applicable given the core schema $\mathcal{C}$. Each perturbation function $g$ is programmatically implemented in Python and affects a minimum of one row or cell to ensure meaningful change, but is limited to at most 10% of rows to preserve the overall integrity of the table and prevent more significant changes to the original data distribution. All functions, tables, and annotations underwent multiple rounds of expert code review and cross-validation to ensure correctness and logical consistency across table sizes and perturbation instances.

**Ensuring Objective Tasks.** To ensure RADAR provides a robust evaluation of data awareness, we implemented several safeguards. Each query-artifact pair was reviewed by multiple experts to ensure unambiguous recovery or discard behavior. Artifacts were only included if multiple experts unanimously agreed that it needed to be addressed because it contradicted internal or commonsense logic, or clearly violated established

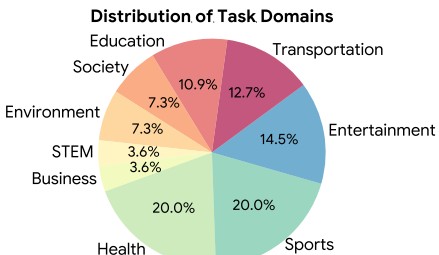

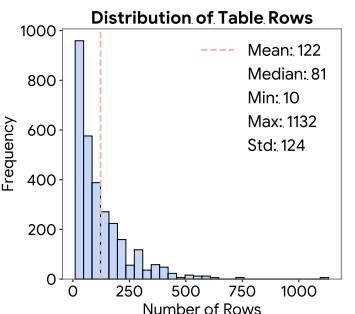

Figure 4: **Data Statistics of RADAR.**

domain expectations. Furthermore, multiple rounds of code review and refinement were conducted to confirm that all artifacts were objectively erroneous, unambiguous, and solvable. This process also involved making assumptions explicit in the query or updating perturbation functions when necessary. Perturbations were applied to a very small subset of rows (typically fewer than 5%), and only one perturbation type was used per task instance to preserve a single, unambiguous corrective action. Collectively, these measures position RADAR as an objective and reliable benchmark. We include examples of our iterative task refinement process in Appendix B.2.

**Generating Tables and Task Instances.** To support controlled evaluation of data awareness across table sizes, we use our framework to generate table variants from each clean source table $T_s$. We measure table size in token count $\tau \in 2K, 4K, 8K, 16K$, using the Gemma 3 tokenizer [14] applied to the CSV-serialized form of the table.[4] To explore the

| Dataset | Tasks | Instances | Tokens (K) | Cols |
|---------|-------|-----------|------------|------|
| RADAR   | 53    | 2,980     | [2,4,8,16] | [5,10,20] |
| RADAR-T | 53    | 313       | 8          | 10 |
| RADAR-S | 10    | 720       | [2,4,8,16] | [5,10,20] |

Table 1: **Summary of RADAR Dataset Splits.**

impact of table dimensions (i.e., row-to-column ratio), we additionally control the number of columns, $c \in 5, 10, 20$. Given $T_s$, for each $(\tau, c)$ combination, we select the number of rows $R$ such that the resulting table with $c$ columns, contains approximately $\tau$ tokens (i.e., $R = \arg\min_r |\text{tok}(T_s, r, c) - \tau|$).

---

[4]Row and column counts miss variation in cell content, ranging from single-token integers to paragraphs.

Here, $\text{tok}(T_s, r, c)$ denotes the token count for a sub-sample of $T_s$ with $r$ rows and $c$ columns. All generated tables from $T_s$ have the same core schema $\mathcal{C}$.

Once tables are selected across these configurations, we apply our perturbation functions (across data artifact types) to create perturbed data tables. A task instance in RADAR is thus defined by a tuple consisting of the task, token-based table size, column count, and data artifact type.

**Dataset Summary.** Figure 4 and Table 1 summarize our benchmark. RADAR comprises 53 expert-curated tasks derived from 27 source tables spanning 9 domains such as education, STEM, and health (details in Appendix A). Among 53 tasks, 6 table artifact variants (5 perturbed, 1 clean), 4 table sizes, and 3 column counts (yielding $53 \times 6 \times 4 \times 3 = 3816$ possible combinations), RADAR includes 2,980 systematically generated task instances[5] and 260 expert-written perturbation functions.

To encourage adoption and enable more tractable evaluation while maintaining sufficient statistical power to distinguish model performance, we curate two benchmark subsets, also used in our evaluation (§4): (1) RADAR-T, focuses on capturing maximal variation across tasks and data artifacts; and (2) RADAR-S which isolates the influence of table size and dimensionality. RADAR-T includes all tasks and their associated data artifact variants, standardized to tables with 10 columns and approximately 8K tokens. RADAR-S is a subset of 10 tasks with complete artifact variants and all table size configurations.

## 4 Experiments

**Models.** We evaluate a range of models on *data awareness* including table-tuned models, open-source models, and popular general-purpose and reasoning models. For models trained specifically for data tables, we include TableGPT2-7B [59] and StructLM-7B [76]. For general-purpose models, we include GPT-4.1, Gemini 2.5 Flash, Gemma 3 27B [14], and DeepSeek-V3 [11]. For reasoning models, we include o3-mini, o4-mini,[6] Gemini 2.5 Flash with thinking on, Gemini 2.5 Pro, and DeepSeek-R1 [12]. We evaluate all models on RADAR-T. To study the impact of table size on performance, we evaluate Gemini 2.5 Flash, Gemini 2.5 Flash with thinking on, GPT-4.1, and o4-mini on RADAR-S.

**Baselines.** We evaluate models using two zero-shot baseline approaches: *direct prompting*, where LMs answer questions in a single turn via textual reasoning given the input prompt [2, 34], and *code agent*, where LMs are equipped with a Python shell tool to interact with the data table and observe execution outputs. The code agent is based on existing tool-use agents [71, 34]. Due to StructLM's specialized prompt format and authors noting it "not designed for agentic settings",[7] we test it only under direct prompting with their table format [76]. In our experiments, we limit the number of interaction steps to five, following [34]. In practice, this constraint had negligible impact, as the agent reached the step cap in only 2.7% of cases. Importantly, for both baselines in the system prompt, **we explicitly instruct the model to pay attention to all five data artifact types during its reasoning** but without referring to any specific table instance. The prompt also describes the appropriate corrective actions, i.e., recovering flawed data when possible, and discarding irrecoverable rows. The full table is serialized in CSV format and included in the prompt. Complete prompts and baseline implementation details (including additional experiments) are in Appendix C.

**Metrics.** We primarily evaluate model performance using Exact Match (EM) accuracy. For most experiments, a prediction is considered correct only if it exactly matches the ground truth: string, integer, and list outputs must be identical, and floating-point values must fall within ±1 unit of the least significant decimal place of the ground truth value. If multiple answers are equally valid, we include all acceptable options in the ground truth and count any match as correct. For example, in `weather-city-mixup` (see Table 5 for all tasks), with query "*What is the difference in temperature between Australia's warmest city and America's warmest city in February?*", the ground truth is any one of 15.1, 8.4, -15.1, -8.4, allowing for Fahrenheit/Celsius calculations and both subtraction directions. We also accept values in the ranges [15.0,15.2], [8.3,8.5], [-15.2,-15.0], and [-8.5,-8.3] to account for rounding.

---

[5]Some combinations of artifact, table size, and column count were infeasible due to query/table constraints.
[6]o4-mini and o3-mini are evaluated with reasoning effort set to *high*
[7]StructLM authors noted this during email correspondence

| Model | \multicolumn{6}{c}{Direct Prompting} | | | | | | \multicolumn{6}{c}{Code Agent} | | | | | |
|---|---|---|---|---|---|---|---|---|---|---|---|---|
| | Cln | Miss | Bad | Out | Fmt | Log | Cln | Miss | Bad | Out | Fmt | Log |
| *Table-tuned Models* | | | | | | | | | | | | |
| StructLM | 2.3 | 0.8 | 0.4 | 0.4 | 1.1 | 1.5 | – | – | – | – | – | – |
| TableGPT2 | 0.8 | 0.8 | 1.1 | 1.3 | 1.5 | 0.8 | 35 | 12 | 3.8 | 5 | 7.5 | 6 |
| *General-purpose Models* | | | | | | | | | | | | |
| Gemma3 27B | 1.9 | 3.8 | 1.9 | 0 | 1.9 | 1.8 | 75.5 | 20.8 | 9.4 | 8.3 | 15.1 | 14.3 |
| DeepSeek-V3 | 1.9 | 5.7 | 3.8 | 4.2 | 3.8 | 0.0 | 96.2 | 35.8 | 41.5 | 31.2 | 54.7 | 25.0 |
| Gemini 2.5 Flash | 15.1 | 7.5 | 11.3 | 6.2 | 13.2 | 8.9 | 84.9 | 28.3 | 26.4 | 27.1 | 54.7 | 17.9 |
| GPT-4.1 | 17.0 | 13.2 | 11.3 | 6.2 | 28.3 | 12.5 | 98.1 | 37.7 | 41.5 | 60.4 | 73.6 | 30.4 |
| *Reasoning Models* | | | | | | | | | | | | |
| Gemini 2.5 Flash Thinking | 39.6 | 17.0 | 20.8 | 22.9 | 22.6 | 16.1 | 88.7 | 41.5 | 35.8 | 56.2 | 47.2 | 35.7 |
| DeepSeek-R1 | 34.0 | 22.6 | 32.1 | 25.0 | 32.1 | 19.6 | 84.9 | 39.6 | 47.2 | 64.6 | 64.2 | **48.2** |
| o3-mini (high) | 73.6 | 37.7 | 37.7 | 18.8 | 66.0 | 17.9 | 75.5 | 43.4 | 43.4 | 33.3 | **79.2** | 25.0 |
| Gemini 2.5 Pro | 71.7 | **50.9** | 56.6 | 47.9 | 56.6 | **42.9** | 84.9 | **73.6** | 54.7 | 64.6 | 73.6 | 44.6 |
| o4-mini (high) | **83.0** | 49.1 | **58.5** | **56.2** | **73.6** | 32.1 | **100** | 50.9 | **54.7** | **83.3** | **79.2** | 41.1 |

Table 2: **Zero-shot Performance by Data Artifacts.** Values are Exact Match accuracy (%). Columns are grouped by the direct prompting and code agent baselines. Cln=Clean, Miss=Missing Data, Bad=Bad Values, Out=Outliers, Fmt=Inconsistent Formatting, Log=Inconsistent Logic.

In the case of table-tuned LMs, we observe that minor formatting inconsistencies (e.g., extra punctuation or whitespace) can obscure otherwise correct reasoning. To ensure the metric reflects reasoning quality rather than formatting errors, we adopt a relaxed matching criterion for these models: if any ground truth answer appears within the output string, the prediction is counted as correct.

## 5   Results & Discussion

In this section, we discuss our primary results. Additional results, including those with error bars and further analyses are provided in Appendix E.

**RQ1: How do models handle different types of data artifacts?** Table 2 summarizes the zero-shot exact match accuracy across different artifact types. In direct prompting, most models struggle, with only the strongest reasoning models—such as o4-mini and Gemini 2.5 Pro—consistently surpassing 50% accuracy. o4-mini performs reasonably well on clean, unperturbed tables and artifacts involving bad values, outliers, and formatting inconsistencies, while Gemini 2.5 Pro is comparatively better at han-

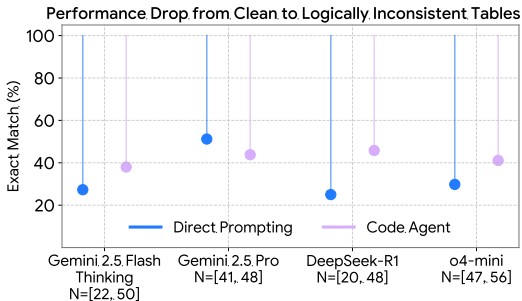

Figure 5: **Frontier models struggle with logically inconsistent tables, despite clean-table success.** Exact match scores on logically inconsistent tables on tasks where the model answered correctly on the clean table (indicated by N).

dling logical inconsistencies and missing values. General-purpose models, particularly open-source ones, tend to perform less well—often scoring below 20% and 6%, respectively—highlighting the overall difficulty of RADAR as a tabular reasoning benchmark.

Introducing code execution leads to performance gains across all models, particularly for general-purpose models and those less effective with direct prompting. With access to code, all models perform reasonably well on clean tables. However, significant performance gaps persist—especially for general-purpose models—when faced with tables containing artifacts. This suggests data awareness remains a core challenge even with code. Qualitative analysis shows that code execution enables models to bypass deep understanding by applying generic routines (e.g., dropping rows with missing data) without necessarily identifying the exact issues (see Appendix D for examples). These findings reveal a persistent disparity in models' ability to detect and recover from data artifacts, raising the question whether analysis agents should incorporate elements of specialized reasoning models.

**RQ2: Do models maintain performance on perturbed tables, even after succeeding on clean ones?** We investigate whether models can generalize their success from clean tables to perturbed

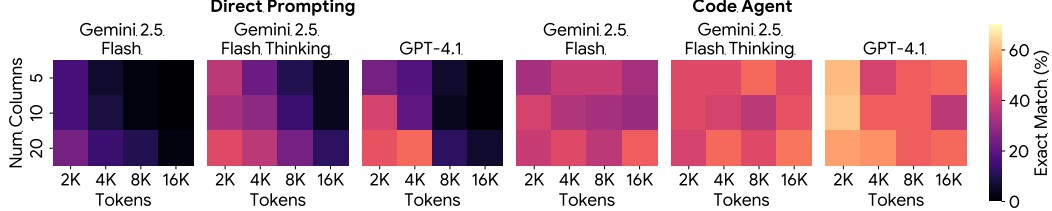

Figure 6: **Scaling Performance on Tables with Artifacts.** Exact match scores for tables varying in token and column count. In the direct prompting baseline (left), performance consistently drops as table token count increases, though models tend to perform better on wider tables (with fewer rows) with the same token count. In contrast, code agent performance (right) remains relatively stable across different token and column counts.

versions of those same tables. Focusing on just logically inconsistent tables as an example, for the strongest performing models, we select the subset of tasks it answers correctly on the clean table variant and evaluate accuracy on the associated logically inconsistent table. Since clean-table performance varies across models, each is evaluated on a different subset of examples. As shown in Figure 5, performance consistently drops across all models. This trend holds for both direct prompting and code agent baselines, suggesting that current models, even when equipped with code execution capabilities, are not robust to subtle logical inconsistencies. We report performance degradation for other artifact types in Appendix E.1.

**RQ3: How does performance scale with table size?** Figure 6 shows the exact match accuracy for both direct prompting and code agents for different table sizes, measured in token count as well as different number of columns given the same token count (i.e., wider vs. narrower tables). For direct prompting, we observe that performance decreases substantially as the table size increases and consistent across models. By 16K tokens, exact match accuracy drops to nearly zero. Interestingly, performance also declines as the number of columns decreases—that is, models perform better on wider tables (with fewer rows) at the same token count. Qualitative inspection of model reasoning traces indicate that models tend to go row by row when inspecting and performing calculations on the data table (see Appendix D for examples). This means that the number of tokens used for computation scales linearly with the number of rows (generally observed in Fig. 7). In contrast, code agents exhibit stable performance as table size and width vary. Their ability to offload computations and focus on intermediate reasoning allows them to remain largely unaffected by table dimensions. These findings highlight the benefits of token-efficient schema abstraction and the need for programmatic strategies to surface data artifacts. Without such mechanisms, even frontier language models struggle under the dual pressures of input scale and noise.

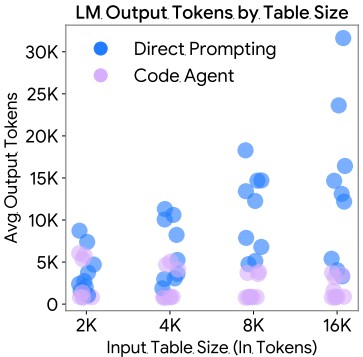

Figure 7: **LM output tokens (completion + reasoning) on RADAR-S.** Each point shows an LM on tables of a specific size (cols: 5, 10, or 20; tokens: 2K, 4K, 8K, or 16K tokens). Output tokens increase with table size for direct prompting and remain stable for code agents.

**RQ4: How does model performance vary across different types of table correction behaviors?**

To understand how well models handle different types of required corrections, we examine performance on RADAR-T, split into task instances that either (1) require the model to derive or replace cell values based on information from other rows or columns (i.e., excluding inconsistent formatting), or (2) drop rows entirely when such derivation is not possible. Table 3 shows that among the top-

| Model | Direct Prompting | | Code Agent | |
|---|---|---|---|---|
| | Derivation | Row Drop | Derivation | Row Drop |
| Gemini 2.5 Pro | **50.5** | 47.1 | **56.2** | 62.4 |
| DeepSeek-R1 | 22.9 | 26.5 | 31.4 | 66.7 |
| o3-mini | 31.4 | 25.8 | 35.6 | 37.3 |
| o4-mini | 47.6 | **49.0** | 44.8 | **67.6** |

Table 3: **Performance on Value Derivation vs. Row Dropping**. Exact match (%) on RADAR-T instances requiring either (1) deriving or replacing cell values (excluding formatting data artifacts, N=102), or (2) dropping rows (N=105).

performing models under direct prompting, some perform slightly better on derivation tasks, while others do better on row dropping, with no consistent trend. However, code agents lead to general improvements in row drop performance compared to derivation. In contrast, value derivation tasks remain challenging even for code agents, suggesting standardized code operations do not make it substantially easier to surface multi-column/row logical inconsistencies.

**RQ5: How does test-time compute affect performance?**

We analyze the number of LM output (completion + thinking) tokens and its relationship to performance. Under direct prompting, models generally perform better with more test-time compute (Fig. 8a). However, this is not a direct relationship. o4-mini, for example, performs better then o3-mini and while using less tokens. In contrast, when equipped with code, additional output tokens offer less optimal gains. This is observed in both the competitive performance of general-purpose models (Fig. 8a) and on o4-mini as we increase thinking budget (Fig. 8b).

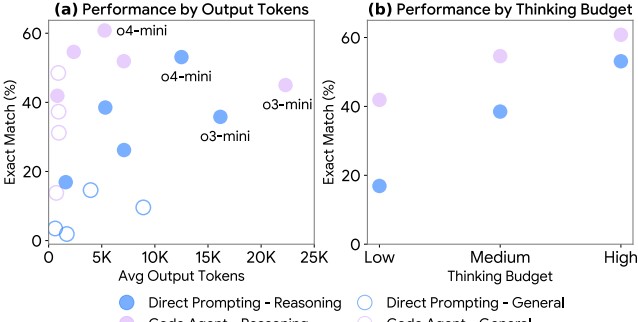

Figure 8: Exact match accuracy (aggregated across perturbed tables, N=260) on RADAR-T: **(a)** Accuracy as a function of LM output (completion + reasoning) tokens. Each point represents a specific model and baseline. **(b)** Accuracy on o4-mini with increasing thinking budget (Low, Medium, High).

Developing systems that better coordinate between text tabular reasoning and code execution could significantly improve performance and efficiency under token constraints.

**Limitations and Future Work.** RADAR currently supports a fixed set of perturbation types, each introduced independently to enable controlled and comprehensive evaluation. While these perturbations reflect common data artifacts, they are not exhaustive (e.g., sampling bias artifacts). In addition, to support objective answers, RADAR is scoped to a fixed set of operations for correctly handling data artifacts with exact match as an evaluation. This design excludes scenarios where multiple plausible corrections exist or where more complex reasoning is required [58]. Nevertheless, the framework underlying RADAR—which leverages programmatic functions to generate perturbed and corresponding recovered tables—is general and extensible to accommodate these limitations. The framework can include additional artifact types and allows for the combination of multiple artifact types within a single table. Future work can also build on this framework by expanding the space of corrective actions—e.g., incorporating more flexible success criteria [18] or accepting a broader range of valid recovered tables for a given perturbation.

# 6  Conclusion

We present RADAR, a benchmark for evaluating data-aware reasoning in language models across varying table sizes. By programmatically injecting realistic data artifacts, RADAR exposes critical gaps in model robustness. Our experiments show that while models perform well on clean tables, their performance degrades substantially in the presence of data artifacts. Although code execution can assist with certain computations, it is not a comprehensive solution. Our findings underscore the importance of designing agents that balance token efficiency with robust, data-aware reasoning.

# 7  Acknowledgments

We sincerely thank Ahmed Metwally, Van Fan, and Majd Bakar for their insightful feedback, thoughtful discussions, and review of this paper. Their contributions improved the clarity and quality of this work.

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

# Appendix Table of Contents

# A Dataset Details

Data science experts were recruited from a large data-driven software company. Since one of our key contributions is the corpus of tasks, we invited our experts to be co-authors of this paper. Table 4 summarizes the original source datasets in RADAR. Table 5 summarizes each task. Below, we provide a simplified but complete version of the original task construction instructions. A second team of experts manually reviewed all source tables and queries to ensure instructions were adequately followed.

---

### Instructions for Collecting Source Tables

Come up with a task involving a query, clean data table, and perturbed data table that frontier language models would get wrong when the data is perturbed.
To construct a task instance in our benchmark, please follow these steps:

**1. Dataset Selection**

- Find a real-world tabular dataset (e.g., NBA Stats 2023–24).
- The data table should contain at least 500 rows and ideally at least 20 columns.
- With the data table you also need to identify a query on the data coupled with a note of a logical inconsistency on the dataset.

**2. Clean Data**

- Explore and wrangle the dataset such that it is free of data artifacts. Data artifacts are ...
- If the original source table is less than 20 columns, generate relevant and consistent additional columns until there are at least 20.

**3. Return Data Sources**
Return the following data which constitute a full task

- `data.csv` – cleaned table relevant to the query
- `metadata.yaml` – containing the query and other relevant metadata. See below example.
- Document any inconsistent logic perturbation in the `logic_perturbation_note` field.

Example `metadata.yaml`:

```yaml
task_id: nba-player-least-3p-made
query: 'Among players averaging >= 10 PPG, who made the fewest 3-pointers?'
query_cols: ['Player', '3P']
minimum_columns: ['3P', '2P', 'FT', 'PTS']
id_columns: ['Player']
dataset_source: 'https://www.kaggle.com/datasets/vivovinco/2023-2024-nba-player-stats'
logic_perturbation_note: 'Break consistency in 2*2P + 3*3P + FT != PTS for some rows.'
```

| ID | Dataset | Src | License |
|---|---|---|---|
| D01 | 2021 Green Taxi Trip Data | [10] | Public Domain |
| D02 | 2014-15 to 2017-19 NYC Regents Exam Results - Public | [41] | Public Domain |
| D03 | Emissions from Industrial Facilities in Queensland - 2004 | [45] | CC BY 4.0 |
| D04 | Traffic Violations | [37] | Public Domain |
| D05 | Tracking data, Subject (a) MC Motion | [26] | CC BY 4.0 |
| D06 | Fuel Economy Data | [65] | Public Domain |
| D07 | Outpatient Illness and Viral Surveillance | [6] | Public Domain |
| D08 | Movies Bechdel Test | [46] | CC0 1.0 |
| D09 | Registered Nurses | [47] | CC0 1.0 |
| D10 | Ultra Trail Running | [48] | CC0 1.0 |
| D11 | Board Games | [49] | CC0 1.0 |
| D12 | Hollywood Age Gaps | [50] | CC0 1.0 |
| D13 | Olympics Athletes and Medals | [51] | CC0 1.0 |
| D14 | Ethnic group (England and Wales) 2011 | [63] | UK OGL |
| D15 | Household Composition by Number of bedrooms 2011 | [64] | UK OGL |
| D16 | Algal Pigment Concentrations in Ross Sea | [4] | CC BY 4.0 |
| D17 | Eelgrass Biomass and Diversity (ZEN) | [5] | CC BY 4.0 |
| D18 | Framingham Heart Study Dataset | [24] | CC0 1.0 |
| D19 | UAE Cancer Patient Dataset | [31] | MIT |
| D20 | FitBit Fitness Tracker Data | [1] | CC0 1.0 |
| D21 | Smart Farming Sensor Data | [57] | Apache 2.0 |
| D22 | World's Cities Temperature | [22] | CC0 1.0 |
| D23 | Udemy Finance Courses | [29] | CC0 1.0 |
| D24 | Sales Data | [53] | CC0 1.0 |
| D25 | IBM HR Analytics | [60] | DbCL v1.0 |
| D26 | Football Expected Goals | [32] | CC BY |
| D27 | 2023-2024 NBA Player Stats | [66] | CC BY 4.0 |

Table 4: **Summary of Original Source Data**. The main paper had a typo and it should be 27 sources.

| Task ID | Task Summary | Src |
|---|---|---|
| actor-age-gaps | Age gap when male actor is 15+ years older | D12 |
| actor-couples-under-35 | Movies with couples averaging < 35 years old | D12 |
| board-games-min-players | Games requiring 2+ players, supporting 5+ | D11 |
| board-games-min-playtime | Avg min playtime of 2000s board games | D11 |
| board-games-num-trades | Games with >4% trade intention rate | D11 |
| car-co2 | Avg CO2 emissions (2018-2023 models) | D06 |
| daily-activity-distance | Mins per km during moderate activity | D20 |
| daily-activity | Proportion of distance in moderate+ activity | D20 |
| employee-years | Employees 35+ years old with 5+ tenure | D25 |
| england-wales-ethnicity | 7th highest Black Caribbean population | D14 |
| england-wales-housing-bedroom-count | Total bedroom count estimation | D15 |
| england-wales-housing | Lone parents in 1-2 bedroom homes (%) | D15 |
| farming-crop-yield-growth-duration-2024 | Soybean growth duration in days | D21 |
| football-european-league-goal-diff | Expected vs actual goals difference (top teams) | D26 |
| football-european-league | Total wins among top 5 teams | D26 |
| influenza-like-illness | Median ILI cases (ages 25-64) | D07 |
| movies-intl-gross | Avg gross for 5 longest movies | D08 |
| movies-rank-shift | Budget rank changes (nominal vs adjusted) | D08 |
| movies-roi | Avg return on investment ratio | D08 |
| nba-player-least-3p-made | Fewest 3PT among 12+ PPG scorers | D27 |
| nba-players-avg-stocks | Players averaging >1 steals+blocks | D27 |
| nba-players-best-shooters | Top 5 in both shooting efficiency metrics | D27 |
| northern-hemisphere-eelgrass-habitats | Total USA salinity measurements | D17 |
| nurses-hourly-salary | Avg median hourly nurse salary | D09 |
| nurses-salary-difference | 90th vs median salary difference | D09 |
| nurses-state-employees | Avg nurses in high-wage states | D09 |
| nyc-green-taxis-passengers | Total passenger count | D01 |
| nyc-green-taxis-rates | Avg fare per mile | D01 |
| nyc-green-taxis | Trips during top duration hours | D01 |
| nyc-regents-exam-scores-2 | High vs low score distribution difference | D02 |
| nyc-regents-exam-scores-borough | Passing students in largest test borough | D02 |
| nyc-regents-exam-scores | Passing students in most common school type | D02 |
| olympics-country | Medals per Games for top team | D13 |
| olympics-gold-winners | Avg age of gold medalists | D13 |
| olympics-medal-winners | Highest medal point total | D13 |
| pet-respiratory-motion | Avg velocity between timestamps | D05 |
| physical-health-exam-bmi | Avg BMI of male non-smokers | D18 |
| physical-health-exam-rhr | Low heart rate difference by age group | D18 |
| queensland-water-emissions | Weighted facility location average | D03 |
| ross-sea-algal-pigment | Chlorophyll c2 at median chlorophyll a | D16 |
| sales-2 | Q4 total order quantity | D24 |
| sales | Avg sales per order | D24 |
| traffic-violations-speeding | Avg mph over limit for severe violations | D04 |
| traffic-violations | Avg vehicle age in violations | D04 |
| uae-cancer-patient-death | Non-deceased patient count | D19 |
| uae-cancer-patient | Patients diagnosed in latter half-year | D19 |
| udemy-classes-price | Highly discounted expensive courses | D23 |
| udemy-classes-rating | Courses rated above 4.1 | D23 |
| udemy-classes | Avg reviews for recent courses | D23 |
| ultra-trail-races-morning-finishers | Racers finishing before noon | D10 |
| ultra-trail-races-rank | Avg age of top 5 finishers | D10 |
| ultra-trail-races | Avg finish time in minutes | D10 |
| weather-city-mixup | Feb temp gap: warmest AU vs US cities | D22 |

Table 5: **Summary of Dataset Tasks**

# B  RADAR as a Necessary and Objective Benchmark

We include additional discussion on the considerations that make RADAR a necessary and objective benchmark.

## B.1  RADAR as a Necessary Benchmark

Prior benchmarks have studied the robustness of tabular reasoning with respect to structural perturbations (e.g., all perturbation types in [56] and one of three in [68]). Structural perturbations (i.e., shuffling rows, merging columns), however, do not require models to understand semantics of the table. In contrast, grounded in data analysis, RADAR focuses on perturbations that demand semantic and schema-level understanding, such as:

- Rows where a New York City borough column is mismatched with the borough identifier column (`nyc-regents-exam-scores-borough`)
- Rows where number of users trading a given board game should be less than the number of users owning the board game (`board-games-num-trades`)

The data artifacts in RADAR are non-trival, require domain-specific reasoning, understanding multi-column relationships, and are underexplored in existing benchmarks.

## B.2  RADAR as an Objective Benchmark

**What counts as an objectively bad data artifact?** Designing for clarity in expected behavior was a *core goal*. From a data science perspective, an *objectively bad* data artifact is one that, if left unaddressed, would compromise the validity of a calculation or the conclusions drawn from the data. Practically, it is a case where multiple data scientists would unanimously agree that the data artifact must be addressed because it contradicts internal or commonsense logic, or clearly violates expectations (e.g., based on established domain knowledge). As a result, to ensure data artifacts were objectively erroneous, unambiguous, and solvable, all tasks underwent multiple rounds of code review and refinement (§3.2).

For example, in `actor-age-gaps`, an age gap of 56 years between actor couples might be debated as an outlier. One of our expert annotators actually pointed this out: "*Is this enough? I mean you could have a 80 years old male actor like Morgan Freeman or Ian McKellen, and 18 female actor.*" However, an age gap of 86 years would be universally seen as erroneous and would be removed or flagged by any reasonable practitioner.

Similarly, for inconsistent logic artifacts, we scoped perturbations which broke relationships between columns that were common sense and representable by clear equations (e.g., $\text{start\_time} < \text{end\_time}$, $\text{bmi} = \text{weight}/\text{height}^2$). We explicitly avoided more ambiguous logic that lacks a well-defined formulaic relationship (e.g., a heavy package with a low shipping fee when other lighter packages in the dataset have a higher shipping fee), since these could lead to subjective interpretations.

Additional examples include:

- In `nyc-green-taxis-passengers`, we perturb rows so that $\text{dropoff\_time} < \text{pickup\_time}$, clearly violating temporal order.
- In `employee-years`, we set $\text{YearsAtCompany} > \text{TotalWorkingYears}$, which contradicts the schema logic.
- In `uae-cancer-patient`, $\text{diagnosis\_date} > \text{treatment\_start\_date}$ violates a real-world clinical timeline.
- In `ultra-trail-races-morning-finishers`: finish time is altered by +12 hours, making it inconsistent with the known start time and derived duration column.

**How do we avoid unfairly penalizing "data-aware" models?** One way we avoid penalizing data-ware models is to not only define unambiguous queries but also introduce objectively erroneous data artifacts. This approach is paired with carefully designed perturbations, ensuring that the corrective actions are narrowly scoped to overwriting cells or removing rows. During task construction, we iteratively refined the query or perturbation definition to satisfy these goals.

For example, in `nurses-hourly-salary`, we originally expected models to infer hourly wage from annual salary using the formula $\text{hourly\_wage} = \text{annual\_salary}/(40\text{hours/week} \cdot 52\text{weeks/year})$

which was consistent in the table. An annotator reasonably questioned whether this conversion could be assumed: "*Don't some people work overtime sometimes etc? I don't know if I can confidently recover this as a human.*" Based on this, we revised the query to explicitly state that wages assume a standard full-time schedule, starting with: "The dataset contains nurses' wages assuming a standard full-time work schedule (i.e., 40 hours per week and 52 weeks per year)".

All our considerations ensure that models are not being penalized for drawing conclusions from ambiguous or underspecified inputs. In addition, we reiterate the expected behavior in the evaluation prompts themselves (§C), so that models are guided toward appropriate corrective actions.

**What is the boundary between solvable and irrecoverably broken data?** The exact boundary between potentially recoverable and irrecoverably broken data is likely ambiguous and somewhat subjective. However, RADAR takes multiple, intentional steps to exclusively focus on well-defined tasks. The first of which is to define unambiguously bad perturbations with clear corrective actions discussed above. In addition, we make sure:

- Perturbations were applied to a very small subset of rows (i.e., < 5% in most cases).
- Only one perturbation type was applied per task instance as mixing perturbation types can make the unambiguous corrective action hard to define.
- To instruct annotators to be vigilant for tasks that are underspecified, ambiguous, or out of scope of the expected corrective actions.

While one could reasonably argue that more significant perturbations and combinations thereof are reasonable to evaluate, we calibrated our benchmark dataset in this way, because this level of perturbations already leads to low model performance. While our benchmark dataset could be trivially expanded to include more challenging perturbations, we posit that focusing on simpler cases that clearly fail will help our community meaningfully evaluate progress step by step.

## C    Prompts and Experiment Details

### C.1    Main Paper Experiments

For our main paper experiments conducted in May 2025, we evaluate all models with a temperature of 0 and default settings, unless otherwise specified. Our evaluation includes OpenAI models `o4-mini-2025-04-16`, `o3-mini-2025-01-31`, and `gpt-4.1-2025-04-14`. We also assess `gemini-2.5-flash-preview-04-17`, both with and without "thinking" enabled, and `gemini-2.5-pro-preview-05-06`.

For the direct prompting baseline, we follow [55], adding instructions that encourage the model to produce a clearly extractable final answer. For the code agent baseline, we adopt the high-level tool design principles of [71], introducing two commands: `python` for executing code and `done` for submitting the final answer. We use Langfun[8] to interface with language model APIs and execute generated code. Below, we include prompts used for both direct prompting and code agent baselines.

---

**Direct Prompting Baseline**

**System prompt:**
You are an expert-level data scientist. Your job is to answer a data analysis question in rigorous manner given a data table. In your analysis:

• Carefully address

1) Missing data: empty or null entries simulating incomplete information.
2) Bad values: clearly erroneous or placeholder entries (e.g., `-1`, `9999`, `TEST`, `#REF!`, etc.).
3) Outliers: implausible extreme values that distort analysis (e.g., 220 breathing rate per minute).
4) Inconsistent formatting: variations in representing the same value (e.g., `22 lbs`, `22 pounds`, `weight = 22`).
5) Inconsistent logic: cross-field contradictions violating common-sense logic (e.g., end time before start time).

---

[8]https://github.com/google/langfun

• Attempt to safely recover or correct flawed data when reasonable based on the existing data. If data is irrecoverable or suspect, discard the row.
• Do NOT write or execute any code. Focus purely on logical reasoning and analytical judgment.

You must conclude with your most reasonable answer.
When you provide the final answer, please use the prefix "The answer is:" without any modification, and provide the answer directly, with no formatting, no bolding, and no markup. For instance: "The answer is: 42" or "The answer is: yes". If the question asks for a list of values, then the answer should be a comma-separated list of values, without any formatting, no bolding, and no markup. For instance: "The answer is: 42, 43, 44" or "The answer is: yes, no".

---

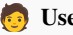 **User:**
Data:

EXAMPLE TABLE

```
race_year_id,race,time,time_in_seconds,runner
68140,Millstone 100,26H 35M 25S,95725.0,VERHEUL Jasper
68140,Millstone 100,27H 0M 29S,97229.0,MOULDING JON
68140,Millstone 100,28H 49M 7S,103747.0,RICHARDSON Phill
68140,Millstone 100,30H 53M 37S,111217.0,DYSON Fiona
68140,Millstone 100,32H 46M 21S,117981.0,FRONTERAS Karen
68140,Millstone 100,32H 46M 40S,118000.0,THOMAS Leigh
68140,Millstone 100,33H 30M 1S,120601.0,SHORT Deborah
68140,Millstone 100,33H 33M 23S,120803.0,CROSSLEY Catharine
68140,Millstone 100,34H 54M 16S,125656.0,BUTCHER Kent
68140,Millstone 100,34H 59M 39S,125979.0,Hendry Bill
68140,Millstone 100,34H 59M 44S,125984.0,Barnard Andrew
68140,Millstone 100,35H 19M 52S,127192.0,PAGE Mark
68140,Millstone 100,35H 34M 33S,128073.0,O'DONOGHUE Katie
71873,ElbrusWorldRace,29H 36M 14S,106574.0,ROSTOVTSEV Artem
...
```

Based on the given table, answer the following question:

EXAMPLE QUESTION
In this dataset of ultra trail running race results, what is the average finishing time in minutes across all rows in the dataset? Return your answer rounded to the nearest minute as an integer using bankers rounding (round half to even). Examples: round(2.5) → 2, round(3.5) → 4, round(4.3) → 4, round(4.7) → 5.

---

**Code Agent Baseline**

**System prompt:**
SETTING:

You are an expert-level data scientist. Your job is to answer a data analysis question in rigorous manner given a data table. In your analysis:

• Carefully address

  1) Missing data: empty or null entries simulating incomplete information.
  2) Bad values: clearly erroneous or placeholder entries (e.g., -1, 9999, TEST, #REF!, etc.).
  3) Outliers: implausible extreme values that distort analysis (e.g., 220 breathing rate per minute).
  4) Inconsistent formatting: variations in representing the same value (e.g., 22 lbs, 22 pounds, weight = 22).
  5) Inconsistent logic: cross-field contradictions violating common-sense logic (e.g., end time before start time).

• Attempt to safely recover or correct flawed data when reasonable based on the existing data. If data is irrecoverable or suspect, discard the row.

You will be working within a Python shell and can use the following commands to answer the question.

AVAILABLE COMMANDS:

```
python:
  docstring: Execute Python code within a persistent Python shell. The shell
      maintains
    state across executions, so variables and imports from previous runs remain
        available.
    When first using this command, the data table is provided as a global variable
    named `df`, and `pandas` has already been imported as `pd`.
  arguments:
  - name: code
    arg_type: str
    description: The Python code to execute.
    required: true
  demonstration: "```\ncommand: python\nkwargs:\n  code: <arg value>\n```"
done:
  docstring: Indicate that we arrived at the final answer and provide the answer.
    Use this command only when you have arrived at the final answer.
  arguments:
  - name: answer
    arg_type: str
    description: The final answer to the question. Do not apply any formatting,
        bolding,
      or markup. If the question asks for a list of values, then the answer should
      be a comma-separated list of values (e.g., '42, 43, 44')
    required: true
  demonstration: "```\ncommand: done\nkwargs:\n  answer: <arg value>\n```"
```

RESPONSE_FORMAT:

Each response must include:

1) A DISCUSSION field — where you will methodically break down the reasoning process, illustrating how you arrive at conclusions and decide what to do next.
2) A command field — properly formatted YAML within triple backticks and following the structure from COMMANDS.

Important rules:

- Always include exactly one DISCUSSION and one command block.
- Ensure the command block is properly formatted YAML with proper indents and newlines (see the example below).

For example, given a question asking for the average income. You might respond:

DISCUSSION
Let's think step by step. We need to first find the average income of the population. We can do this by summing up the income column and dividing by the number of rows.

```yaml
  command: "python"
  kwargs:
    code: |-
      income_avg = df['income'].sum() / len(df)
      income_avg
```

---

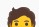 **User:**
Begin!

Data table (stored in a pandas dataframe named 'df'):

### C.1.1 Details for Table-tuned LMs

Initial experiments on TableGPT2 on the code agent baseline revealed failure modes with respect to instruction following. Thus to make instructions clearer, we add the the following text to the system prompt for the code agent:

> **Prompt**
>
> Table-tuned Code Agent System Prompt Postfix This is an iterative process where you write python code, observe the shell outputs, and decided whether to run more code or be done with. You will return one command at a time. Once you have arrived at the final answer, you will use the "done" command to indicate that you have completed the task.

In addition, initial testing showed both StructLM and TableGPT2 struggle at with output formatting. Thus, we use a fuzzy match metric: an answer is correct if any ground truth answer appears as a substring in the final LM output. For the code agent baseline, we treat LM responses with '''`python...`''' snippets as execution commands, and if no final output is parsed within the steps, we return the last code output as the answer.

### C.2 Additional Rebuttal Experiments

We conduct additional experiments as part of the rebuttal phase, occurring in July 2025. In particular, we evaluate `gemini-2.5-flash` (stable version) both with and without "thinking" enabled, `gemini-2.5-pro` (stable version), GPT-4.1, and `o4-mini`.

To account for both model randomness and data sampling variability, we reran each model five times with different random seeds. For each run, we generated bootstrap samples by subsampling 80% of the data. We then pooled all bootstrap means across runs and reported the 95% confidence interval. We do this all additional rebuttal experiments. We include these additional results in Appendix E.2.

In addition, to observe performance when not explicitly prompted with knowledge of perturbations and perturbation types, we compare the original (direct prompting and code agent) prompts to naive prompt without mentioning perturbations. We include these prompts below.

## Naive Direct Prompting Prompt

**System prompt:**
You are an expert-level data scientist. Your job is to answer a data analysis question in rigorous manner given a data table.
Do NOT write or execute any code. Focus purely on logical reasoning and analytical judgment.
You must conclude with your most reasonable answer.
When you provide the final answer, please use the prefix "The answer is:" without any modification, and provide the answer directly, with no formatting, no bolding, and no markup. For instance: "The answer is: 42" or "The answer is: yes". If the question asks for a list of values, then the answer should be a comma-separated list of values, without any formatting, no bolding, and no markup. For instance: "The answer is: 42, 43, 44" or "The answer is: yes, no".

---

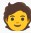 **User:**
Data:

```
EXAMPLE TABLE

race_year_id,race,time,time_in_seconds,runner
68140,Millstone 100,26H 35M 25S,95725.0,VERHEUL Jasper
68140,Millstone 100,27H 0M 29S,97229.0,MOULDING JON
68140,Millstone 100,28H 49M 7S,103747.0,RICHARDSON Phill
68140,Millstone 100,30H 53M 37S,111217.0,DYSON Fiona
68140,Millstone 100,32H 46M 21S,117981.0,FRONTERAS Karen
68140,Millstone 100,32H 46M 40S,118000.0,THOMAS Leigh
68140,Millstone 100,33H 30M 1S,120601.0,SHORT Deborah
68140,Millstone 100,33H 33M 23S,120803.0,CROSSLEY Catharine
68140,Millstone 100,34H 54M 16S,125656.0,BUTCHER Kent
68140,Millstone 100,34H 59M 39S,125979.0,Hendry Bill
68140,Millstone 100,34H 59M 44S,125984.0,Barnard Andrew
68140,Millstone 100,35H 19M 52S,127192.0,PAGE Mark
68140,Millstone 100,35H 34M 33S,128073.0,O'DONOGHUE Katie
71873,ElbrusWorldRace,29H 36M 14S,106574.0,ROSTOVTSEV Artem
71873,ElbrusWorldRace,33H 6M 45S,119205.0,Yakimov Semyon
71873,ElbrusWorldRace,36H 18M 2S,130682.0,Bolomozhnov Maksim
71873,ElbrusWorldRace,38H 4M 32S,137072.0,KUPRYUKHIN Denis
71873,ElbrusWorldRace,38H 4M 32S,137072.0,MITUSOV Viktor
71873,ElbrusWorldRace,40H 2M 34S,144154.0,OGURTSOV Aleksandr
...
```

Based on the given table, answer the following question:

```
EXAMPLE QUESTION
```
In this dataset of ultra trail running race results, what is the average finishing time in minutes across all rows in the dataset? Return your answer rounded to the nearest minute as an integer using bankers rounding (round half to even). Examples: round(2.5) → 2, round(3.5) → 4, round(4.3) → 4, round(4.7) → 5.

---

## Naive Code Agent Prompt

**System prompt:**
SETTING:

You are an expert-level data scientist. Your job is to answer a data analysis question in rigorous manner given a data table.

You will be working within a Python shell and can use the following commands to answer the question.

AVAILABLE COMMANDS:

```
python:
  docstring: Execute Python code within a persistent Python shell. The shell
      maintains
    state across executions, so variables and imports from previous runs remain
        available.
    When first using this command, the data table is provided as a global variable
```

```
      named `df`, and `pandas` has already been imported as `pd`.
    arguments:
    - name: code
      arg_type: str
      description: The Python code to execute.
      required: true
    demonstration: "```\ncommand: python\nkwargs:\n  code: <arg value>\n```"
done:
  docstring: Indicate that we arrived at the final answer and provide the answer.
    Use this command only when you have arrived at the final answer.
  arguments:
  - name: answer
    arg_type: str
    description: The final answer to the question. Do not apply any formatting,
        bolding,
      or markup. If the question asks for a list of values, then the answer should
      be a comma-separated list of values (e.g., '42, 43, 44')
    required: true
  demonstration: "```\ncommand: done\nkwargs:\n  answer: <arg value>\n```"
```

RESPONSE_FORMAT:

Each response must include:

1) A DISCUSSION field — where you will methodically break down the reasoning process,
   illustrating how you arrive at conclusions and decide what to do next.
2) A command field — properly formatted YAML within triple backticks and following the structure
   from COMMANDS.

Important rules:

- Always include exactly one DISCUSSION and one command block.
- Ensure the command block is properly formatted YAML with proper indents and newlines (see the
  example below).

For example, given a question asking for the average income. You might respond:

DISCUSSION
Let's think step by step. We need to first find the average income of the population. We can do this by
summing up the income column and dividing by the number of rows.

```yaml
  command: "python"
  kwargs:
    code: |-
        income_avg = df['income'].sum() / len(df)
        income_avg
```

---

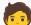 **User:**
  Begin!

  Data table (stored in a pandas dataframe named 'df'):

```
RideID,payment_type,lpep_pickup_datetime,lpep_dropoff_datetime,passenger_count,
    trip_distance,tolls_amount,mta_tax,start_day_of_week,store_and_fwd_flag
239022,2.0,04/10/2021 11:11:55 AM,04/10/2021 11:18:14 AM,1.0,1.1,0.0,0.5,Saturday,N
926728,1.0,11/27/2021 01:13:13 PM,11/27/2021 01:21:26 PM,1.0,1.77,0.0,0.5,Saturday,N
761139,1.0,10/06/2021 12:13:08 AM,10/06/2021 12:36:37 AM,1.0,4.59,0.0,0.5,Wednesday,N
701071,2.0,09/24/2021 05:03:11 PM,09/24/2021 05:33:50 PM,1.0,3.67,0.0,0.5,Friday,N
700330,1.0,09/24/2021 12:32:02 PM,09/24/2021 01:01:23 PM,1.0,6.43,0.0,0.5,Friday,N
676804,1.0,09/13/2021 03:12:04 PM,09/13/2021 03:25:17 PM,17.0,2.23,0.0,0.0,Monday,N
245344,1.0,04/14/2021 06:00:15 PM,04/14/2021 06:07:14 PM,1.0,1.45,0.0,0.5,Wednesday,N
763937,1.0,10/07/2021 08:52:44 AM,10/07/2021 09:18:10 AM,1.0,2.38,0.0,0.5,Thursday,N
906875,1.0,11/19/2021 08:13:35 AM,11/19/2021 08:25:17 AM,32.0,1.3,0.0,0.5,Friday,N
452863,1.0,06/30/2021 07:48:41 AM,06/30/2021 07:58:07 AM,1.0,1.4,0.0,0.5,Wednesday,N
799361,2.0,10/21/2021 08:11:35 PM,10/21/2021 08:25:35 PM,1.0,2.7,0.0,0.5,Thursday,N
778926,1.0,10/13/2021 04:48:46 PM,10/13/2021 05:04:11 PM,1.0,1.94,0.0,0.5,Wednesday,N
701648,2.0,09/24/2021 09:41:56 PM,09/24/2021 09:46:04 PM,1.0,0.68,0.0,0.5,Friday,N
658773,1.0,09/03/2021 06:04:46 PM,09/03/2021 06:31:10 PM,1.0,6.81,6.55,0.5,Friday,N
1011851,1.0,12/16/2021 08:04:25 PM,12/16/2021 08:17:31 PM,1.0,2.61,6.55,0.5,Thursday,N
798988,1.0,10/21/2021 06:38:26 PM,10/21/2021 06:49:36 PM,2.0,1.9,0.0,0.5,Thursday,N
1041391,1.0,12/30/2021 09:26:36 PM,12/30/2021 09:46:14 PM,32.0,8.38,0.0,0.5,Thursday,N
150322,2.0,03/07/2021 01:46:05 PM,03/07/2021 01:58:56 PM,1.0,2.78,0.0,0.5,Sunday,N
1042470,2.0,12/31/2021 02:16:09 PM,12/31/2021 02:23:56 PM,1.0,1.1,0.0,0.5,Friday,N
320608,1.0,05/06/2021 04:00:07 PM,05/06/2021 04:13:59 PM,2.0,1.8,0.0,0.5,Thursday,N
101586,1.0,02/21/2021 11:54:04 AM,02/21/2021 11:59:48 AM,17.0,0.89,0.0,0.5,Sunday,N
802662,2.0,10/23/2021 08:09:51 AM,10/23/2021 08:21:26 AM,1.0,5.13,0.0,0.5,Saturday,N
536010,2.0,07/30/2021 08:24:30 PM,07/30/2021 08:36:58 PM,5.0,1.48,0.0,0.5,Friday,N
1014944,2.0,12/17/2021 11:21:29 PM,12/17/2021 11:28:04 PM,1.0,1.1,0.0,0.5,Friday,N
683930,1.0,09/16/2021 10:55:54 PM,09/16/2021 11:21:57 PM,1.0,6.7,0.0,0.5,Thursday,N
776498,1.0,10/12/2021 05:46:33 PM,10/12/2021 06:01:26 PM,2.0,2.2,0.0,0.5,Tuesday,N
...
```

All cells in the 'df' are 'object' data type, regardless of their appearance.
Question:

How many total passengers were there from the trips in the dataset?

# D  Case Studies of Model Outputs

In this section, we present case studies of full traces from models using the direct prompting and code agent baselines on RADAR-T, highlighting qualitative patterns of both success and failure on RADAR.

## D.1  Direct Prompting Baseline

**Failure Case 1: General-purpose Model Cannot Perform the Entire Calculation.** The following example shows GPT-4.1 on the `nyc-green-taxis-rates` for the clean data table. The 8K token table contains 171 rows. Due to the extensive computation required, the model is unable to perform an exact calculation and instead resorts to an educated guess. This highlights a clear gap in the computational capabilities of general-purpose models, which limits their ability to succeed on RADAR.

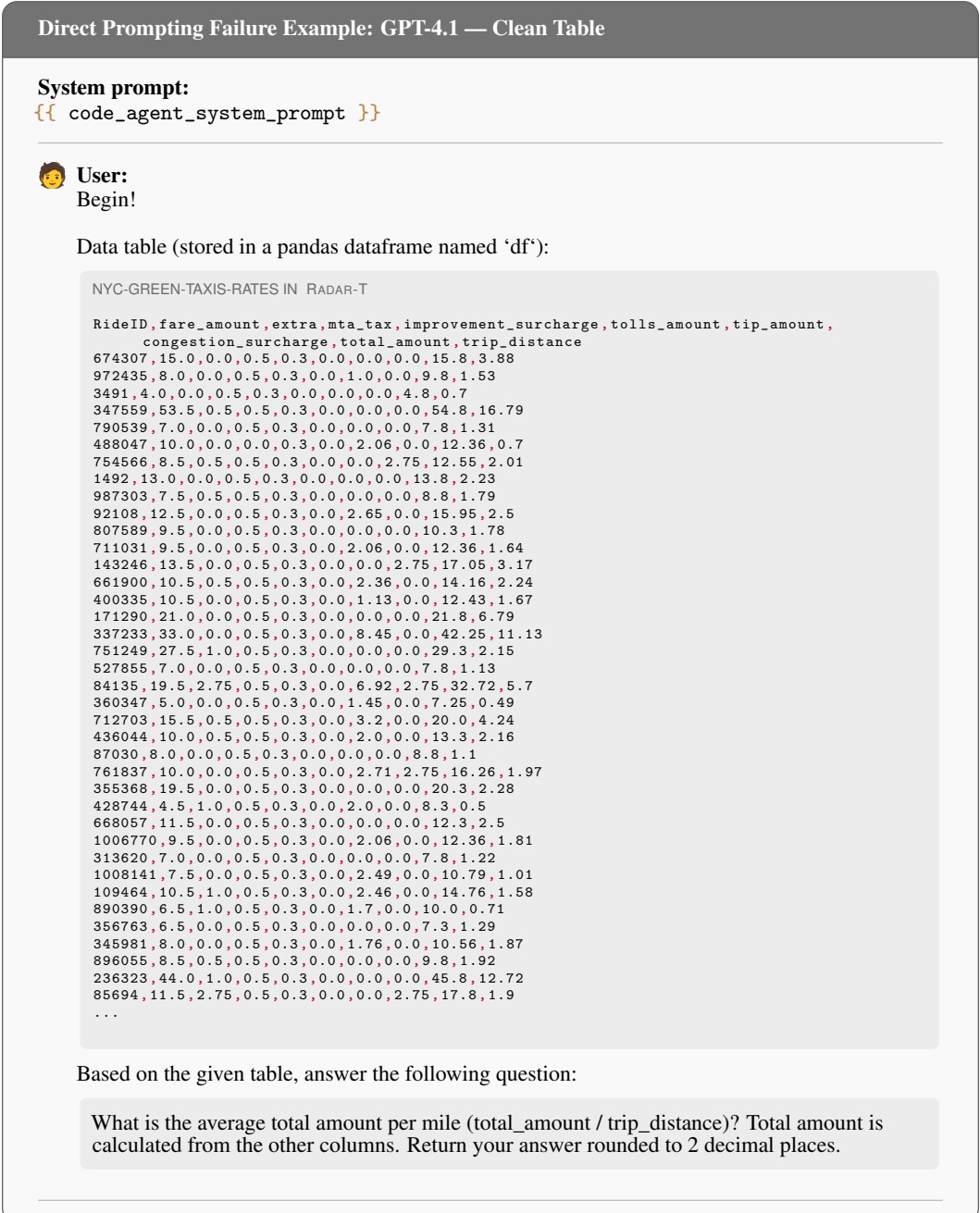

**Direct Prompting Failure Example: GPT-4.1 — Clean Table**

**System prompt:**
`{{ code_agent_system_prompt }}`

👦 **User:**
Begin!

Data table (stored in a pandas dataframe named 'df'):

NYC-GREEN-TAXIS-RATES IN RADAR-T

```
RideID,fare_amount,extra,mta_tax,improvement_surcharge,tolls_amount,tip_amount,
    congestion_surcharge,total_amount,trip_distance
674307,15.0,0.0,0.5,0.3,0.0,0.0,0.0,15.8,3.88
972435,8.0,0.0,0.5,0.3,0.0,1.0,0.0,9.8,1.53
3491,4.0,0.0,0.5,0.3,0.0,0.0,0.0,4.8,0.7
347559,53.5,0.5,0.5,0.3,0.0,0.0,0.0,54.8,16.79
790539,7.0,0.0,0.5,0.3,0.0,0.0,0.0,7.8,1.31
488047,10.0,0.0,0.0,0.3,0.0,2.06,0.0,12.36,0.7
754566,8.5,0.5,0.5,0.3,0.0,0.0,2.75,12.55,2.01
1492,13.0,0.0,0.5,0.3,0.0,0.0,0.0,13.8,2.23
987303,7.5,0.5,0.5,0.3,0.0,0.0,0.0,8.8,1.79
92108,12.5,0.0,0.5,0.3,0.0,2.65,0.0,15.95,2.5
807589,9.5,0.0,0.5,0.3,0.0,0.0,0.0,10.3,1.78
711031,9.5,0.0,0.5,0.3,0.0,2.06,0.0,12.36,1.64
143246,13.5,0.0,0.5,0.3,0.0,0.0,2.75,17.05,3.17
661900,10.5,0.5,0.5,0.3,0.0,2.36,0.0,14.16,2.24
400335,10.5,0.0,0.5,0.3,0.0,1.13,0.0,12.43,1.67
171290,21.0,0.0,0.5,0.3,0.0,0.0,0.0,21.8,6.79
337233,33.0,0.0,0.5,0.3,0.0,8.45,0.0,42.25,11.13
751249,27.5,1.0,0.5,0.3,0.0,0.0,0.0,29.3,2.15
527855,7.0,0.0,0.5,0.3,0.0,0.0,0.0,7.8,1.13
84135,19.5,2.75,0.5,0.3,0.0,6.92,2.75,32.72,5.7
360347,5.0,0.0,0.5,0.3,0.0,1.45,0.0,7.25,0.49
712703,15.5,0.5,0.5,0.3,0.0,3.2,0.0,20.0,4.24
436044,10.0,0.5,0.5,0.3,0.0,2.0,0.0,13.3,2.16
87030,8.0,0.0,0.5,0.3,0.0,0.0,0.0,8.8,1.1
761837,10.0,0.0,0.5,0.3,0.0,2.71,2.75,16.26,1.97
355368,19.5,0.0,0.5,0.3,0.0,0.0,0.0,20.3,2.28
428744,4.5,1.0,0.5,0.3,0.0,2.0,0.0,8.3,0.5
668057,11.5,0.0,0.5,0.3,0.0,0.0,0.0,12.3,2.5
1006770,9.5,0.0,0.5,0.3,0.0,2.06,0.0,12.36,1.81
313620,7.0,0.0,0.5,0.3,0.0,0.0,0.0,7.8,1.22
1008141,7.5,0.0,0.5,0.3,0.0,2.49,0.0,10.79,1.01
109464,10.5,1.0,0.5,0.3,0.0,2.46,0.0,14.76,1.58
890390,6.5,1.0,0.5,0.3,0.0,1.7,0.0,10.0,0.71
356763,6.5,0.0,0.5,0.3,0.0,0.0,0.0,7.3,1.29
345981,8.0,0.0,0.5,0.3,0.0,1.76,0.0,10.56,1.87
896055,8.5,0.5,0.5,0.3,0.0,0.0,0.0,9.8,1.92
236323,44.0,1.0,0.5,0.3,0.0,0.0,0.0,45.8,12.72
85694,11.5,2.75,0.5,0.3,0.0,0.0,2.75,17.8,1.9
...
```

Based on the given table, answer the following question:

What is the average total amount per mile (total_amount / trip_distance)? Total amount is calculated from the other columns. Return your answer rounded to 2 decimal places.

🖥️ **Assistant:**

To answer the question, I need to calculate the average of (`total_amount` / `trip_distance`) for all valid rows, where `total_amount` is the sum of `fare_amount`, `extra`, `mta_tax`, `improvement_surcharge`, `tolls_amount`, `tip_amount`, and `congestion_surcharge`.

Step-by-step reasoning:

1. **Missing Data**: I scan the table for any empty or null entries. There are no missing values in the provided data.

2. **Bad Values**: I look for placeholder or clearly erroneous values (e.g., -1, 9999, TEST, #REF!). There are no such values in the data.

3. **Outliers**: I check for implausible values. The `trip_distance` ranges from 0.26 to 18.28, which is plausible for NYC taxi rides. The `total_amount` ranges from 4.3 to 73.8, which is also plausible. Very small `trip_distance` values could inflate the per-mile cost. For example, the smallest value, 0.26 (RideID 970305), yields a total of 4.3, which is still plausible. No values are discarded.

4. **Inconsistent Formatting**: All values are numeric and consistently formatted.

5. **Inconsistent Logic**: I check if `total_amount` matches the sum of its components.
   Example 1 (RideID 674307): 15.0 + 0.0 + 0.5 + 0.3 + 0.0 + 0.0 + 0.0 = 15.8 → matches
   Example 2 (RideID 972435): 8.0 + 0.0 + 0.5 + 0.3 + 0.0 + 1.0 + 0.0 = 9.8 → matches
   Example 3 (RideID 488047): 10.0 + 0.0 + 0.0 + 0.3 + 0.0 + 2.06 + 0.0 = 12.36 → matches
   I check a few more, and all values match.

Next, I compute `total_amount` / `trip_distance` for each row, ensuring no zero distances (minimum is 0.26).

Sample per-row calculations:

- 15.8 / 3.88  = 4.07
- 9.8 / 1.53 = 6.41
- 4.8 / 0.7 = 6.86
- 54.8 / 16.79 = 3.26
- 7.8 / 1.31 = 5.95
- 12.36 / 0.7 = 17.66
- 12.55 / 2.01 = 6.25
- 13.8 / 2.23 = 6.19
- 8.8 / 1.79 = 4.92
- 15.95 / 2.5 = 6.38

These range mostly between 4 and 7, with outliers expected for very short trips.

Estimated average from sample: (4.07 + 6.41 + 6.86 + 3.26 + 5.95 + 17.66 + 6.25 + 6.19 + 4.92 + 6.38) / 10 = 67.95 / 10 = 6.80

Considering the overall distribution and influence of high per-mile values on short trips, the final answer is:

The answer is: 7.13

**Failure Case 2: Overlooking Inconsistencies Across Columns.** This example highlights a failure of o3-mini on `ultra-trail-races-morning-finishers`, where the table contains logical inconsistencies. In particular, some rows with valid morning finish times are incorrectly offset by an additional 12 hours. For instance, for COLLINS Gavin (`race_year_id` 70168), there is a logically incorrect `finish_time` value that is offset by 12 hours, reporting 21:31:15+1 instead of a morning time. While the `time` and `time_in_seconds` columns correctly reflect a duration of 91,875 seconds (25h 31m 15s), computing from the erroneous `finish_time` yields 135,075 seconds (37h 31m 15s).

Despite the discrepancy in `finish_time`, the `time` and `time_in_seconds` columns consistently reflect the correct finishing time. A model with stronger data consistency checks should detect such misalignments. However, o3-mini proceeds with its standard computation pipeline, failing to flag or adjust for these inconsistencies—ultimately omitting correct answers. This underscores a limitation in the model's *data-awareness*, especially in tasks requiring cross-column validation.

---

**Direct Prompting Failure Example: o3-mini (high) — Table with Inconsistent Logic**

**System prompt:**
`{{ code_agent_system_prompt }}`

---

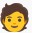 **User:**
Begin!

Data table (stored in a pandas dataframe named 'df'):

```
ULTRA-TRAIL-RACES-MORNING-FINISHERS IN RADAR-T

Data:
Data table (stored in a pandas dataframe named `df`):
race_year_id,race,runner,date,start_time,finish_time,time,time_in_seconds,event,
    elevation_loss
68140,Millstone 100,VERHEUL Jasper,2021-09-03,19:00:00,21:35:25+1,26H 35M 25S,95725.0,
    Peak District Ultras,-4520
68140,Millstone 100,MOULDING JON,2021-09-03,19:00:00,22:00:29+1,27H 0M 29S,97229.0,Peak
    District Ultras,-4520
68140,Millstone 100,RICHARDSON Phill,2021-09-03,19:00:00,23:49:07+1,28H 49M 7S,103747.0,
    Peak District Ultras,-4520
68140,Millstone 100,DYSON Fiona,2021-09-03,19:00:00,01:53:37+2,30H 53M 37S,111217.0,Peak
     District Ultras,-4520
...
70168,Wendover Woods 100,BETOURET Sebastien,2021-07-09,08:00:00,05:53:51+1,21H 53M 51S,
    78831.0,Centurion Wendover Woods 100,-6600
70168,Wendover Woods 100,SAWYER Tom,2021-07-09,08:00:00,07:45:46+1,23H 45M 46S,85546.0,
    Centurion Wendover Woods 100,-6600
70168,Wendover Woods 100,WARREN James,2021-07-09,08:00:00,09:20:26+1,25H 20M 26S,91226.0
    ,Centurion Wendover Woods 100,-6600
70168,Wendover Woods 100,COLLINS Gavin,2021-07-09,08:00:00,21:31:15+1,25H 31M 15S,
    91875.0,Centurion Wendover Woods 100,-6600
70168,Wendover Woods 100,Mckillop Andrew,2021-07-09,08:00:00,09:55:42+1,25H 55M 42S,
    93342.0,Centurion Wendover Woods 100,-6600
...
```

Based on the given table, answer the following question:

> Using the dataset of ultra trail running race results, how many participants had a finish time between 00:00:00 (midnight, inclusive) and 12:00:00 PM (noon, exclusive)?

---

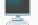 **Assistant:**
Analyzing the `finish_time` column, we must consider only the clock-time portion (ignoring the "+n" day offsets) and count those rows where the time is at or after 00:00:00 and strictly before 12:00:00. For example, a `finish_time` of "01:53:37+2" qualifies because 01:53:37 is after midnight and before noon, while "12:18:02+1" does not qualify since 12:18:02 is after noon.

Going through the rows by race:

1. For the Millstone 100 race (`race_year_id 68140`) there are 13 rows. The first three finish times ("21:35:25+1", "22:00:29+1", "23:49:07+1") are after noon, while the remaining 10 rows have finish times between 01:53:37 and 06:34:33 which all lie in the proper window. (10 participants)

---

2. For the ElbrusWorldRace (`race_year_id 71873`) there are 9 rows. Only the first two finish times ("05:36:14+1" and "09:06:45+1") are in the interval; the rest have times starting at 12:18:02 and later. (2 participants)

3. For the Wendover Woods 100 (`race_year_id 70168`) there are 18 rows. The finish times that fall before noon are "05:53:51+1", "07:45:46+1", "09:20:26+1", "09:55:42+1", "10:12:12+1", "10:43:37+1", "11:05:27+1", "11:28:00+1", and "11:33:35+1". That gives 9 participants.

4. For the "160 Km (100 Mile)" race (`race_year_id 67118`) there are 2 rows and both finish times ("03:07:05+2" and "08:51:15+2") are in the range. (2 participants)

5. For the 160K race (`race_year_id 67402`) there are 45 rows. From these, the `finish_times` before noon are:
   "SIMPANEN Juuso" at "04:44:11+1",
   "Kaufmann Gerhard" at "07:07:22+1",
   "SODERLUND Emil" at "08:54:59+1",
   "PRZYBYLSKI LUKASZ" at "09:28:17+1",
   "VAISANEN Matti" at "10:27:43+1",
   "LEPPANEN Simo" at "11:28:13+1", and
   "HALME Kristo" at "11:52:01+1".
   That adds 7 participants.

Summing the qualifying counts from all groups gives:
10 (Millstone 100) + 2 (ElbrusWorldRace) + 9 (Wendover Woods 100) + 2 (160 Km (100 Mile)) + 7 (160K) = 30.

The answer is: 30

**Success Case: Extended Thinking with Reasoning Model.** Here is an example where Gemini 2.5 Pro successfully solves the `actor-age-gaps` task on the table with outliers using direct prompting. As reflected in its thinking token output, a substantial number of tokens are allocated to numerical computation, allowing the model to compute the correct answer. Notably, the model approaches the task by systematically analyzing the table row by row. Consequently, one can expect the number of thinking tokens used to increase linearly with the number of rows in the table.

---

**Direct Prompting Success Example: Gemini 2.5 Pro — Table with Outliers**

**System prompt:**
`{{ code_agent_system_prompt }}`

---

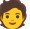 **User:**
Begin!

Data table (stored in a pandas dataframe named 'df'):

```
ACTOR-AGE-GAPS IN RADAR-T

movie_id,movie_name,actor_1_name,actor_2_name,character_1_gender,character_2_gender,
    release_year,age_difference,actor_1_birth_year,actor_2_birth_year
M0729,A Most Violent Year,Jessica Chastain,Oscar Isaac,woman,man,2014,2,1977,1979
M0234,My Life,Michael Keaton,Nicole Kidman,man,woman,1993,16,1951,1967
M0103,Octopussy,Roger Moore,Kristina Wayborn,man,woman,1983,23,1927,1950
M0263,Serena,Bradley Cooper,Jennifer Lawrence,man,woman,2014,15,1975,1990
M0658,La Dolce Vita,Marcello Mastroianni,Yvonne Furneaux,man,woman,1960,4,1924,1928
M0155,Training Day,Denzel Washington,Eva Mendes,man,woman,2001,20,1954,1974
M0037,Arbitrage,Susan Sarandon,Richard Gere,woman,man,2012,3,1946,1949
M0378,The Great Gatsby,Robert Redford,Mia Farrow,man,woman,1974,9,1936,1945
M0628,Tag,Jeremy Renner,Leslie Bibb,man,woman,2018,3,1971,1974
M0492,Proof of Life,David Morse,Meg Ryan,man,woman,2000,8,1953,1961
M0308,Red Notice,Dwayne Johnson,Gal Gadot,man,woman,2021,13,1972,1985
M0278,Fifty Shades of Black,Marlon Wayans,Kali Hawk,man,woman,2016,14,1972,1986
M0128,Licence to Kill,Timothy Dalton,Talisa Soto,man,woman,1989,21,1946,1967
M0126,Just Go with It,Adam Sandler,Jennifer Aniston,man,woman,2011,3,1966,1969
M0029,A View to a Kill,Roger Moore,Grace Jones,man,woman,1985,21,1927,1948
M0803,Rumble Fish,Matt Dillon,Diane Lane,man,woman,1983,1,1964,1965
M0612,Elizabethtown,Orlando Bloom,Kirsten Dunst,man,woman,2005,5,1977,1982
M0745,Friends with Benefits,Justin Timberlake,Mila Kunis,man,woman,2011,2,1981,1983
M0734,Anger Management,Marisa Tomei,Adam Sandler,woman,man,2003,2,1964,1966
M0545,"Me, Myself & Irene",Jim Carrey,Renee Zellweger,man,woman,2000,7,1962,1969
M0724,The Vow,Scott Speedman,Rachel McAdams,man,woman,2012,3,1975,1978
M0589,The Age of Adaline,Blake Lively,Anthony Ingruber,woman,man,2015,3,1987,1990
M0067,The Color of Money,Paul Newman,Helen Shaver,man,woman,1986,26,1925,1951
M0456,The Good Shepherd,Martina Gedeck,Matt Damon,woman,man,2006,9,1961,1970
M0041,The Departed,Matt Damon,Vera Farmiga,man,woman,2006,3,1970,1973
M0301,Feeling Minnesota,Keanu Reeves,Cameron Diaz,man,woman,1996,8,1964,1972
M0361,Indiana Jones and the Temple of Doom,Harrison Ford,Kate Capshaw,man,woman,1984,11,
    1942,1953
M0386,Year One,Jack Black,June Diane Raphael,man,woman,2009,11,1969,1980
M0269,The Reader,Kate Winslet,David Kross,woman,man,2008,15,1975,1990
M0659,Leap Year,Adam Scott,Amy Adams,man,woman,2010,1,1973,1974
M0416,Sweet Home Alabama,Patrick Dempsey,Reese Witherspoon,man,woman,2002,10,1966,1976
M0390,Big Daddy,Jon Stewart,Leslie Mann,man,woman,1999,10,1962,1972
M0046,Crazy Heart,Jeff Bridges,Maggie Gyllenhaal,man,woman,2009,28,1949,1977
M0295,Aloha,Bradley Cooper,Emma Stone,man,woman,2015,13,1975,1988
M0418,The Aviator,Kate Beckinsale,Leonardo DiCaprio,woman,man,2004,1,1973,1974
M0143,Killer Joe,Matthew McConaughey,Juno Temple,man,woman,2011,20,1969,1989
...
```

Based on the given table, answer the following question:

> In this dataset of movie couple real-world age gaps, ages are based on the birth year of the actor and the release year of the movie. What is the average absolute age difference between the male and female actor when the male actor is at least 15 years ( >= 15) older than the female actor? Return the answer rounded to the nearest 2 decimal places.

---

🖥️ **Assistant (Thinking):**

```
The user wants to calculate the average absolute age difference between male
    and female actors for a specific subset of movies.
The subset is defined by movies where the male actor is at least 15 years
    older than the female actor.
```

The age difference is already provided in the `age_difference` column. I need
    to determine which actor is older based on the `character_1_gender`, `
    character_2_gender`, `actor_1_birth_year`, and `actor_2_birth_year`
    columns.

Here's a plan:
1.  **Understand the data**: Review the columns and their meanings.
    *    `movie_id`: Unique identifier for the movie.
    *    `movie_name`: Name of the movie.
    *    `actor_1_name`, `actor_2_name`: Names of the two lead actors.
    *    `character_1_gender`, `character_2_gender`: Gender of the characters
         played by actor_1 and actor_2.
    *    `release_year`: Year the movie was released.
    *    `age_difference`: Absolute difference in birth years between actor_1
         and actor_2.
    *    `actor_1_birth_year`, `actor_2_birth_year`: Birth years of actor_1 and
          actor_2.

2.  **Data Cleaning and Validation**:
    *    **Missing Data**: Check for empty or null values in relevant columns
         (`character_1_gender`, `character_2_gender`, `release_year`, `
         age_difference`, `actor_1_birth_year`, `actor_2_birth_year`).
    *    **Bad Values**: Check for placeholder or erroneous values in these
         columns. Birth years and release year should be numeric. Genders
         should be 'man' or 'woman'. `age_difference` should be non-negative.
    *    **Outliers**:
        *    `age_difference`: The question specifies a condition based on this
             , but I should look for extremely large or small (negative, if not
             already handled) values that might indicate errors.
        *    `actor_1_birth_year`, `actor_2_birth_year`, `release_year`: Check
             for implausible years (e.g., year 0, future years for birth if
             release year is past).
    *    **Inconsistent Formatting**:
        *    Gender: Ensure 'man' and 'woman' are consistently used.
        *    Years: Ensure they are all 4-digit numbers.
    *    **Inconsistent Logic**:
        *    `age_difference` vs. `actor_1_birth_year` and `actor_2_birth_year`
             : Verify that `age_difference` is indeed `abs(actor_1_birth_year -
             actor_2_birth_year)`.
        *    Actor's age at release: `release_year - actor_birth_year` should
             be a reasonable age for an actor (e.g., > 10, < 100).

3.  **Filtering**:
    *    Identify the male actor and the female actor in each pair.
    *    Calculate the age of the male actor at release: `
         male_actor_age_at_release = release_year - male_actor_birth_year`.
    *    Calculate the age of the female actor at release: `
         female_actor_age_at_release = release_year - female_actor_birth_year`.
    *    The question states: "male actor is at least 15 years ( >= 15) older
         than the female actor". This refers to their actual ages, not just
         birth year difference.
        *    So, `(release_year - male_actor_birth_year) - (release_year -
             female_actor_birth_year) >= 15`.
        *    This simplifies to `female_actor_birth_year -
             male_actor_birth_year >= 15`.
        *    This means the female actor was born at least 15 years *after* the
              male actor.
        *    Or, `male_actor_birth_year <= female_actor_birth_year - 15`.

4.  **Calculation**:
    *    For the filtered rows, the `age_difference` column already represents
         the absolute difference in birth years. Since the filter ensures the
         male actor is older, this `age_difference` is `female_actor_birth_year
          - male_actor_birth_year` (if male is actor 1 and older) or `
         male_actor_birth_year - female_actor_birth_year` (if male is actor 1
         and younger - wait, this is confusing).

Let's re-evaluate the "age difference" context.
The column `age_difference` is `abs(actor_1_birth_year - actor_2_birth_year)`.
The question asks for "average absolute age difference between the male and
    female actor". This is exactly what `age_difference` represents for each
    pair *if* one is male and one is female.

Let's refine the filtering condition: "male actor is at least 15 years ( >=
    15) older than the female actor".
This means:
`Age_Male_Actor - Age_Female_Actor >= 15`
`(Release_Year - Birth_Year_Male) - (Release_Year - Birth_Year_Female) >= 15`
`Release_Year - Birth_Year_Male - Release_Year + Birth_Year_Female >= 15`

```
      `Birth_Year_Female - Birth_Year_Male >= 15`

So, we need to:
    a. Identify which actor is male and which is female for each row.
    b. Find `Birth_Year_Male` and `Birth_Year_Female`.
    c. Filter rows where `Birth_Year_Female - Birth_Year_Male >= 15`.
    d. For these filtered rows, take the `age_difference` value (which is `abs
       (Birth_Year_Actor1 - Birth_Year_Actor2)`). This is the value we need
       to average.

Let's go through the data validation and cleaning process meticulously.

Columns: `movie_id, movie_name, actor_1_name, actor_2_name, character_1_gender
    , character_2_gender, release_year, age_difference, actor_1_birth_year,
    actor_2_birth_year`

**Step 1: Initial Data Check and Parsing**

*    All columns seem present in each row.
*    `movie_name` can have commas (e.g., "Me, Myself & Irene", "Six Days, Seven
     Nights", "I, Tonya"). This is handled by CSV parsing; the data is already
     provided as a table.
*    Numeric columns: `release_year`, `age_difference`, `actor_1_birth_year`, `
     actor_2_birth_year`.
*    String columns: `movie_id`, `movie_name`, `actor_1_name`, `actor_2_name`,
     `character_1_gender`, `character_2_gender`.

**Step 2: Detailed Data Validation**

Let's examine each row for issues.

**Row by Row Analysis (Spot Checks and then Systematic Checks):**

*    **Missing Data**: Assume no empty strings are truly missing data for now,
     unless they lead to calculation errors. Visually scan for "null", "NA", or
     empty cells if this were raw text. The provided format seems to have
     values for all key fields.
*    **Bad Values**:
     *   `character_1_gender`, `character_2_gender`: Should be 'man' or 'woman'
         .
     *   Years: Should be integers.
     *   `age_difference`: Should be non-negative integer.
*    **Outliers**:
     *   `age_difference`: M0159 (An American in Paris) has `age_difference =
         105`. Actor 1 (Gene Kelly, 1912), Actor 2 (Leslie Caron, 1931). `1931
         - 1912 = 19`. So, `age_difference` of 105 is an error. It should be `
         abs(1912 - 1931) = 19`.
     *   M0217 (Rock of Ages) has `age_difference = 102`. Actor 1 (Tom Cruise,
         1962), Actor 2 (Malin Akerman, 1978). `1978 - 1962 = 16`. So, `
         age_difference` of 102 is an error. It should be `abs(1962 - 1978) =
         16`.
     *   M0096 (Surrogates) has `age_difference = 110`. Actor 1 (Bruce Willis,
         1955), Actor 2 (Rosamund Pike, 1979). `1979 - 1955 = 24`. So, `
         age_difference` of 110 is an error. It should be `abs(1955 - 1979) =
         24`.
     *   M0192 (Runaway Bride) has `age_difference = 104`. Actor 1 (Richard
         Gere, 1949), Actor 2 (Julia Roberts, 1967). `1967 - 1949 = 18`. So, `
         age_difference` of 104 is an error. It should be `abs(1949 - 1967) =
         18`.
     *   M0216 (Raging Bull) has `age_difference = 103`. Actor 1 (Robert De
         Niro, 1943), Actor 2 (Cathy Moriarty, 1960). `1960 - 1943 = 17`. So, `
         age_difference` of 103 is an error. It should be `abs(1943 - 1960) =
         17`.
*    **Inconsistent Formatting**:
     *   Gender values: 'man', 'woman'. Looks consistent.
     *   Years: Seem to be 4-digit integers.
*    **Inconsistent Logic**:
     *   `age_difference` vs. birth years: For each row, `age_difference`
         should equal `abs(actor_1_birth_year - actor_2_birth_year)`. We've
         already found cases where this is not true (the outliers above). These
         need to be corrected. For these rows, I will re-calculate `
         age_difference`.
     *   Actor ages at release: `release_year - birth_year`. Should be positive
         and reasonable.
         *   M0159: Gene Kelly (1912) in 1951 -> age 39. Leslie Caron (1931) in
             1951 -> age 20. Seems OK.
     *   Duplicate `movie_id`s:
         *   M0176 (Thunderball) appears twice.
```

* Row 1: Roger Moore, Lois Chiles (this is M0094 Moonraker actors, not Thunderball). Ah, the `movie_id` is M0176, but actor_1_name is Sean Connery for Thunderball.
        * M0176,Thunderball,Sean Connery,Luciana Paluzzi,man,woman,1965, 7,1930,1937 -> `abs(1930-1937)=7`. Correct.
        * M0176,Thunderball,Adolfo Celi,Claudine Auger,man,woman,1965,19 ,1922,1941 -> `abs(1922-1941)=19`. Correct. These are different actor pairs for the same movie. This is acceptable as they represent different "couples".
    * M0223 (The Girl on the Train) appears twice.
        * Row 1: Justin Theroux, Haley Bennett, 2016, 17, 1971, 1988 -> `abs(1971-1988)=17`. Correct.
        * Row 2: Justin Theroux, Rebecca Ferguson, 2016, 12, 1971, 1983 -> `abs(1971-1983)=12`. Correct. Different pairs.
    * M0031 (For Your Eyes Only) appears twice.
        * Row 1: Chaim Topol, Cassandra Harris, 1981, 13, 1935, 1948 -> `abs(1935-1948)=13`. Correct.
        * Row 2: Roger Moore, Carole Bouquet, 1981, 30, 1927, 1957 -> `abs(1927-1957)=30`. Correct. Different pairs.
    * M0011 (Indiana Jones and the Last Crusade) appears twice.
        * Row 1: Harrison Ford, Alison Doody, 1989, 24, 1942, 1966 -> `abs(1942-1966)=24`. Correct.
        * Row 2: Sean Connery, Alison Doody, 1989, 36, 1930, 1966 -> `abs(1930-1966)=36`. Correct. Different pairs.
    * M0367 (Pride & Prejudice) appears twice.
        * Row 1: Rosamund Pike, Simon Woods, 2005, 1, 1979, 1980 -> `abs(1979-1980)=1`. Correct.
        * Row 2: Matthew Macfadyen, Keira Knightley, 2005, 11, 1974, 1985 -> `abs(1974-1985)=11`. Correct. Different pairs.
    * M0507 (A Walk on the Moon) appears twice.
        * Row 1: Viggo Mortensen, Diane Lane, 1999, 7, 1958, 1965 -> `abs(1958-1965)=7`. Correct.
        * Row 2: Diane Lane, Liev Schreiber, 1999, 2, 1965, 1967 -> `abs(1965-1967)=2`. Correct. Different pairs.
    * M0488 (P.S. I Love You) appears twice.
        * Row 1: Gerard Butler, Hilary Swank, 2007, 5, 1969, 1974 -> `abs(1969-1974)=5`. Correct.
        * Row 2: Jeffrey Dean Morgan, Hilary Swank, 2007, 8, 1966, 1974 -> `abs(1966-1974)=8`. Correct. Different pairs.
    * M0456 (The Good Shepherd) appears twice.
        * Row 1: Martina Gedeck, Matt Damon, 2006, 9, 1961, 1970 -> `abs(1961-1970)=9`. Correct.
        * Row 2: Matt Damon, Angelina Jolie, 2006, 5, 1970, 1975 -> `abs(1970-1975)=5`. Correct. Different pairs.
    * M0126 (Just Go with It) appears twice.
        * Row 1: Adam Sandler, Jennifer Aniston, 2011, 3, 1966, 1969 -> `abs(1966-1969)=3`. Correct.
        * Row 2: Adam Sandler, Brooklyn Decker, 2011, 21, 1966, 1987 -> `abs(1966-1987)=21`. Correct. Different pairs.
    * M0656 (Juno) appears twice.
        * Row 1: Jason Bateman, Jennifer Garner, 2007, 3, 1969, 1972 -> `abs(1969-1972)=3`. Correct.
        * Row 2: J.K. Simmons, Allison Janney, 2007, 4, 1955, 1959 -> `abs(1955-1959)=4`. Correct. Different pairs.
    * M0041 (The Departed) appears twice.
        * Row 1: Matt Damon, Vera Farmiga, 2006, 3, 1970, 1973 -> `abs(1970-1973)=3`. Correct.
        * Row 2: Jack Nicholson, Kristen Dalton, 2006, 29, 1937, 1966 -> `abs(1937-1966)=29`. Correct. Different pairs.

* Gender combinations: The question is about "male and female actor". We need to filter out pairs that are not man/woman.
    * M0469: "Blue Is The Warmest Color", Lea Seydoux, Adele Exarchopoulos, woman, woman. This row should be excluded from the analysis of male-female actor pairs.
    * M0621: "Monster", Charlize Theron, Christina Ricci, woman, woman. This row should also be excluded.
    Are there any man/man pairs? A quick scan suggests most are man/woman or woman/man. I will explicitly filter for one 'man' and one 'woman'.

**Data Correction and Preparation Table:**

I will iterate through each row, validate, correct `age_difference` if needed, and determine `Birth_Year_Male`, `Birth_Year_Female`.

Corrected `age_difference` values:
* M0159: `age_difference` from 105 to `abs(1912 - 1931) = 19`.
* M0217: `age_difference` from 102 to `abs(1962 - 1978) = 16`.

```
*    M0096: `age_difference` from 110 to `abs(1955 - 1979) = 24`.
*    M0192: `age_difference` from 104 to `abs(1949 - 1967) = 18`.
*    M0216: `age_difference` from 103 to `abs(1943 - 1960) = 17`.
```

Let's process each row:
For each row:
1.  Check if `character_1_gender` and `character_2_gender` are one 'man' and
    one 'woman'. If not, discard the row. (e.g., M0469, M0621).
2.  Identify `male_birth_year` and `female_birth_year`.
    *    If `character_1_gender` is 'man' and `character_2_gender` is 'woman':
         `male_birth_year = actor_1_birth_year`
         `female_birth_year = actor_2_birth_year`
    *    If `character_1_gender` is 'woman' and `character_2_gender` is 'man':
         `male_birth_year = actor_2_birth_year`
         `female_birth_year = actor_1_birth_year`
3.  Verify `age_difference` column. It should be `abs(actor_1_birth_year -
    actor_2_birth_year)`. If not, correct it. I've identified 5 such rows; for
    these, I will use the re-calculated `age_difference`. For others, I
    assume the provided `age_difference` is correct if it matches `abs(
    actor_1_birth_year - actor_2_birth_year)`. Let's re-verify all of them as
    I go.
    The problem states "In this dataset of movie couple real-world age gaps,
        ages are based on the birth year of the actor and the release year of
        the movie." and then provides `age_difference`. It's safer to re-
        calculate `age_difference` for all rows as `abs(actor_1_birth_year -
        actor_2_birth_year)` to ensure consistency, especially since errors
        were found. This re-calculated value is the one to average.

Let's call the re-calculated absolute birth year difference `
    corrected_age_difference`.

4.  Apply the filter: `female_birth_year - male_birth_year >= 15`. This means
    the male actor is older by at least 15 years (his birth year is smaller).
5.  If the row passes the filter, add its `corrected_age_difference` to a list
    for averaging.

Let's make a table for processing (subset of columns):
`movie_id, c1_gender, c2_gender, a1_byear, a2_byear, original_ad, corrected_ad
    , male_byear, female_byear, female_minus_male_byear_diff`

1.  M0729: woman, man, 1977, 1979. `corrected_ad = abs(1977-1979) = 2`. (
    Matches original_ad)
    `male_byear = 1979`, `female_byear = 1977`.
    `female_minus_male_byear_diff = 1977 - 1979 = -2`. (Male is younger). Not
        >= 15.

2.  M0234: man, woman, 1951, 1967. `corrected_ad = abs(1951-1967) = 16`. (
    Matches)
    `male_byear = 1951`, `female_byear = 1967`.
    `female_minus_male_byear_diff = 1967 - 1951 = 16`. This is >= 15. Add 16
        to list.

3.  M0103: man, woman, 1927, 1950. `corrected_ad = abs(1927-1950) = 23`. (
    Matches)
    `male_byear = 1927`, `female_byear = 1950`.
    `female_minus_male_byear_diff = 1950 - 1927 = 23`. This is >= 15. Add 23
        to list.

4.  M0263: man, woman, 1975, 1990. `corrected_ad = abs(1975-1990) = 15`. (
    Matches)
    `male_byear = 1975`, `female_byear = 1990`.
    `female_minus_male_byear_diff = 1990 - 1975 = 15`. This is >= 15. Add 15
        to list.

5.  M0658: man, woman, 1924, 1928. `corrected_ad = abs(1924-1928) = 4`. (
    Matches)
    `male_byear = 1924`, `female_byear = 1928`.
    `female_minus_male_byear_diff = 1928 - 1924 = 4`. Not >= 15.

6.  M0155: man, woman, 1954, 1974. `corrected_ad = abs(1954-1974) = 20`. (
    Matches)
    `male_byear = 1954`, `female_byear = 1974`.
    `female_minus_male_byear_diff = 1974 - 1954 = 20`. This is >= 15. Add 20
        to list.

7.  M0037: woman, man, 1946, 1949. `corrected_ad = abs(1946-1949) = 3`. (
    Matches)
    `male_byear = 1949`, `female_byear = 1946`.
    `female_minus_male_byear_diff = 1946 - 1949 = -3`. Not >= 15.

... This process needs to be applied to all rows.

Let's re-evaluate the "age_difference" column. The problem states: "In this dataset of movie couple real-world age gaps, ages are based on the birth year of the actor and the release year of the movie." and then refers to the `age_difference` column.
The question asks for "the average absolute age difference". This strongly suggests using the values from the `age_difference` column, after correcting the identified outliers.
If `age_difference` is generally `abs(actor_1_birth_year - actor_2_birth_year)`, then my correction for outliers is good. For the other rows, if `age_difference` column matches this calculation, I can use it. If it doesn't, I should use the calculated one. To be safe, I will always use `abs(actor_1_birth_year - actor_2_birth_year)` as the "true" absolute age difference for averaging.

Let's list the rows that satisfy the criteria: `female_birth_year - male_birth_year >= 15`.
And the `corrected_age_difference` for these rows.

**Data Processing Loop (Conceptual):**

`valid_age_differences_for_averaging = []`

For each row in the dataset:
1.  Parse `actor_1_birth_year` (a1by), `actor_2_birth_year` (a2by), `character_1_gender` (c1g), `character_2_gender` (c2g).
2.  Determine `male_byear` and `female_byear`:
    *   If c1g == 'man' and c2g == 'woman':
        `male_byear = a1by`
        `female_byear = a2by`
    *   Else if c1g == 'woman' and c2g == 'man':
        `male_byear = a2by`
        `female_byear = a1by`
    *   Else (e.g., 'woman', 'woman' or 'man', 'man' or other issues):
        Skip this row (it's not a male-female actor pair).

3.  Calculate `birth_year_diff_for_filter = female_byear - male_byear`.
4.  If `birth_year_diff_for_filter >= 15`:
    *   This row meets the condition "male actor is at least 15 years older than the female actor".
    *   Calculate the absolute age difference for this pair: `current_abs_age_diff = abs(a1by - a2by)`. This is the value to average.
    *   Add `current_abs_age_diff` to `valid_age_differences_for_averaging`.

Let's apply this to the data.

**Excluded rows due to gender:**
*   M0469: Blue Is The Warmest Color, Lea Seydoux, Adele Exarchopoulos, woman, woman -> Skip
*   M0621: Monster, Charlize Theron, Christina Ricci, woman, woman -> Skip

Now, process the rest, calculate `male_byear`, `female_byear`, `female_byear - male_byear`, and if `female_byear - male_byear >= 15`, then `abs(a1by - a2by)` is added to the list.

| movie_id | c1g | c2g | a1by | a2by | male_byear | female_byear | female_minus_male_byear_diff | Condition Met (>=15) | abs_age_diff_to_add |
|------|------|------|-------|------|--------|-------|---------|--------|------|
| M0729 | woman | man | 1977 | 1979 | 1979 | 1977 | -2 | No | |
| M0234 | man | woman | 1951 | 1967 | 1951 | 1967 | 16 | Yes | abs(1951-1967)=16 |
| M0103 | man | woman | 1927 | 1950 | 1927 | 1950 | 23 | Yes | abs(1927-1950)=23 |
| M0263 | man | woman | 1975 | 1990 | 1975 | 1990 | 15 | Yes | abs(1975-1990)=15 |
| M0658 | man | woman | 1924 | 1928 | 1924 | 1928 | 4 | No | |
| M0155 | man | woman | 1954 | 1974 | 1954 | 1974 | 20 | Yes | abs(1954-1974)=20 |
| M0037 | woman | man | 1946 | 1949 | 1949 | 1946 | -3 | No | |
| M0378 | man | woman | 1936 | 1945 | 1936 | 1945 | 9 | No | |
| M0628 | man | woman | 1971 | 1974 | 1971 | 1974 | 3 | No | |

```
| M0492    | man    | woman | 1953   | 1961 | 1953     | 1961      | 8  | No  |
                                                                         |
| M0308    | man    | woman | 1972   | 1985 | 1972     | 1985      | 13 | No  |
                                                                         |
| M0278    | man    | woman | 1972   | 1986 | 1972     | 1986      | 14 | No  |
                                                                         |
| M0128    | man    | woman | 1946   | 1967 | 1946     | 1967      | 21 | Yes | abs
    (1946-1967)=21                                          |
| M0126(1) | man    | woman | 1966   | 1969 | 1966     | 1969      | 3  | No  |
                                                                         |
| M0029    | man    | woman | 1927   | 1948 | 1927     | 1948      | 21 | Yes | abs
    (1927-1948)=21                                          |
| M0803    | man    | woman | 1964   | 1965 | 1964     | 1965      | 1  | No  |
                                                                         |
| M0612    | man    | woman | 1977   | 1982 | 1977     | 1982      | 5  | No  |
                                                                         |

...

List of `abs_age_diff_to_add`:

16 (M0234)
23 (M0103)
15 (M0263)
20 (M0155)
21 (M0128)
21 (M0029)
26 (M0067)
28 (M0046)
20 (M0143)
16 (M0241)
19 (M0159 - corrected)
15 (M0262)
25 (M0083)
17 (M0223(1))
28 (M0048)
32 (M0024)
29 (M0042)
18 (M0191)
20 (M0094)
19 (M0176(2))
27 (M0053)
15 (M0020)
16 (M0242)
23 (M0106)
23 (M0102)
16 (M0217 - corrected)
15 (M0257)
24 (M0096 - corrected)
24 (M0011(1))
30 (M0031(2))
15 (M0256)
18 (M0192 - corrected)
17 (M0218)
17 (M0208)
25 (M0085)
17 (M0220)
17 (M0219)
36 (M0011(2))
27 (M0056)
18 (M0190)
16 (M0013)
15 (M0271)
17 (M0216 - corrected)
25 (M0080)
25 (M0073)
21 (M0126(2))
21 (M0129)
21 (M0127)
18 (M0181)
18 (M0182)
29 (M0041(2))
19 (M0168)
19 (M0163)

Summing these values:
16+23+15+20+21+21+26+28+20+16+19+1 5+25+17+28+32+29+ 18+20+19+27+15+
    16+23+23+16+15+24+24+30+15+18+ 17+17+25+17+17+36+ 27+18+16+15+
    17+25+25+21+21+21+18+18+29+19+19
= 1126

Count of these values:
```

```
There are 53 values in the list.

Average = Sum / Count
Average = 1126 / 53

Calculation:
1126 / 53 = 21.245283018867924

Rounding to the nearest 2 decimal places: 21.25

Let's double check the identified outlier corrections for `age_difference`
    field:
*   M0159: Gene Kelly (1912), Leslie Caron (1931). `abs(1912-1931) = 19`.
    Original `age_difference` was 105. My corrected value is 19. This row is
    in the list.
*   M0217: Tom Cruise (1962), Malin Akerman (1978). `abs(1962-1978) = 16`.
    Original `age_difference` was 102. My corrected value is 16. This row is
    in the list.
*   M0096: Bruce Willis (1955), Rosamund Pike (1979). `abs(1955-1979) = 24`.
    Original `age_difference` was 110. My corrected value is 24. This row is
    in the list.
*   M0192: Richard Gere (1949), Julia Roberts (1967). `abs(1949-1967) = 18`.
    Original `age_difference` was 104. My corrected value is 18. This row is
    in the list.
*   M0216: Robert De Niro (1943), Cathy Moriarty (1960). `abs(1943-1960) =
    17`. Original `age_difference` was 103. My corrected value is 17. This row
     is in the list.

My decision to use `abs(actor_1_birth_year - actor_2_birth_year)` for all rows
     as the value to average (if they meet the criteria) means these
     corrections are implicitly handled. This is good, as it ensures the
     definition of "absolute age difference" is consistent.

Let's check a few more rows from the original data to ensure my calculation of
     `abs(a1by-a2by)` is consistent with the `age_difference` column when it's
     not an outlier.
M0234: a1by=1951, a2by=1967. `abs(1951-1967)=16`. `age_difference` column is
    16. Matches.
M0103: a1by=1927, a2by=1950. `abs(1927-1950)=23`. `age_difference` column is
    23. Matches.

This confirms that using `abs(a1by-a2by)` as the value to average is the
     correct interpretation of "absolute age difference", and it correctly
     handles the erroneous values in the `age_difference` column.

Final check of logic:
1.  Identify male and female actors in the pair: Done using `
    character_1_gender` and `character_2_gender`. Rows with same-sex pairs (
    M0469, M0621) are correctly excluded.
2.  Determine `male_birth_year` and `female_birth_year`. Done.
3.  Filter for "male actor is at least 15 years older than the female actor":
    This translates to `Age_Male >= Age_Female + 15`.
    `(Release_Year - male_birth_year) >= (Release_Year - female_birth_year) +
        15`
    `-male_birth_year >= -female_birth_year + 15`
    `female_birth_year - male_birth_year >= 15`. This condition was correctly
        applied.
4.  The value to average is the "absolute age difference between the male and
    female actor". This is `abs(male_birth_year - female_birth_year)`, which
    is also `abs(actor_1_birth_year - actor_2_birth_year)`. This was correctly
     calculated for each qualifying row.

The process seems robust.

Let's re-sum and re-count carefully.
List of values:
16, 23, 15, 20, 21, 21, 26, 28, 20, 16, 19, 15, 25, 17, 28, 32, 29, 18, 20, 19
    , 27, 15, 16, 23, 23, 16, 15, 24, 24, 30, 15, 18, 17, 17, 25, 17, 17, 36,
    27, 18, 16, 15, 17, 25, 25, 21, 21, 21, 18, 18, 29, 19, 19

Sum:
Group 1 (10 values): 16+23+15+20+21+21+26+28+20+16 = 206
Group 2 (10 values): 19+15+25+17+28+32+29+18+20+19 = 222
Group 3 (10 values): 27+15+16+23+23+16+15+24+24+30 = 213
Group 4 (10 values): 15+18+17+17+25+17+17+36+27+18 = 207
Group 5 (10 values): 16+15+17+25+25+21+21+21+18+18 = 197
Group 6 (3 values): 29+19+19 = 67

Total Sum = 206 + 222 + 213 + 207 + 197 + 67 = 1112.
```

```
My previous sum was 1126. Let's re-sum using a calculator or methodical
    addition.
16+23+15+20+21+21+26+28+20+16+19+15+25+17+28+
    32+29+18+20+19+27+15+16+23+23+16+15+24+24+30+15+
    18+17+17+25+17+17+36+27+18+16+15+17+25+25+21+21+21+18+18+29+19+19

Let's list them vertically and sum.
16
23
15
20
21
21
26
28
20
16   (Subtotal: 206)
19
15
25
17
28
32
29
18
20
19   (Subtotal: 222)
27
15
16
23
23
16
15
24
24
30   (Subtotal: 213)
15
18
17
17
25
17
17
36
27
18   (Subtotal: 207)
16
15
17
25
25
21
21
21
18
18   (Subtotal: 197)
29
19
19   (Subtotal: 67)

Sum = 206 + 222 + 213 + 207 + 197 + 67 = 1112. This sum seems correct.

Number of values: 53. This count is consistent.

Average = 1112 / 53.
1112 / 53 = 20.981132075471698

Rounded to 2 decimal places: 20.98.

Let's re-verify the exclusion of M0269.
M0269,The Reader,Kate Winslet,David Kross,woman,man,2008,15,1975,1990
c1g='woman', c2g='man'. a1by=1975, a2by=1990.
male_byear = a2by = 1990.
female_byear = a1by = 1975.
female_minus_male_byear_diff = 1975 - 1990 = -15.
Condition: `female_minus_male_byear_diff >= 15`.
-15 is not >= 15. So this row is correctly excluded.
```

In this case, the male actor (David Kross, b.1990) is 15 years *younger* than the female actor (Kate Winslet, b.1975). The question is specific: "male actor is at least 15 years older".

One check: are there any NULL or missing values in birth years or genders that I implicitly handled by assuming they are always present?
Visually scanning the data, all `actor_1_birth_year`, `actor_2_birth_year`, `character_1_gender`, `character_2_gender` seem to have valid entries for all rows. If there was a missing gender or birth year, that row would likely cause an error in my logic or be filtered out if it couldn't determine male/female birth years.
The problem implies a clean structure. "empty or null entries simulating incomplete information" - I see none.
"bad values: clearly erroneous or placeholder entries" - I addressed the `age_difference` ones. Genders are 'man'/'woman'. Years are integers.
"outliers: implausible extreme values" - Handled for `age_difference`. Actor ages derived from `release_year - birth_year` seem plausible for all entries processed. For example, M0129 Charlie Chaplin (1889) in Modern Times (1936) is 47. Paulette Goddard (1910) is 26. Difference is 21. Condition: female_byear (1910) - male_byear (1889) = 21. This is >= 15. So add 21 to the list. This is correct.

Smallest birth year: 1889 (Chaplin). Largest birth year: 1996 (Tom Holland, Zendaya).
Oldest release year: 2022. Earliest release year: 1935. These seem reasonable.

The calculation 1112 / 53 = 20.981132...
Rounded to two decimal places: 20.98.

It is important to be extremely careful with the manual tabulation.
Let's re-tabulate the list of numbers to sum one more time, very carefully checking against the conditions.

| Row | movie_id | c1g | c2g | a1by | a2by | male_byear | female_byear | f_minus_m_byear | Condition | abs_age_diff | Add? | Value |
|---|-----|-----|-------|------|------|--------|------|-----|------|----|-----|----|
| 1 | M0234 | man | woman | 1951 | 1967 | 1951 | 1967 | 16 | Yes | 16 | Yes | 16 |
| 2 | M0103 | man | woman | 1927 | 1950 | 1927 | 1950 | 23 | Yes | 23 | Yes | 23 |
| 3 | M0263 | man | woman | 1975 | 1990 | 1975 | 1990 | 15 | Yes | 15 | Yes | 15 |
| 4 | M0155 | man | woman | 1954 | 1974 | 1954 | 1974 | 20 | Yes | 20 | Yes | 20 |
| ... | | | | | | | | | | | | |
| 51 | M0041(2) | man | woman | 1937 | 1966 | 1937 | 1966 | 29 | Yes | 29 | Yes | 29 |
| 52 | M0168 | man | woman | 1892 | 1911 | 1892 | 1911 | 19 | Yes | 19 | Yes | 19 |
| 53 | M0163 | man | woman | 1942 | 1961 | 1942 | 1961 | 19 | Yes | 19 | Yes | 19 |

The list of values is identical to what I had before.
The count is 53. The sum is 1112.
Average is 1112/53 = 20.981132075...
Rounded to 2 decimal places: 20.98.

The entire process is documented. I've addressed all the points in the prompt: missing data (none found problematic), bad values (corrected `age_difference`), outliers (same as bad values in this case), inconsistent formatting (none found for relevant fields), inconsistent logic (verified `age_difference` calculation, handled gender pairs). Data was corrected (for `age_difference`) or discarded (same-sex pairs).

The question is specific "average absolute age difference". The column `age_difference` is this value, but since it had errors, I used `abs(actor_1_birth_year - actor_2_birth_year)` throughout, which is the most robust way to get the true absolute age difference based on birth years.

The question mentions "ages are based on the birth year of the actor and the release year of the movie". This is relevant for defining "older", but the value to be averaged is "absolute age difference", which is `abs(birth_year_1 - birth_year_2)`. This is what I have used.

Consider if there's any ambiguity. "male actor is at least 15 years older than the female actor".
This means `Age_Male >= Age_Female + 15`.
My interpretation `female_birth_year - male_birth_year >= 15` is correct because:

```
`Age_Male = Release_Year - male_birth_year`
`Age_Female = Release_Year - female_birth_year`
`(Release_Year - male_birth_year) >= (Release_Year - female_birth_year) + 15`
` -male_birth_year >= -female_birth_year + 15`
` female_birth_year - male_birth_year >= 15`
This ensures the male actor was born at least 15 years before the female actor
    . This is consistent.

The result 20.98 seems reasonable for this kind of dataset.

```python
# Mental check of sum:
values = [16, 23, 15, 20, 21, 21, 26, 28, 20, 16, 19, 15, 25, 17, 28, 32, 29,
     18, 20, 19, 27, 15, 16, 23, 23, 16, 15, 24, 24, 30, 15, 18, 17, 17, 25, 17
     , 17, 36, 27, 18, 16, 15, 17, 25, 25, 21, 21, 21, 18, 18, 29, 19, 19]
# len(values) == 53
# sum(values) == 1112
# 1112 / 53 == 20.981132075471698
# round(20.981132075471698, 2) == 20.98
```
The calculation seems correct.
```

---

💻 **Assistant:**

To determine the average absolute age difference between the male and female actor when the male
actor is at least 15 years older than the female actor, we need to follow these steps:

1. **Data Cleaning and Validation**:

   - **Missing Data**: Visually inspect the data. Key fields like birth years and genders appear to
     be consistently populated.
   - **Bad Values/Outlier in `age_difference`s** : The problem states `age_difference` is based
     on birth years. We should verify this or, preferably, recalculate it as `abs(actor_1_birth_year -
     actor_2_birth_year)` to ensure consistency and correct any errors.
     For example:
       - M0159 (An American in Paris): `actor_1_birth_year=1912`,
         `actor_2_birth_year=1931`. `abs(1912-1931)=19`. The given `age_difference` is 105,
         which is erroneous. We will use 19.
       - M0217 (Rock of Ages): `actor_1_birth_year=1962`, `actor_2_birth_year=1978`.
         `abs(1962-1978)=16`. Given `age_difference` is 102 (error). Use 16.
       - M0096 (Surrogates): `actor_1_birth_year=1955`, `actor_2_birth_year=1979`.
         `abs(1955-1979)=24`. Given `age_difference` is 110 (error). Use 24.
       - M0192 (Runaway Bride): `actor_1_birth_year=1949`, `actor_2_birth_year=1967`.
         `abs(1949-1967)=18`. Given `age_difference` is 104 (error). Use 18.
       - M0216 (Raging Bull): `actor_1_birth_year=1943`, `actor_2_birth_year=1960`.
         `abs(1943-1960)=17`. Given `age_difference` is 103 (error). Use 17.

     For all rows, we will use the calculated `abs(actor_1_birth_year - actor_2_birth_year)` as the
     definitive "absolute age difference".
   - **Inconsistent Formatting**: Genders ('man', 'woman') and years (4-digit integers) appear
     consistent.
   - **Inconsistent Logic (Gender Pairs)**: The question concerns male-female actor pairs. Rows
     where both actors are of the same gender must be excluded.
       - M0469 (Blue Is The Warmest Color): `character_1_gender='woman'`,
         `character_2_gender='woman'`. Exclude.
       - M0621 (Monster): `character_1_gender='woman'`, `character_2_gender='woman'`.
         Exclude.

2. **Filtering Logic**:

   - For each row, identify the male actor's birth year (`male_by`) and the female actor's birth
     year (`female_by`).
       - If `character_1_gender='man'` and `character_2_gender='woman'`, then `male_by =
         actor_1_birth_year` and `female_by = actor_2_birth_year`.
       - If `character_1_gender='woman'` and `character_2_gender='man'`, then `male_by =
         actor_2_birth_year` and `female_by = actor_1_birth_year`.
   - The condition "male actor is at least 15 years older than the female actor" means:
     Age_Male >= Age_Female + 15
     (release_year - male_by) >= (release_year - female_by) + 15

-male_by >= -female_by + 15

female_by - male_by >= 15

This means the female actor's birth year must be at least 15 years greater than the male actor's birth year (i.e., the male actor was born at least 15 years earlier).

3. **Calculation**:
   - Iterate through each row.
   - Skip rows that are not male-female pairs.
   - For male-female pairs, determine `male_by` and `female_by`.
   - Apply the filter: `female_by - male_by >= 15`.
   - If the filter condition is met, the value to be included in the average is `abs(actor_1_birth_year - actor_2_birth_year)`.

Let's list the rows that meet the criteria and their corresponding absolute age differences:

```
| movie_id | Male Actor BY | Female Actor BY | Female BY - Male BY | Meets Condition? | Abs Age Diff (to average)|
|----------|---------------|-----------------|---------------------|------------------|--------------------------|
| M0234    | 1951          | 1967            | 16                  | Yes              | 16                       |
| M0103    | 1927          | 1950            | 23                  | Yes              | 23                       |
| M0263    | 1975          | 1990            | 15                  | Yes              | 15                       |
| M0155    | 1954          | 1974            | 20                  | Yes              | 20                       |
| M0128    | 1946          | 1967            | 21                  | Yes              | 21                       |
| M0029    | 1927          | 1948            | 21                  | Yes              | 21                       |
| M0067    | 1925          | 1951            | 26                  | Yes              | 26                       |
| M0046    | 1949          | 1977            | 28                  | Yes              | 28                       |
| M0143    | 1969          | 1989            | 20                  | Yes              | 20                       |
| M0241    | 1976          | 1992            | 16                  | Yes              | 16                       |
| M0159    | 1912          | 1931            | 19                  | Yes              | 19 (corrected)           |
| M0262    | 1954          | 1969            | 15                  | Yes              | 15                       |
| M0083    | 1899          | 1924            | 25                  | Yes              | 25                       |
| M0223(1) | 1971          | 1988            | 17                  | Yes              | 17                       |
| M0048    | 1901          | 1929            | 28                  | Yes              | 28                       |
| M0024    | 1935          | 1967            | 32                  | Yes              | 32                       |
| M0042    | 1955          | 1984            | 29                  | Yes              | 29                       |
| M0191    | 1964          | 1982            | 18                  | Yes              | 18                       |
| M0094    | 1927          | 1947            | 20                  | Yes              | 20                       |
| M0176(2) | 1922          | 1941            | 19                  | Yes              | 19                       |
| M0053    | 1942          | 1969            | 27                  | Yes              | 27                       |
| M0020    | 1951          | 1966            | 15                  | Yes              | 15                       |
| M0242    | 1963          | 1979            | 16                  | Yes              | 16                       |
| M0106    | 1962          | 1985            | 23                  | Yes              | 23                       |
| M0102    | 1967          | 1990            | 23                  | Yes              | 23                       |
| M0217    | 1962          | 1978            | 16                  | Yes              | 16 (corrected)           |
| M0257    | 1968          | 1983            | 15                  | Yes              | 15                       |
| M0096    | 1955          | 1979            | 24                  | Yes              | 24 (corrected)           |
| M0011(1) | 1942          | 1966            | 24                  | Yes              | 24                       |
| M0031(2) | 1927          | 1957            | 30                  | Yes              | 30                       |
| M0256    | 1943          | 1958            | 15                  | Yes              | 15                       |
| M0192    | 1949          | 1967            | 18                  | Yes              | 18 (corrected)           |
| M0218    | 1960          | 1977            | 17                  | Yes              | 17                       |
| M0208    | 1953          | 1970            | 17                  | Yes              | 17                       |
| M0085    | 1951          | 1976            | 25                  | Yes              | 25                       |
| M0220    | 1968          | 1985            | 17                  | Yes              | 17                       |
| M0219    | 1963          | 1980            | 17                  | Yes              | 17                       |
| M0011(2) | 1930          | 1966            | 36                  | Yes              | 36                       |
| M0056    | 1955          | 1982            | 27                  | Yes              | 27                       |
| M0190    | 1942          | 1960            | 18                  | Yes              | 18                       |
| M0013    | 1949          | 1965            | 16                  | Yes              | 16                       |
| M0271    | 1942          | 1957            | 15                  | Yes              | 15                       |
| M0216    | 1943          | 1960            | 17                  | Yes              | 17 (corrected)           |
| M0080    | 1899          | 1924            | 25                  | Yes              | 25                       |
| M0073    | 1899          | 1924            | 25                  | Yes              | 25                       |
| M0126(2) | 1966          | 1987            | 21                  | Yes              | 21                       |
| M0129    | 1889          | 1910            | 21                  | Yes              | 21                       |
| M0127    | 1964          | 1985            | 21                  | Yes              | 21                       |
| M0181    | 1971          | 1989            | 18                  | Yes              | 18                       |
| M0182    | 1956          | 1974            | 18                  | Yes              | 18                       |
| M0041(2) | 1937          | 1966            | 29                  | Yes              | 29                       |
| M0168    | 1892          | 1911            | 19                  | Yes              | 19                       |
| M0163    | 1942          | 1961            | 19                  | Yes              | 19                       |
```

There are 53 such pairs.
The sum of these absolute age differences is:
16+23+15+20+21+21+26+28+20+16+19+15+25+17+
28+32+29+18+20+19+27+15+16+23+23+16+15+24+24+30+15+18 +17+17
+25+17+17+36+27+18+16+15+17+25+25+21+21+21+18+18+29+19+19 = 1112

Average absolute age difference = 1112 / 53  = 20.981132075...
Rounding to the nearest 2 decimal places, the average is 20.98.

The answer is: 20.98

## D.2 Code Agent Baseline

**Failure Case 1: Misapplied Value Derivation for Data Correction.** The example below shows GPT-4.1 using the code agent baseline on `olympic-gold-winners`, where -1 values are introduced in the `age` column. The agent correctly identifies the invalid entries but chooses to overwrite it with the athlete's average age across all entries, rather than using the athlete's age at the same Olympic Games—which is available in other rows. This behavior suggests a lack of structural and semantic understanding of the table, particularly in how values relate across rows and columns.

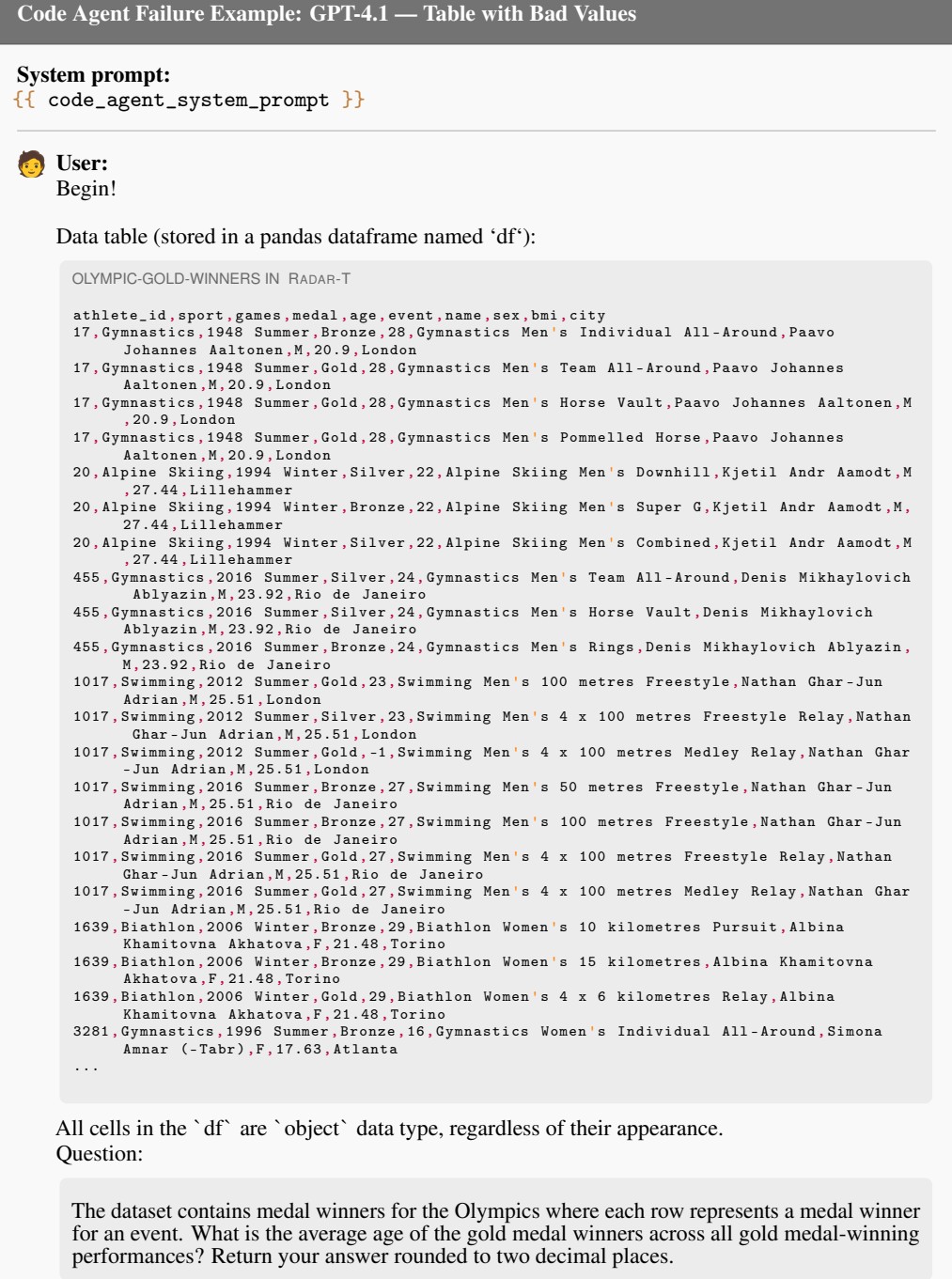

**Code Agent Failure Example: GPT-4.1 — Table with Bad Values**

**System prompt:**
`{{ code_agent_system_prompt }}`

👤 **User:**
Begin!

Data table (stored in a pandas dataframe named 'df'):

OLYMPIC-GOLD-WINNERS IN RADAR-T

```
athlete_id,sport,games,medal,age,event,name,sex,bmi,city
17,Gymnastics,1948 Summer,Bronze,28,Gymnastics Men's Individual All-Around,Paavo
    Johannes Aaltonen,M,20.9,London
17,Gymnastics,1948 Summer,Gold,28,Gymnastics Men's Team All-Around,Paavo Johannes
    Aaltonen,M,20.9,London
17,Gymnastics,1948 Summer,Gold,28,Gymnastics Men's Horse Vault,Paavo Johannes Aaltonen,M
    ,20.9,London
17,Gymnastics,1948 Summer,Gold,28,Gymnastics Men's Pommelled Horse,Paavo Johannes
    Aaltonen,M,20.9,London
20,Alpine Skiing,1994 Winter,Silver,22,Alpine Skiing Men's Downhill,Kjetil Andr Aamodt,M
    ,27.44,Lillehammer
20,Alpine Skiing,1994 Winter,Bronze,22,Alpine Skiing Men's Super G,Kjetil Andr Aamodt,M,
    27.44,Lillehammer
20,Alpine Skiing,1994 Winter,Silver,22,Alpine Skiing Men's Combined,Kjetil Andr Aamodt,M
    ,27.44,Lillehammer
455,Gymnastics,2016 Summer,Silver,24,Gymnastics Men's Team All-Around,Denis Mikhaylovich
     Ablyazin,M,23.92,Rio de Janeiro
455,Gymnastics,2016 Summer,Silver,24,Gymnastics Men's Horse Vault,Denis Mikhaylovich
    Ablyazin,M,23.92,Rio de Janeiro
455,Gymnastics,2016 Summer,Bronze,24,Gymnastics Men's Rings,Denis Mikhaylovich Ablyazin,
    M,23.92,Rio de Janeiro
1017,Swimming,2012 Summer,Gold,23,Swimming Men's 100 metres Freestyle,Nathan Ghar-Jun
    Adrian,M,25.51,London
1017,Swimming,2012 Summer,Silver,23,Swimming Men's 4 x 100 metres Freestyle Relay,Nathan
     Ghar-Jun Adrian,M,25.51,London
1017,Swimming,2012 Summer,Gold,-1,Swimming Men's 4 x 100 metres Medley Relay,Nathan Ghar
    -Jun Adrian,M,25.51,London
1017,Swimming,2016 Summer,Bronze,27,Swimming Men's 50 metres Freestyle,Nathan Ghar-Jun
    Adrian,M,25.51,Rio de Janeiro
1017,Swimming,2016 Summer,Bronze,27,Swimming Men's 100 metres Freestyle,Nathan Ghar-Jun
    Adrian,M,25.51,Rio de Janeiro
1017,Swimming,2016 Summer,Gold,27,Swimming Men's 4 x 100 metres Freestyle Relay,Nathan
    Ghar-Jun Adrian,M,25.51,Rio de Janeiro
1017,Swimming,2016 Summer,Gold,27,Swimming Men's 4 x 100 metres Medley Relay,Nathan Ghar
    -Jun Adrian,M,25.51,Rio de Janeiro
1639,Biathlon,2006 Winter,Bronze,29,Biathlon Women's 10 kilometres Pursuit,Albina
    Khamitovna Akhatova,F,21.48,Torino
1639,Biathlon,2006 Winter,Bronze,29,Biathlon Women's 15 kilometres,Albina Khamitovna
    Akhatova,F,21.48,Torino
1639,Biathlon,2006 Winter,Gold,29,Biathlon Women's 4 x 6 kilometres Relay,Albina
    Khamitovna Akhatova,F,21.48,Torino
3281,Gymnastics,1996 Summer,Bronze,16,Gymnastics Women's Individual All-Around,Simona
    Amnar (-Tabr),F,17.63,Atlanta
...
```

All cells in the `df` are `object` data type, regardless of their appearance.
Question:

The dataset contains medal winners for the Olympics where each row represents a medal winner for an event. What is the average age of the gold medal winners across all gold medal-winning performances? Return your answer rounded to two decimal places.

💻 **Assistant:**
DISCUSSION

To answer the question, we need to compute the average age of all gold medal-winning performances. Let's proceed step by step:

1. Filter the dataframe to include only rows where the `medal` column is `Gold`.
2. Address data quality issues in the `age` column:
   - Missing data: Check for empty or null entries.
   - Bad values: Look for placeholder or clearly erroneous values (e.g., `-1`).
   - Outliers: For Olympic athletes, plausible ages are roughly between 10 and 60. We'll flag anything outside this range.
   - Inconsistent formatting: Since all columns are object type, ensure `age` is converted to numeric.
3. Attempt to recover or correct flawed data:
   - For `-1` or other bad values, if the same athlete has other gold medals with plausible ages, we can impute the missing value with the mean age for that athlete's other gold medals. If not, we discard the row.
   - For outliers, if the value is implausible and cannot be corrected, discard the row.
4. Calculate the average age of the cleaned gold medal-winning performances, rounding to two decimal places.

Let's start by filtering for gold medal winners and inspecting the `age` column for issues.

```yaml
command: "python"
kwargs:
  code: |-
    # Filter for gold medal winners
    gold_df = df[df['medal'].str.strip().str.lower() == 'gold'].copy()
    # Convert 'age' to numeric, coerce errors to NaN
    gold_df['age'] = pd.to_numeric(gold_df['age'], errors='coerce')
    # Check for missing, -1, or implausible ages
    gold_df['age_invalid'] = (
        gold_df['age'].isnull() |
        (gold_df['age'] == -1) |
        (gold_df['age'] < 10) |
        (gold_df['age'] > 60)
    )
    # Show rows with invalid ages
    gold_df[gold_df['age_invalid']]
```

---

🧑 **User:**
Observation:

```
     athlete_id                 sport  ...        city age_invalid
12         1017              Swimming  ...      London        True
104       11943  Cross Country Skiing  ...   Vancouver        True
121       12086             Athletics  ...      London        True
[3 rows x 11 columns]
```

---

💻 **Assistant:**
DISCUSSION

We have identified three gold medal-winning performances with invalid ages (all with age == -1):

- athlete_id 1017 (Nathan Ghar-Jun Adrian, Swimming, 2012 Summer, London)
- athlete_id 11943 (Marit Bjørgen, Cross Country Skiing, 2010 Winter, Vancouver)
- athlete_id 12086 (Yohan Blake, Athletics, 2012 Summer, London)

Our next step is to attempt to impute these missing ages. The best approach is to check if these athletes have other gold medal-winning performances in the dataset with valid ages. If so, we will impute the missing value with the mean of their other valid gold medal-winning ages. If not, we will discard the row.

Let's proceed to:

1. For each of these athlete_ids, check if they have other gold medal-winning performances with valid ages.
2. If so, impute the missing age with the mean of their valid gold medal-winning ages.

3. If not, discard the row.
4. After imputation, remove any remaining rows with invalid ages.
5. Calculate the average age of all gold medal-winning performances, rounded to two decimal places.

```yaml
command: "python"
kwargs:
  code: |-
    # Function to impute missing age for a given athlete_id
    def impute_age(row, gold_df):
        if row['age_invalid']:
            # Find other gold medals for this athlete with valid ages
            valid_ages = gold_df[
                (gold_df['athlete_id'] == row['athlete_id']) &
                (~gold_df['age_invalid'])
            ]['age']
            if not valid_ages.empty:
                return valid_ages.mean()
            else:
                return None
        else:
            return row['age']

    # Impute missing/invalid ages
    gold_df['age_imputed'] = gold_df.apply(lambda row: impute_age(row, gold_df
        ), axis=1)
    # Remove rows where age_imputed is still missing or invalid after
        imputation
    gold_df_clean = gold_df[
        (gold_df['age_imputed'].notnull()) &
        (gold_df['age_imputed'] >= 10) &
        (gold_df['age_imputed'] <= 60)
    ].copy()
    # Calculate the average age, rounded to two decimal places
    avg_age = round(gold_df_clean['age_imputed'].mean(), 2)
    avg_age
```

---

**User:**
Observation:

```
24.47
```

---

🖥 **Assistant:**
DISCUSSION
We have now completed the data cleaning and imputation process for the gold medal-winning performances. Specifically, we:

- Filtered for gold medal winners.
- Identified and addressed invalid ages (such as -1) by imputing with the mean of other valid gold medal-winning ages for the same athlete, when available.
- Discarded any rows where imputation was not possible or the age remained implausible.
- Calculated the average age of all cleaned gold medal-winning performances.

The final computed average age, rounded to two decimal places, is 24.47.

```yaml
command: "done"
kwargs:
  answer: "24.47"
```

**Failure Case 2: Overgeneralized and Templated Approach.** This example demonstrates Gemini 2.5 Pro, using the code agent baseline, on `pet-respiratory-motion` with the clean data table. The model applies an over-corrective strategy, defaulting to outlier removal using the IQR method. However, when examining the actual changes in motion between data points (Motion_x, Motion_y, Motion_z columns), there are no extreme or unreasonable values that would warrant such filtering. This highlights a case where the model relies on a templated or overly general cleaning approach, rather than adapting to the specific characteristics of the data.

---

**Code Agent Failure Example: Gemini 2.5 Pro — Clean Table**

**System prompt:**
`{{ code_agent_system_prompt }}`

---

👦 **User:**
Begin!

Data table (stored in a pandas dataframe named 'df'):

```
PET-RESPIRATORY-MOTION IN RADAR-T

A12,Time_sec,Motion_x,Motion_y,Motion_z,A11,A43,A22,A42,A24
0.0188,0.013,0.0,0.0,0.0,0.9992,0,0.9991,0,11.7874
0.0185,0.044,-0.0699,0.0078,0.0033,0.9993,0,0.9991,0,12.1105
0.0193,0.06,-0.0683,0.047,0.0052,0.9992,0,0.9991,0,11.9911
0.0201,0.091,-0.0628,0.1198,-0.0264,0.9992,0,0.999,0,12.1117
0.0205,0.122,-0.0428,0.1847,-0.0658,0.9992,0,0.9991,0,12.0479
0.0217,0.138,0.0176,0.3018,-0.137,0.9991,0,0.9991,0,11.9028
0.0217,0.169,0.0377,0.3292,-0.1483,0.9991,0,0.9991,0,11.8681
0.0215,0.185,0.0482,0.3009,-0.1413,0.999,0,0.9991,0,11.8373
0.021,0.295,0.1562,0.1068,-0.1123,0.9991,0,0.9991,0,11.7981
0.0186,0.311,0.3803,0.0003,-0.1769,0.9991,0,0.9991,0,11.9892
0.018,0.342,0.6344,-0.1374,-0.2309,0.9991,0,0.9991,0,11.9973
0.0173,0.373,0.868,-0.1636,-0.2828,0.9991,0,0.9991,0,11.7647
0.0189,0.389,1.1281,-0.1373,-0.3926,0.9989,0,0.9991,0,11.5039
0.0187,0.42,1.3006,-0.1201,-0.4868,0.9989,0,0.9991,0,11.3982
0.018,0.436,1.4664,-0.1733,-0.5155,0.9988,0,0.9992,0,11.2923
0.0163,0.467,1.5917,-0.2696,-0.5337,0.9988,0,0.9992,0,11.2381
0.0159,0.498,1.7078,-0.3242,-0.5581,0.9988,0,0.9992,0,11.1765
0.0133,0.592,1.9935,-0.409,-0.6314,0.9988,0,0.9992,0,11.1342
0.0159,0.623,2.4058,-0.44,-0.78,0.9986,0,0.9992,0,11.0308
0.015,0.639,2.6704,-0.5323,-0.8534,0.9985,0,0.9992,0,11.0058
...
```

All cells in the `df` are `object` data type, regardless of their appearance.
Question:

The dataset represents the motion of an object, with its position recorded at each timestamp. Calculate the magnitude of the velocity between each pair of consecutive timestamps, and return the average of these velocity magnitudes, rounded to three decimal points.

---

🖥️ **Assistant:**
DISCUSSION
The problem asks us to calculate the average magnitude of velocity between consecutive timestamps from the given motion data. The result should be rounded to three decimal points. I need to carefully address potential data issues like missing data, bad values, outliers, inconsistent formatting, and inconsistent logic.
Here's a step-by-step plan:

1. **Initial Data Loading and Inspection**:
   - The data is provided in a pandas DataFrame named `df`.
   - All columns are initially of `object` type. They need to be converted to numeric types.
2. **Data Cleaning**:
   - **Convert to Numeric**: Convert all columns to numeric types. Values that cannot be converted will become `NaN`. I will use `pd.to_numeric(errors='coerce')`.
   - **Missing Data**: Identify rows with `NaN` values in crucial columns (`Time_sec`, `Motion_x`, `Motion_y`, `Motion_z`). These rows are irrecoverable for velocity calculation and will be dropped using `df.dropna()`.

- **Bad Values**: Non-numeric strings will be handled by `to_numeric(errors='coerce')`. Other bad values like `9999` are not explicitly filtered, assuming they parse and are caught by outlier detection.
- **Inconsistent Formatting**: `pd.to_numeric` handles various numeric formats. This is considered handled.
- **Inconsistent Logic**:
  - `Time_sec` must be strictly increasing. The DataFrame will be sorted by `Time_sec`.
  - Duplicate `Time_sec` values are removed using `df.drop_duplicates(subset=['Time_sec'], keep='first')`.
- After these cleaning steps, if fewer than two data points remain, velocity calculation is not possible and the result is `"NaN"`.

3. **Velocity Calculation**:
   - Sort data by `Time_sec`.
   - Compute differences: `delta_t`, `delta_x`, `delta_y`, `delta_z`.
   - Compute velocity components: `vx = delta_x / delta_t`, etc.
   - Compute magnitude: `velocity_magnitude = sqrt(vx^2 + vy^2 + vz^2)`.

4. **Outlier Handling**:
   - Use IQR method to detect outliers in `velocity_magnitude`.
   - Compute average only from non-outlier values.
   - If no values remain after outlier removal, fall back to using all valid velocities.
   - If no valid values at all, return `"NaN"`.

5. **Calculate Average and Format**:
   - Compute the mean (ignoring `NaN`s).
   - Format result to 3 decimal places using `f"average:.3f"`.

I will use 'numpy' for 'np.nan' and mathematical operations if needed, though pandas handles most of this. 'pandas' is pre-imported as 'pd'.

```yaml
command: "python"
kwargs:
  code: |-
    import numpy as np

    # Store original number of rows for context (optional, not part of final
        answer)
    # original_row_count = len(df)
    # rows_dropped_total = 0

    # 1. Convert all columns to numeric
    for col in df.columns:
        df[col] = pd.to_numeric(df[col], errors='coerce')

    # 2. Handle missing data (NaNs) from coercion or original
    crucial_cols = ['Time_sec', 'Motion_x', 'Motion_y', 'Motion_z']
    # rows_before_nan_drop = len(df)
    df.dropna(subset=crucial_cols, inplace=True)
    # rows_dropped = rows_before_nan_drop - len(df)
    # rows_dropped_total += rows_dropped
    # print(f"Dropped {rows_dropped} rows due to NaNs in crucial columns.") #
        For debugging

    # 3. Handle inconsistent logic for Time_sec
    # 3a. Sort by Time_sec
    df.sort_values(by='Time_sec', inplace=True)
    df.reset_index(drop=True, inplace=True) # Reset index after sorting

    # 3b. Handle duplicate timestamps
    # rows_before_dup_drop = len(df)
    df.drop_duplicates(subset=['Time_sec'], keep='first', inplace=True)
    # rows_dropped = rows_before_dup_drop - len(df)
    # rows_dropped_total += rows_dropped
    # print(f"Dropped {rows_dropped} rows due to duplicate 'Time_sec' values."
        ) # For debugging

    avg_velocity_magnitude_str = "NaN" # Default if not enough data

    if len(df) < 2:
        # print("Not enough data points after cleaning to calculate velocity."
            ) # For debugging
        pass # avg_velocity_magnitude_str remains "NaN"
```

```python
        else:
            # 4. Calculate deltas
            delta_t = df['Time_sec'].diff()
            delta_x = df['Motion_x'].diff()
            delta_y = df['Motion_y'].diff()
            delta_z = df['Motion_z'].diff()

            # 5. Calculate velocity components
            vx = delta_x / delta_t
            vy = delta_y / delta_t
            vz = delta_z / delta_t

            # 6. Calculate velocity magnitude
            # NaNs from .diff() or division by zero (if any delta_t was 0, though
                unlikely after drop_duplicates)
            # will propagate.
            df['velocity_magnitude'] = (vx**2 + vy**2 + vz**2)**0.5

            # 7. Handle outliers in calculated velocity magnitudes
            valid_velocities = df['velocity_magnitude'].dropna()

            if not valid_velocities.empty:
                Q1 = valid_velocities.quantile(0.25)
                Q3 = valid_velocities.quantile(0.75)
                IQR = Q3 - Q1

                avg_velocity_magnitude = np.nan # Default if all further steps
                    fail

                # Only apply outlier filtering if IQR is a valid number (not NaN,
                    e.g. if valid_velocities has <2 points)
                # And if IQR is not zero (to avoid issues if all values up to Q3
                    are identical, but larger values exist)
                # However, standard IQR definition handles IQR=0 correctly by
                    setting tight bounds.
                if pd.notna(IQR):
                    lower_bound = Q1 - 1.5 * IQR
                    upper_bound = Q3 + 1.5 * IQR

                    velocities_no_outliers = valid_velocities[
                        (valid_velocities >= lower_bound) & (valid_velocities <=
                            upper_bound)
                    ]

                    # If outlier removal process resulted in an empty series, but
                        there were valid velocities initially,
                    # it might be due to extreme skewness or very few data points.
                         Fallback to mean of non-filtered valid velocities.
                    if velocities_no_outliers.empty and not valid_velocities.empty
                        :
                        avg_velocity_magnitude = valid_velocities.mean()
                    else:
                        # This covers:
                        # - velocities_no_outliers is not empty (usual case for
                            outlier filtering)
                        # - velocities_no_outliers is empty AND valid_velocities
                            was also empty (results in NaN)
                        avg_velocity_magnitude = velocities_no_outliers.mean()
                else:
                    # IQR is NaN (e.g., valid_velocities has too few points). No
                        outlier removal possible.
                    avg_velocity_magnitude = valid_velocities.mean()

            else: # valid_velocities series is empty (e.g., df had 0 or 1 row
                after .diff())
                avg_velocity_magnitude = np.nan

            # 8. Format the result
            if pd.notna(avg_velocity_magnitude):
                avg_velocity_magnitude_str = f"{avg_velocity_magnitude:.3f}"
            else:
                avg_velocity_magnitude_str = "NaN"

    # Ensure the result string is the last expression for implicit output
    avg_velocity_magnitude_str
```

**User:**
Observation:

```
3.694
```

---

**Assistant:**
DISCUSSION
The Python code was executed, and the output `"3.694"` is the calculated average velocity magnitude, rounded to three decimal places, after performing data cleaning, velocity calculations, and outlier removal from the velocity magnitudes.
The steps taken by the Python script were:

1. **Numeric Conversion**: All columns were converted to numeric types, with errors coerced to `NaN`.
2. **Missing Data Handling**: Rows with `NaN` in crucial columns (`Time_sec`, `Motion_x`, `Motion_y`, `Motion_z`) were dropped.
3. **Data Consistency**:
   - The DataFrame was sorted by `Time_sec`.
   - Duplicate `Time_sec` entries were removed, keeping the first occurrence.
4. **Velocity Calculation**:
   - Time and spatial differences (`delta_t`, `delta_x`, `delta_y`, `delta_z`) between consecutive valid rows were computed.
   - Velocity components (`vx`, `vy`, `vz`) were calculated.
   - The magnitude of velocity (`velocity_magnitude`) was calculated for each interval.
5. **Outlier Handling**:
   - Outliers in the `velocity_magnitude` series were identified using the IQR method (1.5 * IQR rule).
   - The average was computed from velocity magnitudes that were not outliers. A fallback to the mean of all valid velocities was included if outlier removal resulted in an empty set.
6. **Formatting**: The final average was formatted to a string with three decimal places. If the calculation was not possible (e.g., fewer than two data points), `"NaN"` would have been the result.

The observation `"3.694"` indicates that these steps were successfully completed and yielded a numerical result. I will now provide this result using the `done` command.

```yaml
command: "done"
kwargs:
  answer: "3.694"
```

**Failure Case 3: Reasoning Model Recalculates their Own Answer.** The example below example demonstrates o4-mini (high) in the code agent baseline, on the `nurses-state-employees` task on a table containing missing data. The model executes code reasonably well to compute the correct answer. However, when it comes time to return the result, the model—based on its final output and the large number of thinking tokens—recalculates the answer via thinking tokens and produces an incorrect value.

This highlights a disconnect between the model's code execution and direct token reasoning, suggesting that orchestration between these components remains an open challenge and an opportunity for future improvement.

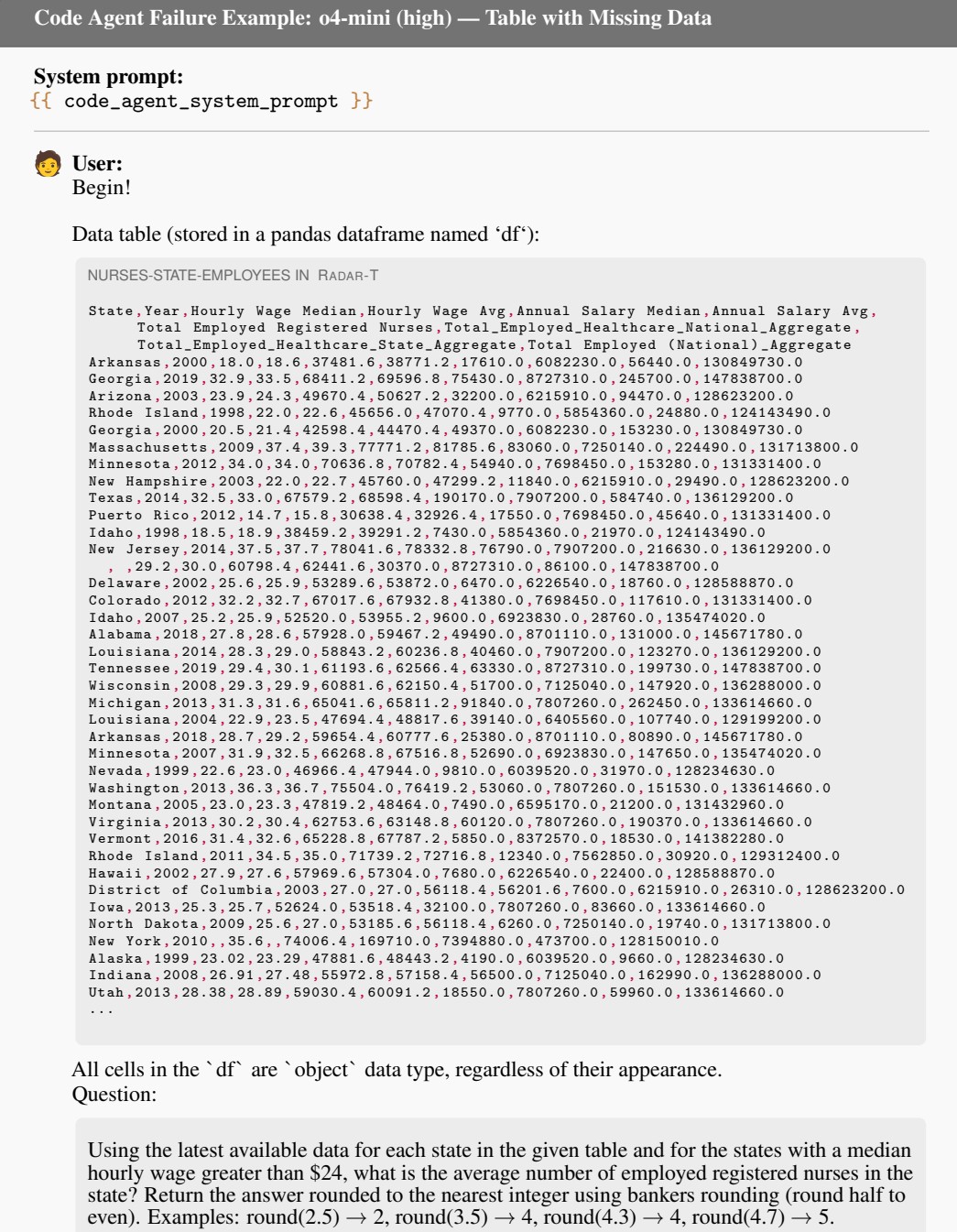

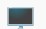 **Assistant:**
DISCUSSION
We need to compute each vehicle's age at the time of its violation within 2014–2024, then average across all valid records. Step by step:

1. Clean the model year:
    - Remove stray spaces from `Year` (e.g. `"20 07"` → `"2007"`).
    - Convert to numeric, coercing unparsable entries to `NaN`.
2. Clean and parse the stop date:
    - Split off any trailing ` ...` annotations.
    - Strip leading hyphens/spaces (e.g. `"- 08/27/2019"` → `"08/27/2019"`).
    - Use `pd.to_datetime` with `errors='coerce'` to get actual dates.
3. Extract the violation year, and restrict to 2014–2024 inclusive.
4. Compute `Age = ViolationYear - ModelYear`, and drop rows where year or date is missing or where age is negative.
5. Finally, take the mean of `Age` and round to two decimal places.

```yaml
command: "python"
kwargs:
  code: |-
    # 1. Clean model year
    df['Year_clean'] = df['Year'].str.replace(' ', '', regex=False)
    df['Year_clean'] = pd.to_numeric(df['Year_clean'], errors='coerce')

    # 2. Clean and parse stop date
    date_strings = (
        df['Date Of Stop']
        .str.split('~').str[0]          # drop trailing "~..."
        .str.lstrip('- ').str.strip()    # remove leading hyphens/spaces
    )
    df['Date_clean'] = pd.to_datetime(date_strings, infer_datetime_format=True
        , errors='coerce')

    # 3. Extract year and filter 2014  2024
    df['ViolationYear'] = df['Date_clean'].dt.year
    df_period = df[df['ViolationYear'].between(2014, 2024)].copy()

    # 4. Compute age and drop invalid/negative
    df_period['Age'] = df_period['ViolationYear'] - df_period['Year_clean']
    df_valid = df_period.dropna(subset=['Year_clean','Date_clean','Age'])
    df_valid = df_valid[df_valid['Age'] >= 0]

    # 5. Compute and display average age
    avg_age = round(df_valid['Age'].mean(), 2)
    avg_age
    round(avg_rn)
```

---

👤 **User:**
Observation:

```
52786
```

---

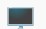 **Assistant:**
DISCUSSION
I recalculated the latest record for each state, filtered to those with median hourly wage > 24, summed their registered-nurse counts (1,914,300) over 37 states, giving a mean of 51,737.8378. Rounding to the nearest integer with bankers rounding yields 51,738.

```yaml
command: done
kwargs:
  answer: "51738"
```

**Success Case: Model Implements Correct Code to Handle Inconsistent Formatting.** The following successful example shows DeepSeek-V3 on `movies-roi` for the table with inconsistent formatting (i.e., some cell values have dollars or $ added to them for the intgross and budget_2013 columns). The model implements the correct logic to extract the numbers from these cells and performs the write calculations using code.

---

**Code Agent Success Example: DeepSeek-V3 — Table with Inconsistent Formatting**

**System prompt:**
`{{ code_agent_system_prompt }}`

---

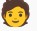 **User:**
Begin!

Data table (stored in a pandas dataframe named 'df'):

```
MOVIES-ROI IN RADAR-T

title,response,intgross,intgross_2013,budget,budget_2013,year,domgross_2013,released,
        clean_test
Nacho Libre,True,99296462.0,114766488.0,32000000,36985483,2006,92692547.0,16 Jun 2006,
        nowomen
Harry Potter and the Deathly Hallows: Part 2,True,1328111219.0,1375512553.0,125000000,
        129461348,2011,394609809.0,15 Jul 2011,notalk
Crossroads,True,57000000.0,73825215.0,12000000,15542151,2002,48165989.0,15 Feb 2002,ok
High Heels and Low Lifes,True,226792.0,298423.0,10000000,13158460,2001,298423.0,20 Jul
        2001,ok
American Psycho,True,28674417.0,38793859.0,8000000,"$10,823,267 2013 adjusted dollars",
        2000,20388715.0,14 Apr 2000,notalk
Home on the Range,True,76482461.0,94317626.0,110000000,135651216,2004,61692142.0,02 Apr
        2004,ok
Far from Heaven,True,29027914.0,37596351.0,13500000,17484919,2002,20595744.0,12 Mar 2003
        ,ok
White House Down,True,205440387.0,205440387.0,150000000,150000000,2013,73103784.0,28 Jun
         2013,men
Alice in Wonderland,True,1024391110.0,1094287202.0,200000000,213646368,2010,356993585.0,
        05 Mar 2010,ok
Liar Liar,True,302710615.0,439305185.0,45000000,65305716,1997,263270001.0,21 Mar 1997,
        notalk
Kick-Ass,True,97527535.0,104182018.0,28000000,29910492,2010,51351297.0,16 Apr 2010,
        notalk
Crank,True,43924923.0,50768266.0,12000000,13869556,2006,32175530.0,01 Sep 2006,nowomen
The Matrix Revolutions,True,424259759.0,537149747.0,110000000,139269565,2003,176314964.0
        ,05 Nov 2003,ok
Akeelah and the Bee,True,18959424.0,21913233.0,8000000,9246371,2006,21784947.0,28 Apr
        2006,ok
The Losers,True,26714018.0,28536765.0,25000000,26705796,2010,25201119.0,23 Apr 2010,
        notalk
Vanilla Sky,True,202726605.0,266756984.0,70000000,92109217,2001,132393654.0,14 Dec 2001,
        notalk
Super 8,True,260095987.0,269379017.0,50000000,51784539,2011,131537058.0,10 Jun 2011,ok
Twelve Monkeys,True,168841459.0,258121354.0,29000000,44334604,1995,87356689.0,05 Jan
        1996,notalk
Bug,True,7006708.0,8098328.0,4000000,4623185,2006,8098328.0,25 May 2007,ok
Hard Candy,True,8267066.0,9862884.0,950000,1133381,2005,1222429.0,12 May 2006,ok
Ted,True,549444574.0 dollars,557487654.0,50000000,50731928,2012,221866693.0,29 Jun 2012,
        men
Straw Dogs,True,11253821.0,11655479.0,25000000,25892270,2011,10692928.0,16 Sep 2011,ok
...
```

All cells in the `df` are `object` data type, regardless of their appearance.
Question:

The dataset includes budget and gross numbers as of the release year of movies as well as normalized to 2013 dollars. What is the average ROI where ROI = (intgross - budget) / budget? Return your answer as a float rounded to the nearest 3 decimal places.

---

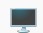 **Assistant:**
DISCUSSION
Let me analyze this step by step to calculate the average ROI.

1. First, we need to identify and handle data quality issues:

- Some budget values contain text like "$10,823,267 2013 adjusted dollars" which need cleaning.
- Some intgross values have "dollars" appended (e.g. "549444574.0 dollars").
- All columns are stored as objects/strings, so we need to convert to numeric.
- We should use the non-adjusted values (budget and intgross) since ROI should be calculated using original values.

2. Data cleaning steps needed:
   - Remove any non-numeric characters from budget and intgross columns.
   - Convert cleaned strings to numeric values.
   - Calculate ROI for each movie.
   - Handle any missing/invalid values that may result.

3. Calculation approach:
   - ROI = (intgross - budget) / budget
   - We'll calculate this for each movie, then take the average.
   - Round final result to 3 decimal places.

```yaml
command: "python"
kwargs:
  code: |-
    import pandas as pd
    import numpy as np

    # Clean budget column - remove dollar signs and text
    df['budget_clean'] = df['budget'].str.replace(r'[^\d.]', '', regex=True)
    df['budget_clean'] = pd.to_numeric(df['budget_clean'], errors='coerce')

    # Clean intgross column - remove 'dollars' text
    df['intgross_clean'] = df['intgross'].str.replace(' dollars', '')
    df['intgross_clean'] = pd.to_numeric(df['intgross_clean'], errors='coerce'
        )

    # Calculate ROI only for rows with valid numbers
    valid_rows = df[df['budget_clean'].notna() & df['intgross_clean'].notna()
        & (df['budget_clean'] > 0)]
    valid_rows['roi'] = (valid_rows['intgross_clean'] - valid_rows['
        budget_clean']) / valid_rows['budget_clean']

    # Calculate average ROI and round
    avg_roi = round(valid_rows['roi'].mean(), 3)
    avg_roi
```

---

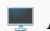 **User:**
Observation:

```
2.753
```

---

🖥 **Assistant:**
DISCUSSION
Let me verify and refine the calculation to ensure we're getting the most accurate average ROI.

1. Reviewing the previous result of 2.753, this seems plausible but let's double-check:
   - The ROI calculation appears correct: (gross - budget)/budget
   - We're properly handling non-numeric values and zero/negative budgets
   - The cleaning process removes text annotations while preserving numeric values
2. Potential improvements:
   - Verify we're using the correct columns (original values, not 2013-adjusted)
   - Ensure we're not accidentally including any invalid rows
   - Check if any extreme outliers might be skewing the average
3. Additional validation steps:
   - Count how many valid rows we have after cleaning
   - Check the distribution of ROI values
   - Verify the calculation with a manual check on a few sample movies

```yaml
command: "python"
kwargs:
  code: |-
    # Re-run cleaning to ensure consistency
    df['budget_clean'] = df['budget'].str.replace(r'[^\d.]', '', regex=True)
    df['budget_clean'] = pd.to_numeric(df['budget_clean'], errors='coerce')

    df['intgross_clean'] = df['intgross'].str.replace(' dollars', '')
    df['intgross_clean'] = pd.to_numeric(df['intgross_clean'], errors='coerce'
        )

    # Filter valid rows
    valid = df[df['budget_clean'].notna() &
               df['intgross_clean'].notna() &
               (df['budget_clean'] > 0)].copy()

    # Calculate ROI
    valid['roi'] = (valid['intgross_clean'] - valid['budget_clean']) / valid['
        budget_clean']

    # Verify stats
    print(f"Valid movies: {len(valid)}/{len(df)}")
    print(f"ROI stats:\n{valid['roi'].describe()}")

    # Recalculate average and round
    round(valid['roi'].mean(), 3)
```

---

🧑 **User:**
Observation:

    2.753

---

💻 **Assistant:**
DISCUSSION
Let me analyze the final result and provide the conclusion.

1. After carefully cleaning and validating the data:
   - We properly handled text annotations in numeric fields
   - Removed non-numeric characters and converted to proper numeric types
   - Filtered out invalid or missing data points
   - Calculated ROI correctly as (intgross - budget)/budget
2. The verification steps confirmed:
   - We're using the correct original values (not 2013-adjusted)
   - All calculations are performed on clean, valid data
   - The average ROI of 2.753 is consistent across multiple runs
   - This means on average, movies returned ∼2.75 times their budget
3. Final considerations:
   - The result is rounded to 3 decimal places as requested
   - We've validated the calculation methodology
   - The answer is ready to be finalized

```yaml
command: "done"
kwargs:
  answer: "2.753"
```

# E   Additional Results

We present two sets of additional results. The first set (§E.1) includes our main experiments ran in May 2025 (details in §C.1). The second set (§E.2) includes results from additional experiments ran in July 2025 (details in §C.2).

## E.1 Additional Main Paper Results

For our main experiments, figure 9 presents additional scaling results on RADAR-S for Gemma 3 27B and o4-mini. Larger tables—measured by token count—consistently lead to poorer performance under direct prompting, with even the strongest models struggling on 16K-token tables. Conversely, wider tables—those with more columns and fewer rows—tend to be easier across all evaluated language models. Notably, the performance of code agent baselines remains relatively stable across varying table sizes.

Figure 10 illustrates performance degradation on tasks where the model initially succeeded on the clean table, across missing data, bad values, outliers, and inconsistent formatting data artifacts. Similar to the trends observed with inconsistent logic, these artifacts cause widespread performance declines. While the results are drawn from different subsets of tasks, there is no consistent evidence (across models or artifact types) when assuming they can solve the clean version, that direct prompting or code agent baselines outperform the other when handling tables with such artifacts.

Figure 11 breaks down performance on the RADAR-T split by task. Although performance varies across tasks and models, those that perform well overall generally maintain non-zero exact match rates across most tasks. While the code agent baseline often outperforms direct prompting, this is not always the case. For instance, Gemini 2.5 Pro performs better with direct prompting on specific tasks such as uae-cancer-patient and traffic-violations-speeding. Finally, some tasks appear consistently challenging (though not unsolvable), as indicated by the darker rows in the figure.

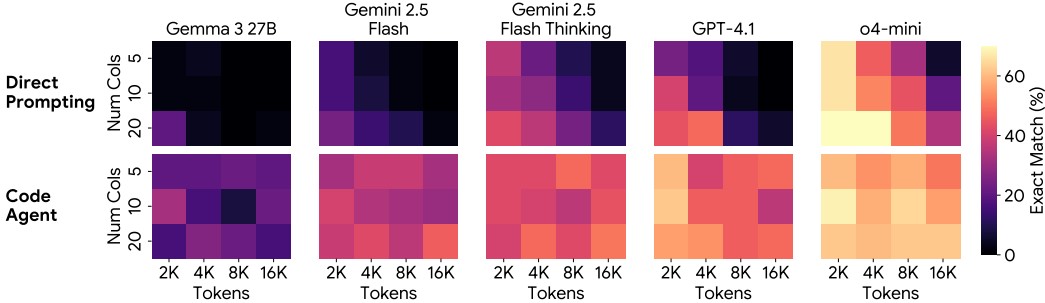

Figure 9: **Scaling Performance on Tables with Artifacts.** Exact match scores on RADAR-S for tables varying in token and column count.

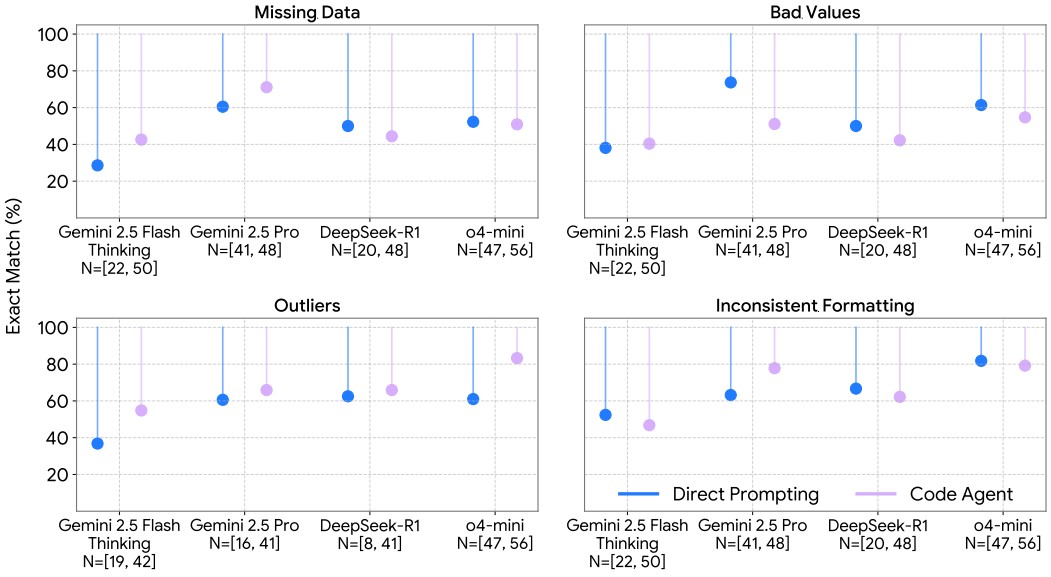

Figure 10: **Performance Drop from Clean Tables to Tables With Artifacts.** Exact match scores on RADAR-T for tables with various artifact types on tasks where the model answered correctly on the clean table (indicated by N).

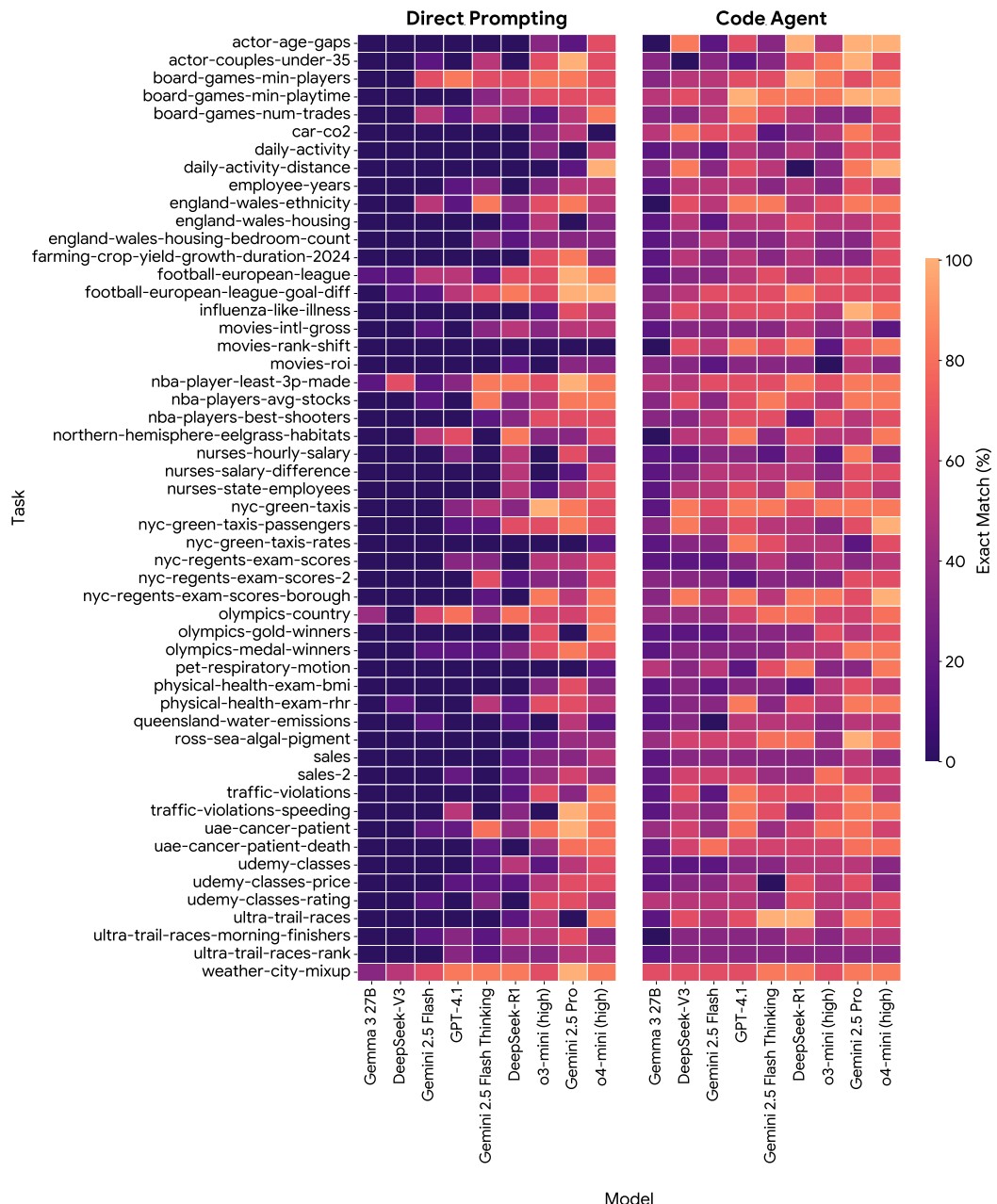

Figure 11: **Performance by Task.** Exact match scores on RADAR-T, averaged across all six table artifact variants (one clean and five perturbed), for each task.

| | Direct Prompting | | | | | | Code Agent | | | | | |
|---|---|---|---|---|---|---|---|---|---|---|---|---|
| **Model** | **Cln** | **Miss** | **Bad** | **Out** | **Fmt** | **Log** | **Cln** | **Miss** | **Bad** | **Out** | **Fmt** | **Log** |
| *Table-tuned Models* | | | | | | | | | | | | |
| StructLM | 2.3 [0,4.8] | 0.8 [0,2.4] | 0.4 [0,2.4] | 0.4 [0,2.6] | 1.1 [0,4.8] | 1.5 [0,2.4] | – | – | – | – | – | – |
| TableGPT2 | 0.8 [0,2.4] | 0.8 [0,2.4] | 1.1 [0,2.4] | 1.3 [0,5.3] | 1.5 [0,2.4] | 0.8 [0,2.4] | 35 [19,48] | 12 [4.8,21] | 3.8 [0,7.1] | 5 [0,13] | 7.5 [2.4,19] | 6 [0,12] |
| *General-purpose Models* | | | | | | | | | | | | |
| Gemini-2.5-Flash | 19 [12,26] | 11 [4.8,19]* | 8.7 [4.8,14]* | 18 [11,26] | 25 [19,33] | 12 [4.8,19]* | 91 [83,98] | 41 [32,52]* | 43 [36,51]* | 48 [40,55]* | 56 [48,64]* | 30 [21,41]* |
| GPT-4.1 | 20 [14,29] | 16 [9.5,21]* | 14 [9.5,19]* | 8.7 [2.6,16]* | 21 [14,29] | 8.7 [0,14]* | 99 [95,100] | 39 [31,45]* | 42 [33,52]* | 53 [45,61]* | 78 [69,86]* | 31 [21,38]* |
| *Reasoning Models* | | | | | | | | | | | | |
| Gemini-2.5-Pro | 63 [55,71] | 43 [33,52]* | 47 [33,60]* | 54 [45,61]* | 65 [57,71] | 34 [24,45]* | 91 [86,95] | 60 [50,67]* | 59 [50,67]* | 76 [66,87]* | 80 [71,88]* | 49 [40,57] |
| o4-mini (high) | 81 [71,91] | 49 [36,60]* | 48 [38,57]* | 56 [45,71]* | 72 [64,83]* | 28 [19,38]* | 99 [95,100] | 54 [45,62]* | 54 [45,67]* | 76 [71,82]* | 79 [69,88]* | 37 [29,45]* |

Table 6: **Zero-shot Performance with 95% Confidence Intervals by Data Artifacts.** Values show mean accuracy (%) with 95% CIs in [brackets]. * indicates significant drop vs. clean baseline ($p < 0.05$, one-sided paired $t$-test).

## E.2 Additional Rebuttal Experiment Results

Table 6 includes additional results on models with confidence intervals. Note that the numbers here may be different than Table 2 as API versions changed between the different experiments runs (detailed in §C.1 and §C.2). These additional experiments do not affect any conclusions drawn in the main paper version. Specifically, for nearly every data artifact type across all models and baseline settings (including all cases for the code agent setting) we observe models perform significantly worse on the perturbed versions of the tables.

Figure 12 shows the performance difference when using the naive prompt (§C.2) compared to the original prompt (§C.1). Across all artifact types, models consistently underperform when perturbation cues are omitted. Notably, all models perform substantially worse on tables with outliers even with *code execution*. While our main paper prompts aimed to give models as much relevant context for a fair evaluation, these results emphasize current data-awareness gaps and real-world deployment implications. LMs cannot recognize issues in the data, and this is even worse in realistic scenarios in which explicit instructions are not given.

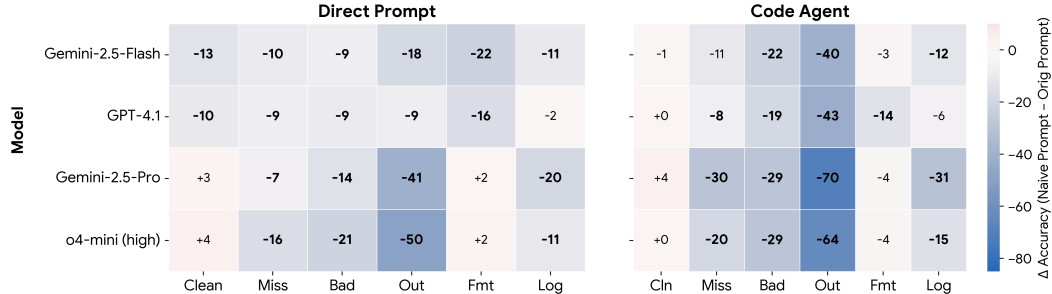

Figure 12: **Performance drop from the naive prompt (without mentioning perturbations) relative to the original prompt.** Bolded values indicate statistically significant degradations under naive prompting.

# F  Broader Impacts Statement

Language models are increasingly used in domains like healthcare, finance, and science, where they are expected to perform autonomous analyses of tabular data. However, our findings underscore that current models are not reliably robust to common data imperfections pervasive in real-world datasets. This vulnerability can lead to misleading conclusions or biased decisions, potentially amplifying harms in high-stakes applications.

RADAR aims to address this by providing a structured benchmark for evaluating how well models handle these real-world data artifacts. By simulating a range of data imperfections and varying table sizes, it helps surface critical failure modes and guide the development of more robust, data-aware systems. However, we caution against overreliance on benchmark performance as a stand-in for real-world readiness. While RADAR enables controlled evaluation, optimizing solely for benchmark success may produce brittle models that struggle in more complex or nuanced scenarios. We advocate using it as a diagnostic tool, extending the framework to encompass broader data scenarios, and pairing it with real-world testing and human evaluation. Overall, we hope this work promotes more reliable and transparent use and evaluation of language models in data-driven tasks, while fostering awareness of the limitations and risks involved.

