# OpenReview forum: "RADAR: Benchmarking Language Models on Imperfect Tabular Data"
_NeurIPS.cc/2025/Datasets_and_Benchmarks_Track — NeurIPS 2025 Datasets and Benchmarks Track poster_

### Official Review · Reviewer_cpPp · 2025-06-28

**Rating:** 5
**Confidence:** 4

**Summary:**

The paper introduces a data-set to assess LLM performance on data analysis of malformed tables. The data set collects tabular data from existing data (my interpretation is that the table curation is done by domain experts), introduces a mechanism to deterministically perturb tables and introduce an "error" from a set of error categories (e.g. missing cells, logical inconsistencies w/ other columns, etc.). The observation is that SOTA LLMs experience a serious drop in performance when operating on malformed tables vs clean well-structured tables.

There are additional results (i) code agents (as opposed to vanilla foundation models and with reasoning) being more robust to attributes like table size, and (ii) corrections not matching the schema.

The appendix provides failure analysis.

**Additional Feedback:**

Code to actually execute the perturbations might be useful. The "perturbation_spec" fields seem unambiguous and thus I am not going to allow this to impact the review.

**Dataset Code Accessibility:**

Yes

**Dataset Code Comments:**

Opened the kaggle link - the data is well organized and it is straightforward to access it and write readers.

**Ethical Comments:**

The data set is based on existing public data sets (described in the appendix, and appropriately attributed in the references) - the tasks themselves are fairly straightforward and no different to standard analysis that is conducted on such data within organizations or during standard workflows.

**Ethical Considerations:**

No, there are no or only very minor ethics concerns

**Final Justification:**

It is a strong submission - looks like all the other reviewers concur. Authors have done a good job addressing my concerns, and have done a good job addressing other reviewers' comments as well. There is a rich set of future work possible in this area and the work is a good fit for the track. I am going to keep my score.

**Limitations Weaknesses:**

No real limitations. The work is a good fit for the track. I have some code-related comments below (and thus they are not limitations with the paper).

**Strengths Contributions:**

The data set is a strong contribution due to (i) broad domain coverage, (ii) human domain expertise, (iii) focus on real world data analysis issues (malformed tables). The data set is well motivated, the description is easy to follow, provides us with insights on how to use foundation models for data analysis. The supplemental appendix contains useful error analysis of the failure cases. Tabular data analysis is commonplace in many domains, and this data set is timely in assessing where frontier models fall short.

The perturbation technique was easy to follow and probably easy to adapt for future work in this area.

Overall, a good fit for the track.

---

> ### Author Rebuttal · Authors · 2025-07-31
>
> Thank you for your supportive comments and acknowledging the broad domain expertise involved in constructing RADAR’s tasks!  The  **tables and tasks were indeed curated by domain experts** to reflect realistic and meaningful data artifacts across a wide range of fields.
>
> Regarding the code used to execute the perturbations, this is included in our submission’s supplementary materials. Specifically, the file `radar/data/perturb.py` contains the logic that applies the `perturbation_spec` to the clean data tables.
>
>
> If your comment refers to the full set of functions used to generate the `perturbation_spec` definitions themselves, we will open-source those as well. However, per NeurIPS rebuttal policy, we are not permitted to share external links during the review process.
>
> That said, the current supplementary materials do include an illustrative example of how perturbations and ground truth answers are defined (see `radar/tasks/funcs/influenza_like_illness.py`). We hope this gives reviewers a concrete sense of the task construction framework and our dataset generation methodology.
>
> We hope this answers your question, and please let us know if there is anything we could do to further improve our score.

---

### Official Review · Reviewer_kvSF · 2025-07-03

**Rating:** 5
**Confidence:** 3

**Summary:**

The authors present RADAR, which is a benchmark to assess language models on data awareness tabular reasoning. They crowdsourced 53 datasets and queries from domains like education, health, and business and implemented 260 query-specific perturbation functions to inject realistic artifacts. These data artifacts include missing data, bad values, outliers, inconsistent formatting, and inconsistent logic. With this benchmark, the authors experiment various general purpose and reasoning models to reveal their respective strengths and limitations on their data awareness.

**Dataset Code Accessibility:**

Yes

**Dataset Code Comments:**

* The dataset can be found here: https://www.kaggle.com/datasets/0757b931079767a72b0c08f2225e658419debccc437dbd98af6b76d3eaca183e
* Additional information on dataset creation is also provided in the supplementary material attached in the submission.

**Ethical Considerations:**

No, there are no or only very minor ethics concerns

**Final Justification:**

The authors have given case-by-case examples for different categories I was concerned about (for instance, bad input) and has provided explicit information on what they will do in their next version to clarify this better for future readers.

**Limitations Weaknesses:**

While I am inclined to accept this paper due to its overall contributions, I have currently marked it as a borderline reject due to concerns with two of the proposed data artifacts: bad values and inconsistent logic.

In particular, Table 2 shows that performance drops across models by at least half for these artifact types. This significant degradation raises the question of whether these inputs are truly "bad" in a meaningful sense, or simply out-of-distribution or ambiguous where the model is doing the best it can given the context. For example, when a model is given inconsistent or incorrect context or data, is it fair to penalize it for generating what may appear to be an incorrect response when in fact it might be "data aware" in the sense that it is simply responding appropriately to the context it was given?

 I think the paper might benefit from providing a discussion around this boundary, i.e., when does the data become so broken that we no longer expect the model to get the "right" answer? This discussion could be supported by examples or qualitative analysis, which would help clarify the distinction and strengthen the justification for these artifacts as fair and meaningful benchmark cases.

Minor note:
* There is a typo with the starting quote in "clean" on line 56.

**Strengths Contributions:**

* The paper is well written with justified motivation.
* Table 2, in particular, is useful to show the extent to which the model performance degrades with every data artifact.
* The dataset on kaggle is well-organized.

---

> ### Author Rebuttal · Authors · 2025-07-31
>
> Thank you for your recognition of RADAR's important motivation, our efforts in preparing an organized dataset, and our overall contributions. We agree that distinguishing between truly “bad” data and ambiguous or out-of-distribution inputs is crucial for evaluating robustness. Consequently, these were **important considerations in our task design and development process from the beginning**. Below, we address three interrelated aspects of your review and explain how we will incorporate these clarifications in the camera-ready version.
>
>  &nbsp;
>
> **1. What counts as “bad” inputs in RADAR?**
> >  This significant degradation raises the question of whether these inputs are truly "bad" in a meaningful sense, or simply out-of-distribution or ambiguous where the model is doing the best it can given the context.
>
> From a data science perspective, a bad input is one that, if left unaddressed, would compromise the validity of a calculation or the conclusions drawn from the data. Practically, it is a case where multiple data scientists would unanimously agree that the data artifact must be addressed because it contradicts internal or commonsense logic, or clearly violates expectations (e.g., based on established domain knowledge). As a result, to ensure data artifacts were objectively erroneous, unambiguous, and solvable, all tasks underwent multiple rounds of code review and refinement.
>
> For example:
> * In `actor-age-gaps`, an age gap of 56 years between actor couples might be debated as an outlier. One of our expert annotators actually pointed this out: "*Is this enough? I mean you could have a 80 years old male actor like Morgan Freeman or Ian McKellen, and 18 female actor*." However, an age gap of 86 years would be universally seen as erroneous and would be removed or flagged by any reasonable practitioner.
>
> Similarly, for inconsistent logic artifacts, we scoped perturbations which broke relationships between columns that were common sense and representable by clear equations (e.g., $\mathrm{start\\_time} < \mathrm{end\\_time}$, $\mathrm{bmi} = \mathrm{weight} / \mathrm{height}^2$). We explicitly avoided more ambiguous logic that lacks a well-defined formulaic relationship (e.g., a heavy package with a low shipping fee when other lighter packages in the dataset have a higher shipping fee), since these could lead to subjective interpretations.
>
> Examples:
> * In `nyc-green-taxis-passengers`, we perturb rows so that $\mathrm{dropoff\\_time} < \mathrm{pickup\\_time}$, clearly violating temporal order.
> * In `employee-years`, we set $\mathrm{YearsAtCompany > TotalWorkingYears}$, which contradicts the schema logic.
> * In `uae-cancer-patient`, $\mathrm{diagnosis\\_date} > \mathrm{treatment\\_start\\_date}$ violates a real-world clinical timeline.
> * In `ultra-trail-races-morning-finishers`: finish time is altered by +12 hours, making it inconsistent with the known start time and derived duration column.
>
> &nbsp;
>
> **2. How do we avoid unfairly penalizing “data-aware” models?**
> >  ...when a model is given inconsistent or incorrect context or data, is it fair to penalize it for generating what may appear to be an incorrect response when in fact it might be "data aware" in the sense that it is simply responding appropriately to the context it was given?
>
> One way we avoid penalizing data-ware models is to not only define **unambiguous queries** but also introduce **objectively erroneous data artifacts**. This approach is paired with carefully designed perturbations, ensuring that the **corrective actions are narrowly scoped** to overwriting cells or removing rows. During task construction, we iteratively refined the query or perturbation definition to satisfy these goals.
>
> For example:
> * In `nurses-hourly-salary`, we originally expected models to infer hourly wage from annual salary using the formula $\mathrm{hourly\\_wage} = \mathrm{annual\\_salary} / (40 \mathrm{hours} / \mathrm{week} \cdot 52 \mathrm{weeks} / \mathrm{year})$ which was consistent in the table.
> * An annotator reasonably questioned whether this conversion could be assumed: "*Don't some people work overtime sometimes etc? I don't know if I can confidently recover this as a human.*"
> * Based on this, we revised the query to explicitly state that wages assume a standard full-time schedule, starting with: *The dataset contains nurses' wages assuming a standard full-time work schedule (i.e., 40 hours per week and 52 weeks per year).*
>
> All our considerations **ensure** that models are not being penalized for drawing conclusions from ambiguous or underspecified inputs.
>
> In addition, we reiterate the expected behavior in the evaluation prompts themselves, so that models are guided toward appropriate corrective actions: “*Attempt to safely recover or correct flawed data when reasonable based on the existing data. If data is irrecoverable or suspect, discard the row.*”
>
> &nbsp;
>
> **3. Where is the boundary between solvable and irrecoverably broken data?**
> > I think the paper might benefit from providing a discussion around this boundary, i.e., when does the data become so broken that we no longer expect the model to get the "right" answer?
>
> The exact boundary between potentially recoverable and irrecoverably broken data is **likely ambiguous** and **somewhat subjective**. However, RADAR takes **multiple**, **intentional** steps to exclusively focus on well-defined tasks. The first of which is to define unambiguously bad perturbations with clear corrective actions discussed above. In addition we make sure:
> * Perturbations were applied to a very small subset of rows  (i.e., < 5% in most cases).
> * Only one perturbation type was applied per task instance as mixing perturbation types can make the unambiguous corrective action hard to define.
> * Instructing annotators to be vigilant for tasks that are underspecified, ambiguous, or out of scope of the expected corrective actions.
>
> While one could reasonably argue that more significant perturbations and combinations thereof are reasonable to evaluate, we calibrated our benchmark dataset in this way, because this level of perturbations already leads to low model performance. While our benchmark dataset could be trivially expanded to include more challenging perturbations, we posit that focusing on simpler cases that clearly fail will help our community meaningfully evaluate progress step-by-step.
>
> &nbsp;
>
> **Summary and concrete updates to the camera-ready version**
>
> We agree that objective, meaningful evaluation requires perturbations that are unambiguously wrong, while ensuring that the remaining data supports reliable inference of schema and relationships. As a result, these considerations fundamentally guided our task and benchmark design.
>
> To emphasize RADAR’s value as a rigorous and interpretable benchmark, we plan to make the following concrete updates to the **camera-ready version:**
>
> * A new section after Section 3.1, clearly defining what are “bad” perturbations in RADAR and how the perturbations we want are objective and scoped to clear corrective actions. We will add illustrative examples of (1) clearly bad perturbations (e.g., $\mathrm{dropoff\\_time} \< \mathrm{pickup\\_time}$), (2) explicitly recoverable values (e.g., missing annual wages column that could be recovered based on a consistent formula involving the hourly wage column), and (3) outliers that cross the threshold into universally unacceptable territory (e.g., a 86-year age gap).
>  * In the new section, a paragraph discussing the boundary between solvable and irrecoverably broken data, outlining how we ensure tasks remain solvable by limiting perturbations to a small subset of rows, a single perturbation type, and exclusion of ambiguous inconsistencies.
> * Details in the current Section 3.2 to emphasize the iterative process in constructing RADAR to ensure unambiguous queries and objectively erroneous data artifacts in order to avoid inaccurately penalizing models for reasonable behavior. This process included making assumptions explicit in the query (e.g., the revised nurse wage query in `nurses-hourly-salary`) or updating perturbation functions (e.g., updated outlier for the `actor-age-gaps`).

---

> > ### Author Response · Authors · 2025-08-04
> >
> > Hi kvSF,
> >
> > We just wanted to follow-up on our rebuttal to see if you have had a chance to read over our response. We are eager to hear your thoughts and would also appreciate it if you considered updating your review and score accordingly. We realize this is a tight window for discussion and appreciate your time!

---

> > ### Comment · Reviewer_kvSF · 2025-08-05
> > **Rebuttal Response**
> >
> > Thank you for the detailed response. I appreciate the quotes and examples in each case to better understand how the benchmark was scoped. I'll increase my score.

---

> > > ### Author Response · Authors · 2025-08-05
> > >
> > > Thank you for the thoughtful review and for taking the time to consider our response. We appreciate your constructive feedback, which helped us clarify these important aspects of our work.

---

### Official Review · Reviewer_YeYx · 2025-07-03

**Ethics Flags:** Improper research involving human sub…
**Rating:** 5
**Confidence:** 4

**Summary:**

The paper presents the RADAR benchmark, which is a tabular reasoning benchmark that enables studying model robustness for tabular reasoning in the presence of corrupt tabular data inputs such as missing values or outliers. The benchmark studies various LLMs for their capability to answer questions correctly in the presence of data errors, when instructed that issues might be present in the data, showing that LLMs are sensitive for data issues in tabular reasoning which is most effectively, but not entirely, resolved by tool integration. The benchmark is reasonably constructed but has some shortcomings with respect to the (clear) specification of desired behavior (e.g. in the presence of missing values), and it remains unclear why a new benchmark for robustness is needed while prior works have surfaced the same insights by adapting existing datasets.

**Dataset Code Accessibility:**

No

**Dataset Code Comments:**

The dataset is only published through Kaggle, but is not supported by the code to, for example, generate ground-truth answers or conduct the analysis.

**Ethical Comments:**

No details are provided about how the data science experts were recruited (e.g. a crowdsourcing platform?) and compensated.

**Ethical Considerations:**

Yes, there are ethics concerns that require attention by the authors

**Final Justification:**

In response to the rebuttal and particular the sharpened positioning w.r.t. prior studies/benchmarks on the same problem, task clarifications, and additional experimental insights, I am updating my review score from 3 to 5.

**Limitations Weaknesses:**

1) The need for a new benchmark to study the robustness of tabular reasoning, versus adapting existing benchmarks such as TQA Bench [1] as done in prior studies assessing robustness against corruptions like missing values [2,3], is unclear. Some of the insights had already surfaced in such prior studies.

2) The desired reasoning behavior for models to handle the corruptions in the data is questionable – the ground-truth answer generation assumes that there is a function that undoes the perturbation and outputs an answer, e.g. by removing perturbed rows. But what is the desired reasoning behavior of an LLM in the presence of, for example, missing values, when calculating  a sum or average?

3) It is unclear what type of operations are posed in the queries, e.g. summation, averaging, etc. or whether these are basic lookups. It would be interesting to understand the performance variance on task/operation-level.

4) It would have been interesting to understand if LLMs natively recognize issues in the data, instead of explicitly instructing them to attend and handle such issues. When people use LLMs for question answering over tabular data, they might not be aware of issues nor instruct models to handle them.

5) Various table-aware LLMs have been developed (e.g. TableGPT2, StructLM), and it would be interesting to see if such table-tuned LLMs recognize and handle data issues better rather than out-of-the-box LLMs.

[1] TQA-Bench: Evaluating LLMs for Multi-Table Question Answering with Scalable Context and Symbolic Extension, Zipeng Qiu, et al, 2024.

[2] Tabular Representation, Noisy Operators, and Impacts on Table Structure Understanding Tasks in LLMs, Ananya Singha, et al., Table Representation Learning workshop at NeurIPS 2023.

[3] How well do LLMs reason over tabular data, really?, Cornelius Wolff, et al., Table Representation Learning workshop at ACL 2025.

**Strengths Contributions:**

1) The benchmark assesses the robustness of LLMs against various noise in the tabular inputs, which is a relevant problem to study.

2) The construction of ground-truth answers by sourcing data cleaning functions to remedy the perturbations is a valuable contribution.

3) The analysis studies the effectiveness of LLMs to handle the data perturbations when instructions are provided in the prompt versus enabling function calls.

---

> ### Author Rebuttal · Authors · 2025-07-31
>
> We appreciate your recognition of the relevance and motivation behind our problem setting and the value of our dataset-agnostic framework. Based on your suggestions, we added substantial new experiments and analysis on RADAR-T involving (1) prompts that omit explicit mention of perturbations and (2) table-tuned LLMs. Our updated results reiterate the challenge posed by domain-specific common-sense reasoning artifacts introduced in RADAR. All experiments report bootstrapped confidence intervals across 5 model runs (see our response to 2KvL).
>
>  &nbsp;
>
> > *The need for a new benchmark to study the robustness of tabular reasoning... is unclear.*
>
> We appreciate this important point. We acknowledge that prior benchmarks have studied the robustness of tabular reasoning with respect to structural perturbations (e.g., all perturbation types in [2] and one of three in [3]). Structural perturbations (i.e., shuffling rows, merging columns), however, do not require models to understand semantics of the table.
>
> In contrast, grounded in data analysis, RADAR focuses on **perturbations that demand semantic and schema-level understanding**, such as:
> * Rows where a New York City borough column is mismatched with the borough identifier column (`nyc-regents-exam-scores-borough`)
> * Rows where number of users trading a given board game should be less than the number of users owning the board game (`board-games-num-trades`)
> * See our other examples in our response to kvSF.
>
> These are **non-trival**, require domain-specific reasoning, understanding multi-column relationships, and are underexplored in existing benchmarks.
>
> [3] includes a "missing values" perturbation, but its operationalization is different from that of RADAR. [3] considers the answer correct when setting the missing value to 0. In contrast, RADAR considers multiple situations in which missing data compromises the calculation (see below in response).
> Because of the **non-triviality** of RADAR’s perturbations, we took a crowd-sourced expert curation approach, collecting data tables from domains where contributors had firsthand familiarity. Perturbations were introduced only when the when the corrective action was clear to multiple experts. We will clarify this in Section 3.2 and Appendix A.
>
> We will also **refine the framing of RADAR** (Sec. 1 and 2) to better emphasize its role as a necessary, complementary benchmark, expanding beyond structural perturbations to include semantically-grounded, table-informed artifacts.
>
>  &nbsp;
>
> > *Some of the insights had already surfaced in such prior studies.*
>
>
> While code execution benefits are known, **RADAR reveals novel failure modes even with code execution** (e.g., Failure Cases 1 & 2 in the Appendix; lines 56, 62). Figure 5 shows persistent errors on logical inconsistencies despite code execution. In table scaling experiments, we quantitatively show how LMs handle wider tables more effectively under direct prompting (Figure 6).
>
> Table-tuned models also perform poorly on RADAR (see below), further validating that our benchmark captures robustness challenges not addressed by prior models and work.
>
>
>  &nbsp;
>
>
> > *ground-truth answer generation assumes that there is a function that undoes the perturbation…what is the desired reasoning behavior of an LLM in the presence of, for example, missing values, when calculating a sum or average?*
>
> Designing for clarity in expected behavior was a **core goal**. Each query-perturbation pair was reviewed by multiple experts to ensure **unambiguous recovery or discard behavior**. For example, for `actor-age-gaps,` an early outlier logic adding 56 to an actor age gap (`df_perturbed.loc[inds, "age_difference"] + 56`) was questioned by an expert annotator: *"Is this enough? I mean you could have a 80 years old male actor like Morgan Freeman or Ian McKellen, and 18 female actor"*). This led to an update to the perturbation function adding 86 instead.
>
>
> Specifically, **missing values** tasks fall under 3 categories in which the desired behavior was objective:
> 1. **Irrecoverable**: In cells required to calculate the answer and the model with no way to recover the value → needs to drop these rows. This is most similar to [3].  Given some operations in code (e.g., `df[‘col’].mean()`) automatically remove missing data and thereby do not test models’ data awareness, these types occur the least.
> 2. **Multiple key fields are missing** →  the row should be ignored. For example, for `daily-activity-calories`, the calorie column is needed for calculation but some rows have the userID, activityDate, and TotalSteps column values all missing (with no other columns indicating activity information).
> 3. **Recoverable:** In cells required to calculate the answer and the value can be recomputed. For example, for the `actor-age-gaps` task, missing values are introduced in the age_difference column. However, this value could be recovered when using  the actor_1_birth_year column actor_2_birth_year columns as all other rows show `age_difference = abs(actor_1_birth_year - actor_2_birth_year)`.
>
> Experiment prompts also emphasized:
>
> “*Attempt to safely recover or correct flawed data when reasonable based on the existing data. If data is irrecoverable or suspect, discard the row.*”
>
> This reinforced a priority of corrective action before dropping missing data and only doing so when justified.
>
> We will incorporate this important discussion on expected model behavior and add qualitative examples to highlight RADAR’s intentional benchmark design for objective evaluation (Sec. 3.1 and 3.2).
>
> &nbsp;
>
> > *It is unclear what type of operations are posed...*
>
> We agree! We categorized operations types and assigned operation sequences for all tasks based on the query and answer function (using GPT-4.1). For brevity, we show GPT-4.1 and o4-mini's performance disaggregated by operation type for direct prompting (DP) and code agent (CA) settings. Tasks in RADAR are not simple lookups, involving an average of 3.2 operations (std=1.6) and covering a range of operation types.
>  |Operation|% Occur|DP: GPT-4.1|DP: o4-mini|CA: GPT-4.1|CA: o4-mini|
> |-|-|-|-|-|-|
> |aggregate|74|14 (8.7, 19)|54 (46, 61)|57 (50, 63)|66 (60, 72)|
> |filter|55|18 (11, 24)|64 (55, 72)|60 (51, 68)|68 (61, 75)|
> |compute column|45|5.8 (2.1, 11)|52 (39, 63)|53 (44, 62)|65 (56, 73)|
> |count|23|18 (7.1, 30)|66 (50, 79)|59 (46, 73)|66 (53, 77)|
> |multi-column filter|21|20 (9.1, 32)|57 (44, 70)|61 (49, 73)|67 (55, 79)|
> |groupby|17|12 (1.9, 25)|56 (42, 70)|51 (36, 64)|63 (49, 76)|
> |rank/sort|15|12 (2.2, 26)|52 (37, 67)|57 (44, 72)|70 (57, 85)|
>
> &nbsp;
>
> > *It would have been interesting to understand if LLMs natively recognize issues in the data…*
>
> Great point! While our current prompt highlights the best capabilities of LMs, it is insightful to observe performance when not explicitly prompted with knowledge of perturbations and perturbation types. We compare the original prompt to a (→)  naive prompt without mentioning perturbations. * indicates significantly worse performance with the naive prompt.
>
> |Model (DP)|Clean|Miss|Bad|Out|Fmt|Log|
> |-|-|-|-|-|-|-|
> |Gemini-2.5-Flash|19→5.7*|11→1.5*|8.7→0.0*|18→0.0*|25→3.0*|12→1.5*|
> |GPT-4.1|20→9.5*|16→7.2*|14→4.9*|8.7→0.0*|21→4.5*|8.7→6.4|
> |Gemini-2.5-Pro|63→66|43→36*|47→33*|54→13*|65→66|34→15*|
> |o4-mini (high)|81→85|49→33*|48→26*|56→6.7*|72→74|28→17*|
>
>  &nbsp;
>
> |Model (CA) |Cln|Miss|Bad|Out|Fmt|Log|
> |-|-|-|-|-|-|-|
> |Gemini-2.5-Flash|91→90|41→30|43→21*|48→7.5*|56→53|30→18*|
> |GPT-4.1|99→99|39→31*|42→23*|53→10*|78→64*|31→25|
> |Gemini-2.5-Pro|91→95|60→30*|59→30*|76→5.8*|80→76|49→18*|
> |o4-mini(high)|99→99|54→34*|54→25*|76→12*|79→75|37→22*|
>
> Except for a few inconsistent formatting cases, models consistently perform worse with naive prompting across all data artifact types. In particular, all models perform substantially worse on tables with outliers **even with code execution**, highlighting current data-awareness gaps and real-world deployment implications.
>
> &nbsp;
>
> > *Various table-aware LLMs have been developed (e.g. TableGPT2, StructLM)…*
>
> Thank you for the suggestion. We evaluated TableGPT2-7B and StructLM-7B. Due to StructLM's specialized prompt format and authors noting it “not designed for agentic settings,” (via email correspondence) we test it only under DP with their table format. TableGPT2-7B is evaluated under both DP and CA.
>
> Initial testing showed both models struggle at with output formatting. Thus, we use a fuzzy match metric: an answer is correct if any ground truth answer appears as a substring in the final LM output. For CA, we treat LM responses with \```python...\``` snippets as execution commands, and if no final output is parsed within the steps, we return the last code output as the answer.
>
>
> |Model (DP)|Clean|Miss|Bad|Out|Fmt|Log|
> |:-|:-|:-|:-|:-|:-|:-|
> |GPT-4.1|20 (14, 29)|16 (9.5, 21)|14 (9.5, 19)|8.7 (2.6, 16)|21 (14, 29)|8.7 (0, 14)|
> |TableGPT2|0.8 (0, 2.4)|0.8 (0, 2.4)|1.1 (0, 2.4)|1.3 (0, 5.3)|1.5 (0, 2.4)|0.8 (0, 2.4)|
> |StructLM|2.3 (0, 4.8)|0.8 (0, 2.4)|0.4 (0, 2.4)|0.4 (0, 2.6)|1.1 (0, 4.8)|1.5 (0, 2.4)|
>
>  &nbsp;
>
> |Model (CA)|Clean|Miss|Bad|Out|Fmt|Log|
> |-|-|-|-|-|-|-|
> |GPT-4.1|99 (95, 100)|39 (31, 45)|42 (33, 52)|53 (45, 61)|78 (69, 86)|31 (21, 38)|
> |TableGPT2|35 (19, 48)|12 (4.8, 21)|3.8 (0, 7.1)|5 (0, 13)|7.5 (2.4, 19)|6 (0, 12)|
>
> These models significantly underperform compared to frontier LMs (e.g., GPT-4.1 with a stricter exact match score), reinforcing RADAR’s difficulty and importance for evaluating additional forms of robustness.
>
> &nbsp;
>
> > *The dataset… is not supported by the code to, for example, generate ground-truth answers or conduct the analysis.*
>
> Due to NeurIPS policies, we can not share links in our rebuttal. However, we will open-source code to generate perturbations and ground-truth answers (note: perturbations may vary with random seed). The supplementary material includes an illustrative example (see `radar/tasks/funcs/influenza_like_illness.py`) of the task construction code.

---

> > ### Comment · Reviewer_YeYx · 2025-08-07
> >
> > Dear authors,
> >
> > Apologies for the delayed response, the discussion period overlaps with being OOO in a wifi-restricted area. I appreciate the clarifications and additional details provided, along with some new experimental results that, in my view, strengthen the completeness and novelty of the benchmark.
> >
> >
> > *Novelty of benchmark and insights:*
> >
> > Thank you for elaborating on these points, I agree on the complementary value of the benchmark and insights, and recommend improving the positioning of the benchmark in context of these prior findings and benchmarks.
> >
> >
> > *Clarification of desired behaviors and tasks:*
> >
> > Thanks for clarifying these points, integrating these clarifications would indeed improve the clarity of the paper and benchmark itself.
> >
> >
> > *Insights regarding (native) LLM behavior and table-tuned models:*
> >
> > I think it is a valuable experiment to see how the LLM natively recognizes and handles these perturbations in the data. Thanks for running these experiments! The experiment with table-tuned LLMs, which seems good to add for completeness, shows a significant gap, informing future work.
> >
> >
> > *Review update:*
> >
> > In response to the sharpened positioning w.r.t. prior studies/benchmarks, task clarifications, and additional experimental insights, I am updating my assessment score.

---

> > > ### Author Response · Authors · 2025-08-07
> > >
> > > Thank you for acknowledging our rebuttal contributions. We appreciate your thoughtful feedback and are glad the clarifications and additions helped strengthen the paper.

---

> ### Author Response · Authors · 2025-08-04
>
> Hi YeYx,
>
> We just wanted to follow-up on our rebuttal to see if you have had a chance to read over our response. We are eager to hear your thoughts and would also appreciate it if you considered updating your review and score accordingly. We realize this is a tight window for discussion and appreciate your time!

---

> ### Author Response · Authors · 2025-08-06
>
> Hi YeYx,
>
> Given there are only 2 days left in the discussion period, we wanted to confirm whether there were any final clarifications you would like. We hope we've addressed your concerns about the novelty and completeness of our work. If our clarifications have been helpful, we'd be grateful if you could update your review and score to reflect that as it would confirm our response has been received and considered. Thank you again for your time and thoughtful engagement in the review process.

---

### Official Review · Reviewer_2KvL · 2025-07-06

**Rating:** 5
**Confidence:** 4

**Summary:**

The paper introduces RADAR, a benchmark designed to assess the data-aware reasoning capabilities of language models over tabular data. Unlike prior datasets that assume clean and well-structured tables, RADAR simulates real-world imperfections such as missing data, outliers, and logical inconsistencies. It includes 2980 table-query pairs across 9 domains and supports controlled perturbations and table size variations. The benchmark is constructed using expert-curated data and programmatic perturbation and answer functions, enabling scalable and reproducible evaluations under diverse artifact scenarios and table configurations.

**Dataset Code Accessibility:**

Yes

**Dataset Code Comments:**

The datasets in the submission is readily accessible, available in a usable format, and well-documented. There is sufficient detail to support reproducibility.

**Ethical Considerations:**

No, there are no or only very minor ethics concerns

**Final Justification:**

Overall, RADAR fills an important gap in the tabular reasoning landscape by introducing a scalable, artifact-rich, and reproducible benchmark. The authors' detailed responses and clarifications further strengthen the case for acceptance.

**Limitations Weaknesses:**

W1. While five artifact types are included, other real-world issues such as duplicated records, schema mismatch, or temporal inconsistencies are not explored. The authors acknowledge this limitation and suggest future extensions, but current coverage may still limit representativeness for some real-world applications.

W2. The use of Exact Match (EM) as the sole evaluation metric may understate partial understanding or close-but-inexact reasoning, especially for ambiguous or open-ended queries. The benchmark could benefit from incorporating relaxed accuracy or robustness-aware metrics, particularly for noisy outputs or semantically equivalent responses.

W3. Despite extensive results, the paper does not report error bars or statistical significance. This makes it hard to assess whether observed performance differences are meaningful or consistent across runs.

W4. The code-agent interaction is limited to five steps, which may constrain the model’s capacity to reason over complex corrections or large tables. There’s also no exploration of adaptive strategies or multi-turn reasoning, which are relevant for real-world agent deployment.

**Strengths Contributions:**

S1. RADAR addresses a critical and under-explored challenge—evaluating LMs' robustness to data artifacts in tabular reasoning. This marks a significant advancement over existing benchmarks that typically assume ideal data conditions.

S2. The authors develop a dataset-agnostic framework for injecting realistic, targeted data artifacts (e.g., missing values, logical inconsistencies), enabling fine-grained control over artifact types and table size. The benchmark supports scaling to larger tables and varied schemas, which is essential given that real-world tabular data is often large and noisy.

S3. The benchmark includes both direct prompting and code-agent settings, revealing stark performance gaps even in SoTA models like GPT-4 and Gemini Pro when faced with noisy data. The experiments highlight nuanced failure modes, such as brittleness to logically inconsistent entries and reduced performance with increasing table size, offering actionable insights for model improvement.

S4. Perturbations and answer functions are programmatically defined, ensuring objective, reproducible evaluations. The perturbation types are clearly defined and grounded in realistic data issues (e.g., bad values like #REF!, unit inconsistencies, time contradictions). The benchmark includes curated subsets (RADAR-T and RADAR-S) to facilitate tractable yet rigorous evaluations.

---

> ### Author Rebuttal · Authors · 2025-07-31
>
> Thank you for the thoughtful feedback and for recognizing the **critical importance** of robust tabular reasoning in the presence of real-world data imperfections. We are especially encouraged by your positive assessment of our dataset-agnostic framework, the realistic nature of the artifacts we introduce, and the value of our reproducible and controlled evaluation setup. Below, we address your concerns in detail.
>
> &nbsp;
>
> > *W1 …other real-world issues such as duplicated records, schema mismatch, or temporal inconsistencies are not explored…*
>
>
> We agree that the current set of data artifact types do not yet cover the full range of real-world scenarios present in real-world data. However, some of these cases mentioned are **already partially addressed** in RADAR or related work.
>
> * Temporal inconsistencies are featured in several tasks:
>   * `uae-cancer-patient`: Some rows contain a Diagnosis_Date that is after the Treatment_Start_Date.
>   * `nyc-green-taxis-passengers`: Perturbations make the pickup time (lpep_pickup_datetime) after the dropoff time (lpep_dropoff_datetime).
>   * `ultra-trail-races-morning-finishers` (Failure Case 2 in Appendix, line 37): Perturbations add 12 hours to the finish_time, breaking the existing, consistent relationship with start_time + time = finish_time.
> * Schema mismatch is reflected at least partially in RADAR through inconsistent formatting or bad value artifacts where corrupted values introduce type inconsistencies or unexpected formats (e.g., strings in numerical columns).
> * While duplicate records are not explicitly included in RADAR, they have been explored in prior work (e.g.,[1]), and we view this as complementary to our contributions.
>
> That said, given these artifact types are not introduced consistently across tasks we do not emphasize them in our analysis. Importantly, the RADAR framework is **modular and extensible**, enabling future inclusion of such artifact types in a systematic and controlled manner. These are important directions we leave for future work (consistent with our benchmark focus on complex data artifacts that may require reasoning across multiple columns). We will extend our discussion of RADAR’s limitations at the end of Section 5 to include examples of other artifact types. In Table 2 of the Appendix, we will indicate tasks that exhibit temporal inconsistencies.
>
> [1] Wolff, Cornelius, and Madelon Hulsebos. "How well do LLMs reason over tabular data, really?." arXiv preprint arXiv:2505.07453 (2025).
>
> &nbsp;
>
> > *W2. The use of Exact Match (EM) as the sole evaluation metric may understate partial understanding or close-but-inexact reasoning, especially for ambiguous or open-ended queries…*
>
> We share your concern that EM can be overly strict, especially for open-ended or ambiguous queries. To address this we carefully design tasks to be unambiguous and validate them through expert review (see examples in our responses to YeYx and kvSF). If multiple answers are equally valid, we include all acceptable options in the ground truth and count any match as correct.
>
> For floating point answers, we accept a margin of ±1 unit in the least significant digit (line 280). For example, in `weather-city-mixup`, with query "*What is the difference in temperature between Australia's warmest city and America's warmest city in February?*", the ground truth is any of {15.1, 8.4, -15.1, -8.4}, allowing for Fahrenheit/Celsius calculations and both subtraction directions. We also accept values in the ranges [15.0,15.2], [8.3,8.5], [-15.2,-15.0], and [-8.5,-8.3] to account for rounding. We will clarify this in the paper.
>
> While EM may not capture partial understanding, for RADAR’s **well-specified, and clearly solvable tasks**, we believe EM is a meaningful and fair metric, especially in the context of evaluating models as autonomous data analysis agents in high-stakes domains (e.g., in scientific discovery). LMs *should* solve RADAR’s tasks completely and reliably.
>
> Nevertheless, when evaluating table-tuned LMs (i.e., StructLM and TableGPT2 in additional experiments to address YeYx’s comments), we find table-tuned LMs sometimes struggle with output formatting, impacting our ability to measure correct reasoning. For these LMs, we apply a fuzzier matching criterion: if any acceptable answer is present in the output string, we count the prediction as correct. We will document this relaxation explicitly in the paper, provide examples, and discuss its implications.
>
> &nbsp;
>
> > *W3. Despite extensive results, the paper does not report error bars or statistical significance…*
>
> Thank you for pointing this out. We agree that uncertainty estimates are important. To address this, we have enhanced our evaluation with both confidence intervals and statistical significance testing, applied to all main experiments and new analyses introduced in the rebuttal.
>
> To account for both **model randomness** and **data sampling variability**, we reran each model five times with different random seeds. For each run, we generated bootstrap samples by subsampling 80% of the data. We then pooled all bootstrap means across runs and reported the 95% confidence interval. To assess significance of performance drops of perturbed data tables, we ran one-sided paired t-tests over the five run-level means (clean vs. perturbed). Results with p < 0.05 are marked with *.
>
> For space, we show results on RADAR-T on four frontier models. Note: Gemini-2.5-Flash scores differ slightly from the submission due to the deprecation of gemini-2.5-flash-preview-04-17 (used during the submission) on July 15, 2025.
>
> **Direct Prompting**
> | Model            | Clean       | Miss          | Bad            | Out            | Fmt          | Log           |
> |------------------|-------------|---------------|----------------|----------------|--------------|---------------|
> | Gemini-2.5-Flash | 19 (12, 26) | 11 (4.8, 19)* | 8.7 (4.8, 14)* | 18 (11, 26)    | 25 (19, 33)  | 12 (4.8, 19)* |
> | GPT-4.1          | 20 (14, 29) | 16 (9.5, 21)* | 14 (9.5, 19)*  | 8.7 (2.6, 16)* | 21 (14, 29)  | 8.7 (0, 14)*  |
> | Gemini-2.5-Pro   | 63 (55, 71) | 43 (33, 52)*  | 47 (33, 60)*   | 54 (45, 61)*   | 65 (57, 71)  | 34 (24, 45)*  |
> | o4-mini (high)   | 81 (71, 91) | 49 (36, 60)*  | 48 (38, 57)*   | 56 (45, 71)*   | 72 (64, 83)* | 28 (19, 38)*  |
>
> &nbsp;
>
> **Code Agent**
> |Model|Clean|Miss|Bad|Out|Fmt|Log|
> |:-|:-|:-|:-|:-|:-|:-|
> |Gemini-2.5-Flash|91 (83, 98)|41 (32, 52)*|43 (36, 51)*|48 (40, 55)*|56 (48, 64)*|30 (21, 41)*|
> |GPT-4.1|99 (95, 100)|39 (31, 45)*|42 (33, 52)*|53 (45, 61)*|78 (69, 86)*|31 (21, 38)*|
> |Gemini-2.5-Pro|91 (86, 95)|60 (50, 67)*|59 (50, 67)*|76 (66, 87)*|80 (71, 88)*|49 (40, 57)*|
> |o4-mini (high)|99 (95, 100)|54 (45, 62)*|54 (45, 67)*|76 (71, 82)*|79 (69, 88)*|37 (29, 45)*|
>
> We note that the absence of these error bars did not affect any conclusions drawn in the original version. Specifically, for nearly every data artifact type across all models and baseline settings (including all cases for the code agent setting) we observe models perform significantly worse on the perturbed versions of the tables.
>
> In the camera-ready version, we will:
> 1. Include **bootstrapped 95% confidence intervals** with updated results for the main experiments (Table 2) and all new analyses in this rebuttal
> 2. Report **paired** **t-test results** for all clean vs. perturbed comparisons.
>
> &nbsp;
>
> > *W4. The code-agent interaction is limited to five steps, which may constrain the model’s capacity to reason over complex corrections or large tables…*
>
>  We agree that exploring richer agent architectures is an exciting direction. This is also a key reason why we chose to position this paper as a dataset and benchmark paper, inviting other researchers to contribute to this development and exploration. Our current design follows a simple yet widely used agent framework (e.g., [41, 23]), to keep the evaluation controlled and reproducible. Importantly, we found that the five-step limit rarely acted as a bottleneck: the agent reached the step cap in only 2.7% of evaluated cases. We will include this statistic in the paper to clarify the limited practical impact of the step cap.
>
> Nevertheless, given our current results, we see agents that intelligently coordinate between text tabular reasoning and code execution as an exciting avenue for future research.

---

> > ### Author Response · Authors · 2025-08-04
> >
> > Hi 2KvL,
> >
> > We just wanted to follow-up on our rebuttal to see if you have had a chance to read over our response.  We are eager to hear your thoughts and would also appreciate it if you considered updating your review accordingly. We realize this is a tight window for discussion and appreciate your time!

---

> > ### Comment · Reviewer_2KvL · 2025-08-05
> >
> > Thank you for your detailed responses and clarification. I'll keep my score given it's already positive.

---

### Note · Authors · 2025-08-12

We thank all reviewers for acknowledging our contributions and providing insightful feedback!
We have carefully considered and responded to all concerns raised. In response, we have substantially strengthened our work by:

1. **Emphasizing RADAR’s positioning** as a necessary benchmark, complementary to existing efforts, for studying robust tabular reasoning in the introduction and related work (clarifications based on YeYx's comments). RADAR expands the field's scope beyond structural perturbations by emphasizing semantically-grounded, table-informed data artifacts.
2. **Expanding our discussion on expected model behavior** given a perturbed table as highlighted by YeYx. In addition, we address kvSF’s concern by clarifying our methodology for constructing “objectively” bad tables. We provide additional qualitative examples to underscore the intentional benchmark design to achieve an objective evaluation (Sec. 3.1 and 3.2).
3. **Incorporating extensive additional experiments and analysis (Sec. 4 and 5)** involving (1) prompts that do not explicitly mention potential perturbations and (2) table-tuned LLMs suggested by YeYx. Our updated results reiterate the challenge posed by domain-specific common-sense reasoning artifacts introduced in RADAR.
4. **Enhancing the statistical rigor of our evaluations** (as suggested by 2KvL) by (a) reporting bootstrapped confidence intervals for exact match scores in the main results (Table 2) and new experiments, and (b) conducting statistical significance tests (paired t-tests) to quantify performance drops on perturbed task instances. All experiments, including new analyses introduced in the rebuttal, were repeated across 5 random seeds to account for model variability and ensure robustness of findings.

We appreciate the constructive feedback and are confident RADAR will serve as a valuable benchmark for advancing robust tabular reasoning research in the community.

---

### Decision · Program_Chairs · 2025-09-18

**Decision:**

Accept (poster)

**Comment:**

The paper presents a dataset designed to evaluate the performance of large language models (LLMs) in analyzing malformed tables. This dataset compiles tabular data curated by domain experts from existing sources and employs a deterministic mechanism to introduce errors from various categories, such as missing cells or logical inconsistencies with other columns. The findings reveal that state-of-the-art LLMs experience a significant decline in performance when working with malformed tables compared to clean, well-structured ones.

This paper tackles a crucial and under-explored challenge: assessing the robustness of language models to data artifacts in tabular reasoning. This paper represents a substantial advancement over existing benchmarks, which generally assume ideal data conditions.

===== FINAL UPDATE FROM DB Track PCs ====

The final decision for this paper has been taken by the program chairs after consultation with the SACs. All Senior Area Chairs have ranked papers according to the feedback from the AC during the review process. We decided to leave the original meta-review to reflect the opinion of the AC in light of the initial discussions with reviewers and SAC.